## Registered report

cognition/psychology/health and disease and epidemiology

numeracy, health numeracy, COVID-19, health policy and adherence

**Author for correspondence:**
Daniel Ansari
e-mail: daniel.ansari@uwo.ca

# Numeracy and COVID-19: examining interrelationships between numeracy, health numeracy and behaviour

Nathan T. T. Lau[1], Eric D. Wilkey[1], Mojtaba Soltanlou[1], Rebekka Lagacé Cusiac[1], Lien Peters[1], Paul Tremblay[1], Celia Goffin[1], Isabella Starling Alves[2], Andrew David Ribner[3], Clarissa Thompson[4], Jo Van Hoof[5], Julia Bahnmueller[6], Aymee Alvarez[1], Elien Bellon[7], Ilse Coolen[8], Fanny Ollivier[9] and Daniel Ansari[1]

[1]Department of Psychology, Western University, Canada
[2]Department of Educational Psychology, University of Wisconsin-Madison, Madison, WI, USA
[3]Learning Research and Development Center, University of Pittsburgh, Pittsburgh, PA, USA
[4]Department of Psychological Sciences, Kent State University, Kent, OH, USA
[5]Centre for Instructional Psychology and Technology, KU Leuven, Belgium
[6]Centre for Mathematical Cognition, Loughborough University, UK
[7]Parenting and Special Education, KU Leuven, Belgium
[8]Université de Paris, LaPsyDÉ, CNRS, F-75005 Paris, France
[9]Laboratoire de Psychologie, Cognition, Comportement et Communication, Université Rennes 2, France

 RLC, 0000-0003-1322-7792; EB, 0000-0002-0607-6634; FO, 0000-0003-3235-5685; DA, 0000-0002-7625-618X

During the COVID-19 pandemic, people across the globe have been exposed to large amounts of statistical data. Previous studies have shown that individuals' mathematical understanding of health-related information affects their attitudes and behaviours. Here, we investigate the relation between (i) basic numeracy, (ii) COVID-19 health numeracy, and (iii) COVID-19 health-related attitudes and behaviours. An online survey measuring these three variables was distributed in Canada, the United States (US) and the United Kingdom (UK) ($n = 2032$). In line with predictions, basic numeracy was positively related to COVID-19 health numeracy. However, predictions, neither basic numeracy nor COVID-19 health numeracy was related to COVID-19 health-related attitudes and behaviours (e.g. follow experts' recommendations on social distancing, wearing masks etc.). Multi-group analysis was used to investigate mean differences

and differences in the strength of the correlation across countries. Results indicate there were no between-country differences in the correlations between the main constructs but there were between-country differences in latent means. Overall, results suggest that while basic numeracy is related to one's understanding of data about COVID-19, better numeracy alone is not enough to influence a population's health-related attitudes about disease severity and to increase the likelihood of following public health advice.

## 1. Introduction

The outbreak of the COVID-19 pandemic is an unprecedented event. While human history is dotted with pandemics, never before have we been able to track and model a disease with such sophistication. One of the many ways in which COVID-19 represents a watershed moment is the way in which it has pushed people across the planet to process and interpret rapidly evolving numerical information to inform their behaviour. Websites reporting COVID-19 statistics, such as total number of infections and total number of deaths, have been created (e.g. https://coronavirus.jhu.edu) and are widely cited in media reports on COVID-19, leading to unprecedented levels of attention to dynamic numerical information among the general population.

In addition to the numerical information pertaining to the consequences of the virus (e.g. number of cases, number of deaths), people across the planet are being introduced to unfamiliar and challenging mathematical concepts, such as 'flattening the curve', 'exponential growth', 'false negative rates' of COVID-19 tests, etc. Specifically, the degree to which individuals can make sense of COVID-19 numbers may influence the way they understand these critical health-related concepts which may, in turn, influence their adherence to public health advice and their perception of the risks posed by COVID-19.

Indeed, several experts in the study of mathematical cognition have recently discussed the role that numeracy (e.g. understanding proportions, large numbers) may play in how individuals estimate the risks associated with COVID-19 [1–3]. For example, an online experimental study with over 1200 US participants found that those exposed to a short educational intervention consisting of a worked example with instructions on how to calculate fatality rates were more accurate on post-intervention health decision-making problems than those participants who were randomly assigned to a control condition [3]. Importantly, COVID-19 risk perceptions and worry increased throughout the 10-day study for those in the intervention condition, relative to those in the control condition. Since risk perception inherently involves magnitude comparisons (i.e. higher versus lower risk), improving proportional understanding may have increased the accuracy involved in assessing risk magnitude. However, there are still open questions pertaining to the ways in which numeracy and understanding of health information related to COVID-19 may affect people's behaviour in response to the pandemic. Therefore, the aim of the present study is to address these knowledge gaps. In what follows, we first review the mathematical concepts and misconceptions that are relevant to understanding COVID-19-related information. We then go on to discuss how we aim to study the interrelations between (i) basic numerical processing and understanding (hereafter basic numeracy), (ii) health numeracy directly relevant to COVID-19 (hereafter COVID-19 health numeracy), and (iii) COVID-19 health-related attitudes and behaviours.

During the COVID-19 pandemic, policy makers and public figures frequently cite numerical information in order to defend decisions and influence public action. For example, some have argued against radical actions (e.g. stay-at-home orders and business closures) on the premise that equally large numbers of people die from other causes, such as automobile accidents, every year without causing nationwide shutdowns [4]. As of 7 April 2020 (nearly a month after the first COVID-19 death was recorded in the US), John Hopkins University reported 31.4 thousand COVID-19-related deaths in the US compared with the 39.4 thousand deaths reported by the National Safety Council due to car accidents in the US over the course of 2018. However, comparing the current number of COVID-19-related deaths with the number of people who died in car accidents in a given year is like comparing apples and oranges, in part because both statistics have different denominators (i.e. one is per month while the other is per year). When taking the time period into account, we can observe that 31.4 thousand people died of COVID-19 while approximately 3 thousand people died of car accidents in the same one-month time period ($39\,000/12 = 3250$), leading us to draw drastically different conclusions about the severity of the pandemic. This example shows how important relational information (i.e. proportions) is when making decisions surrounding COVID-19.

Despite the importance of understanding relational information, there is a wealth of empirical evidence to show that processing proportions (fractions, decimals, percentages or odds ratios) are challenging for many adults [5,6]. What underlies these difficulties in processing proportions? One of the possible sources is the whole number bias. The whole number bias refers to the inappropriate use of whole number properties (i.e. natural numbers) when processing rational numbers (e.g. fractions and decimals; [7]). In the specific case of fractions and odds ratios, the whole number bias also manifests when individuals pay more attention to the components of the proportion (i.e. the numerator and/or denominator) rather than to the magnitude of the entire ratio (i.e. the relation between the numerator and denominator; [8,9]). For example, an individual might perceive a chance of 1 in 10 as being smaller than 10 in 100 simply because the numbers of the first figure (i.e. 1 and 10) are smaller than the numbers in the second figure (i.e. 10 and 100) although they represent the same proportion (i.e. 0.1).

In applied settings, paying attention to the magnitude of components rather than to the magnitude of the entire proportion can lead to other types of related biases, such as denominator neglect. Denominator neglect occurs when individuals compare two or more numbers of events without taking into account the total number of opportunities for that event to occur [6]. In the context of the COVID-19 pandemic, examples of this bias are common in the media when public figures state that the US has carried out more tests than any other country while ignoring the fact that the number of tests per capita (i.e. the number of tests accounting for population size) is much lower than several other countries at the time of writing. In addition to these misleading claims, other factors, such as how statistics are presented in the media, can also increase the occurrence of denominator neglect in the general population. For example, the media often reports, either directly or with the use of graphs, absolute magnitudes such as the total number of COVID-19 cases. Although this information is important for health planning (e.g. estimating the number of ventilators needed), it also puts focus on the numerator and leads to individuals comparing different regions to ignore total population size.

As is evident from the above, a strong understanding of proportions in a variety of formats is necessary in order to accurately understand COVID-19-related information. Moreover, the size of the numbers that are contained within COVID-19-related data may be a barrier to understanding that information and acting accordingly. For instance, it has been shown that many individuals have misrepresentations when it comes to their understanding of large numbers (i.e. in the order of thousands, millions and billions; [10,11]). Failing to grasp important magnitude differences between large numbers could result in altered perceptions, such as minimizing the impact of the pandemic on a larger scale. Furthermore, processing proportions with large numbers may create additional cognitive load, even when individuals have an accurate representation of large numbers [12]. Lastly, large numbers in COVID-19 data are a result of the virus's exponential spread, which is another concept that is poorly understood, since people are more often exposed to linear (or arithmetic) growth [13]. All of these factors may translate into difficulty digesting complex mathematical information related to large-scale numbers associated with COVID-19. As a result, poor understanding of these concepts may lead people to undermine efforts to slow the spread of the disease by not following social distancing guidelines.

The review of the literature above suggests that having difficulties understanding mathematical concepts and accurately processing numerical quantities may influence the way in which individuals understand and act upon COVID-19-related information. Indeed, it has been shown that individuals with higher numeracy skills are more likely to pay attention to numerical information, interpret it correctly and make decisions accordingly relative to those with low numeracy skills [14]. By contrast, individuals with relatively low numeracy skills are more susceptible to bias. For example, they are more likely to ignore numerical information and instead rely on intuitions based on emotional states and other extraneous factors, such as their trust in, or distrust of, the information source [14,15]. Furthermore, numeracy has been shown to influence how people evaluate risk [16–18]. For example, individuals with higher numeracy levels have been shown to respond differently to high-risk versus low-risk situations while individuals with low numeracy levels do not [16]. Another study on numeracy and adaptive decision-making found that individuals with low numeracy skills were more likely to choose riskier options when small losses were inevitable [19].

To the best of our knowledge, there has, to date, been no study that has examined the role of basic numeracy alongside applied, health-related numeracy on health outcomes. However, evidence from disparate areas of psychology and health informatics suggests that the three main variables, basic numeracy, COVID-19 health numeracy and COVID-19 health-related attitudes and behaviours are related.

First, there is a wealth of evidence that suggests that those with stronger basic numeracy are better equipped to apply this knowledge to diverse and novel contexts [20–22]. Given that public information about COVID-19 disseminated in the media requires the understanding of numerous mathematical concepts that are typically not encountered in the day-to-day, it is likely that those in the population that possess stronger basic numeracy would have a stronger grasp of the risks relating to the pandemic. Second, there exist many studies on health literacy, which have examined the association between applied mathematical skills, such as health- and non-health-related word problems, and health-related decisions and outcomes. Those who perform better at these problems tend to have better health outcomes in the future [23,24]. Finally, stronger basic numeracy may have an independent effect on health outcomes controlling for health numeracy. At an equal level of health numeracy, those with stronger basic numeracy may be more capable of understanding the underlying numerical concepts of COVID-19 health information. As such, they would be better equipped in spotting common pitfalls to which those less familiar with the underlying numerical concepts may fall prey.

The current study directly addresses this critical knowledge gap by assessing the relation of both (i) basic numeracy and (ii) COVID-19 health numeracy to (iii) COVID-19 health-related attitudes and behaviours in a correlational study. Basic numeracy involves numerical magnitude processing and conceptual understanding of rational numbers (e.g. proportions). COVID-19 health numeracy is our measure of health numeracy relevant for understanding numerical information about COVID-19 (e.g. risk factors and flattening the curve). And lastly, COVID-19 health-related attitudes and behaviours index people's perception of, and adherence to, policies such as social distancing and handwashing. It is hypothesized that all three variables are positively related with one another. Specifically, that basic numeracy is positively correlated with COVID-19 health numeracy, that COVID-19 health numeracy is positively correlated with COVID-19 health-related attitudes and behaviours, and that basic numeracy is positively correlated with COVID-19 health-related attitudes and behaviours.

The primary aims of the current study are (i) to assess the interrelations among basic numeracy, health numeracy and health-related attitudes and behaviours and (ii) determine whether the relations among the three variables differ across three different countries: Canada, the US and the UK. These countries were chosen because (i) they are all members of the G7 and thus have comparable economies and political systems, and (ii) the response to COVID-19 (e.g. when lockdown and re-opening measures were implemented) as well as the case fatality and testing rates differed between the three countries. Furthermore, the majority language for all three countries is the same (i.e. English), thereby limiting potential confounds driven by linguistic differences in number representation [25]. Investigating the between-country differences in the strength of the correlations and between-country differences in how covariates may influence this relation will allow us to comment on whether and how the strength of the relations among the variables differ across countries, which can inform short-term government policies related to COVID-19. For a full summary of the study hypotheses considered, corresponding analyses, contingencies and resulting interpretations in graphical form, see electronic supplementary material, table S1.

# 2. Methods

## 2.1. Participants

After removal of participants according to the preregistered criterion (see Outlier identification and Missing data), the final sample consists of 2032 participants who completed an online survey hosted by Qualtrics. The final sample consists of 680 participants (296 females) from the US, 677 participants (356 females) from Canada, and 675 (359 females, 1 non-binary) from the UK. Other relevant demographic information is reported in the results.

## 2.2. Procedure and materials

Qualtrics collected the data using their participant panels in December 2020. The survey took approximately 20 min to complete and included four sections: demographics and other cultural variables, basic numeracy, COVID-19 health numeracy and COVID-19 health-related attitudes and behaviours. The survey started with the consent form, followed by the demographics section for all respondents. The order of the other three sections (i.e. basic numeracy, COVID-19 health numeracy and COVID-19 health-related attitudes and behaviours) was randomized across respondents (figure 1). The

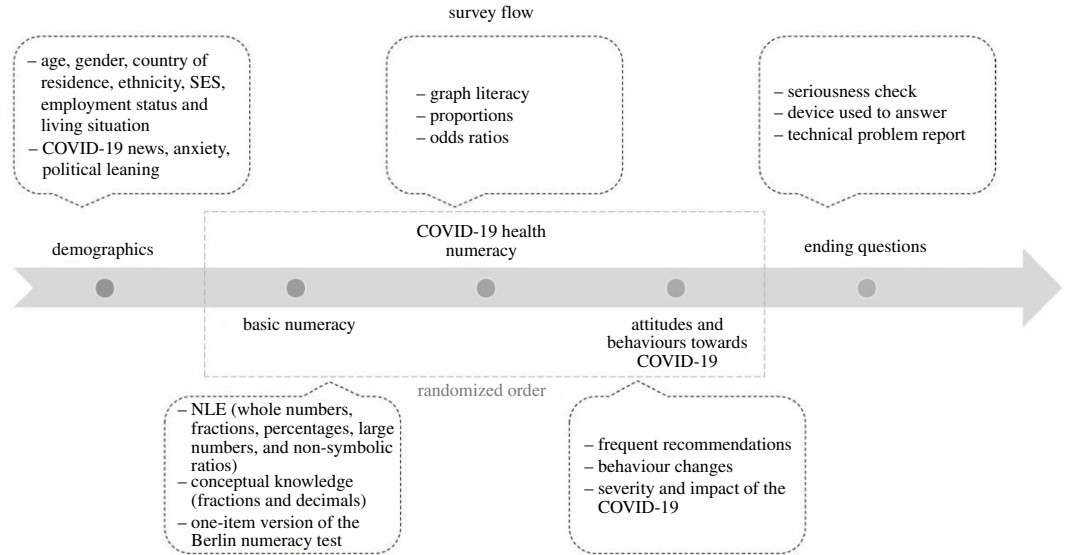

**Figure 1.** Overview of the flow of the survey. SES = socioeconomic status. NLE = number line estimation.

survey ended with a few questions about the device that they used, a seriousness check (in which participants were asked whether they would keep their data if they were the experimenter), and a question about any technical problems they may have faced. For a list of all items, see supplementary material (https://osf.io/qpdnt/).

Additionally, it should be noted that the current study is a registered report with a preliminary measurement study conducted as part of the revision process. Survey, data and results for the preliminary measurement study can be accessed in the electronic supplementary material.

## 2.3. Demographics and other potential covariates

Detailed demographic information was collected using 14 items covering age, gender, country of residence, ethnicity, socioeconomic status (SES; [26]), employment status (adapted from the World Values Survey, WVS-6; [27]) and living situation. In addition to these demographic variables, a short list of potential moderating or confounding variables was collected to test for their possible effect on variables of interest, namely COVID-19 news consumption (three items), anxiety (three items), political leaning (one item). For behaviours related to news consumption, participants indicated how often they accessed news sources (e.g. news websites, government communication) or social sources (e.g. social media, friends and family) for news on a scale of 1 (daily) to 5 (less than monthly) and how well informed they felt about the pandemic on a scale of 1 (not informed) to 4 (very informed). The items measuring anxiety were adapted from the single-item math anxiety scale [28] and measured general anxiety [29], anxiety related to COVID-19 and math anxiety. Participants rated their anxiety level on a scale of 1 (not anxious) to 10 (very anxious). For political leaning, participants were asked to rate their beliefs on a visual analogue scale with 'left/liberal' label at the left end and 'right/ conservative' label at the right end (adapted from the WVS-6; [27]). Finally, information about the type of electronic device used to complete the survey was collected due to the nature of the response to the number line tasks. It was expected that participants would use either computers or mobile devices to respond to the questionnaire, meaning that the total length of the number lines would vary from one device to the other. Therefore, information about the device used to answer the survey was collected to test for differences in number line accuracy due to screen size. Demographics items were presented in a fixed order as no order effect was expected.

## 2.4. Basic numeracy

The basic numeracy measure is designed to capture elements of numeracy that are relevant for understanding health information related to COVID-19 and includes a total of 30 items. It was created by adapting and combining four existing measurement tools focused on the understanding and processing of proportions and large numbers.

First, numerical magnitude processing was measured using the number line estimation task (NLE), a task that has been consistently found to be correlated with different mathematical skills [30]. In this task, the participant was presented with an empty number line bound on each side. The participant was given a number and asked to indicate its location on a number line with respect to the two endpoints. Five versions of the task were used to assess participants' understanding of whole numbers (six items), fractions (six items), percentages (three items), large numbers (five items), and non-symbolic ratios (three items). For all number line items, accuracy was scored using percentage absolute error (PAE), which is the absolute difference between the estimated and the correct answer divided by the scale of the number line. Therefore, a higher PAE indicates lower accuracy (i.e. estimation is further away from true value), while a low PAE indicates higher accuracy. Stimulus numbers were selected because they showed suitable variability for capturing individual differences in previous research [3,10]. For all five number line versions, at least three of the items were matched for relative position on the number line to provide points of comparison across number line types.

Second, conceptual knowledge of fractions and decimals was measured using two multiple-choice items and two fill-in-the-blank items selected from the fraction-knowledge assessment [31] in addition to two open-ended items selected from the rational number sense test [32]. Whereas number line items assess the processing of specific magnitudes, these items assessed broader conceptual understanding of rational numbers, such as fractions and decimals (e.g. How many possible fractions are between 1/4 and 1/2?). These six items were scored correct (1) or incorrect (0). Lastly, the one-item version of the Berlin numeracy test [33] was included in the basic numeracy measure. Although it has been proven as a valid and reliable test of risk literacy, it is often used as a measure of general numeracy and was used to validate the novel basic numeracy measure described in this section. The item included in this survey has been shown to discriminate participants roughly into a top and bottom half of risk literacy in a number of samples across several countries [33]. The item was scored correct (1) or incorrect (0).

Basic numeracy items were presented pseudo-randomly: the items were randomly presented within a group of similar items to reduce task switching. For example, all items consisting of non-symbolic ratios were presented on the same page in a random order and were not mixed with other variants of the number line task. A complete list of items can be found at: https://osf.io/qpdnt/).

## 2.5. COVID-19 health numeracy

COVID-19 health numeracy measured understanding of COVID-19 concepts and statistics using a custom-made questionnaire composed of 18 multiple-choice items. Items covered three concepts important to understanding COVID-19-related data in the media: graph literacy, proportions and ratios. Fictional data were used in all sections in order to avoid bias due to prior COVID-19 knowledge. All items were multiple-choice format and were scored correct (1) or incorrect (0). Therefore, a higher score indicates high understanding of COVID-19-related concepts and statistics, whereas a lower score indicates poor understanding of COVID-19-related concepts and statistics.

Nine items measured graph literacy. Of these, five items were conceptually modelled after the graph literacy scale proposed by Galesic & Garcia-Retamero [34]. Individuals were shown a graph representing the total number of confirmed COVID-19 cases over time for two fictional countries, one with linear growth and the other with exponential growth. Four of these items had one of three difficulty levels related to graph literacy. The first difficulty level consisted of retrieving information, such as data points, directly from the graph (one item). The second difficulty level consisted of evaluating relationships between data points, for example, by finding where two curves intersect each other (two items). The third level consisted of making inferences from the information given by the graph, without the information being directly observable in the graph (one item). Also, one item measured explicit knowledge of linear and exponential functions. The last four graph literacy items tested individuals' knowledge of 'flattening the curve' and were accompanied by a diagram showing curves for the number of cases over time, with and without social distancing. Questions tested participants' understanding of the effects of social distancing on various aspects of potential infection rate scenarios, such as length of outbreak and the height and latency of the peak of infection of the pandemic, based on the diagram.

Six items measured the ability to accurately process absolute and relative magnitudes in the context of a highly infectious disease. A table containing fictional data about COVID-19 (i.e. number of confirmed cases, number of deaths, number of tests and total population) for three fictional countries were

presented. Similar to the section on graph literacy, items had one of two difficulty levels. The first difficulty level assessed the ability to retrieve relevant data from the table (e.g. 'Which country has conducted the most COVID-19 tests?'; three items). The second difficulty level assessed the ability to obtain relevant proportions from the data in order to accurately compare country statistics (e.g. 'Which country has the highest fatality rate for COVID-19?'; three items).

Finally, three items measured participants' ability to compare risk ratios. A situation describing fictional fatality rates for COVID-19 alone as well as for two other hypothetical risk factors were presented using risk ratios. Items involved comparing the odds ratios associated with the different risk factors and calculating relative risk.

Similar to items in the basic numeracy section, COVID-19 health numeracy items were presented pseudo-randomly. In other words, items within a subsection (e.g. items about flattening of the curve) were always presented together in a random order to reduce task switching. For a list of items, see the supplementary material (https://osf.io/qpdnt/).

## 2.6. Attitudes and behaviours toward COVID-19

Attitudes and behaviours toward COVID-19 were measured using a total of 17 items. The first 12 items focused on four frequently issued recommendations by health authorities (e.g. WHO): washing hands frequently and thoroughly, staying home unless travel is essential, social distancing and wearing a mask in public. For each recommendation, participants were asked to indicate (i) when applicable, the extent to which they have followed the respective recommendation on a scale of 1 (Never) to 10 (Consistently all the time), and (ii) how useful they think this recommendation is in the fight against the COVID-19 pandemic on a scale of 1 (Completely useless) to 10 (Extremely useful). Next, two items measured participants' perceived change in (i) their own behaviour and (ii) the behaviour of those people around them (e.g. family, close friends) in response to the COVID-19 pandemic. Participants responded to both items on a scale of 1 (Not at all) to 10 (To a great extent). The last three items assessed participants' perceptions about the severity and impact of the COVID-19 pandemic by assessing the degree to which they believed the COVID-19 pandemic is a serious global threat, COVID-19 is a serious medical condition, and that the benefits of the recommended actions to fight the COVID-19 pandemic outweighed their psychological, economic and cultural costs. For these items, participants responded on a scale of 1 (Not at all) to 10 (To a great extent).

Items of attitudes and behaviours toward COVID-19 were presented in a fixed order to reduce bias in response selection as respondents might justify their responses about their own behaviours based on the local authorities' recommendations. For a complete list of items, see the supplementary material (https://osf.io/k4jy9/).

## 2.7. Power analysis

The current study focused on the interrelations among three main constructs of interest, basic numeracy, COVID-19 health numeracy and COVID-19 health-related attitudes and behaviours. We were particularly interested in the robustness of these intercorrelations controlling for the covariates, and whether there were any between-country differences in these intercorrelations. For the following power analysis, all α-levels are 0.05. We calculated the necessary sample size to achieve 95% power using multiple Monte Carlo simulation analyses in Mplus with 10 000 replications. As the relations between basic numeracy, COVID-19 health numeracy and COVID-19 health-related attitudes and behaviours have yet to be explored, using effect sizes from previous literature for the purpose of power calculations was not feasible. Most of the values in the following analyses were chosen based on reasonable expectation of correlations or the minimum effect size of interest. However, in the multi-group analysis, we decided on detecting a standardized correlation coefficient difference of 0.2 based on practical concerns, as the number of participants needed to detect a smaller difference would make the number of participants required prohibitively large.

### 2.7.1. Simple correlation model

To calculate the power to observe a correlation between the main variables of interest, we simulated a model with three variables with mean of 0 and variance of 1 and all intercorrelations set to be 0.14. A correlation coefficient of 0.14 corresponds to a small effect size [35]. A Monte Carlo simulation

revealed that a total of 675 participants per country (i.e. a total *N* of 2025) would be sufficient to achieve 95% power for each correlation coefficient.

### 2.7.2. Measurement invariance of the main variables

As the current study operationalized the main constructs as latent variables (see Main analysis**)**, and we employed multi-group analysis, measurement invariance between the countries was considered. To test for measurement invariance, we took a number of steps to test whether the factor/latent variable structure was indifferent across countries. These include (i) configural invariance—the overall factor structure (i.e. the number of factors and their indicator variables), (ii) metric invariance—the loadings of the indicator variables, (iii) scalar invariance—the intercepts of the indicator variables, and (iv) strict measurement invariance—the residual variances of the indicator variables [36]. The presence of invariance and non-invariance is often informative, and hypotheses were considered (for a full decision tree see electronic supplementary material, table S1).

Calculations of power for measurement invariance are typically not calculated. This is because violations of different levels of measurement invariance are typically assessed using model fit indices (e.g. comparative fit index and root mean square error of approximation), and these indices are indifferent to sample size. For all models below, a Monte Carlo simulation study was conducted across two countries. For each country, we simulated three latent factors with mean of 0 and variance set to 1. Intercorrelations between all factors were set to 0.14. Each latent variable contained four indicators with factor loading of 0.5, residual of 0.75 and intercept of 3. A factor loading of 0.5 for each indicator variable can be interpreted as a standardized coefficient, which corresponds to 25% explained variance and 75% residual variance.

*Configural invariance*. To simulate a violation of configural invariance, one indicator item for factor 1 from the second country was instead loaded on to another with a factor loading of 0.5. Power was calculated as the proportion of simulations in which the loading of the errant indicator item was not statistically significant. Results indicated 675 participants per country would be sufficient for 95% power to detect a violation.

*Metric invariance*. To simulate a violation of metric invariance, one item each from factor 2 and factor 3 was loaded on their respective factors at 0.2 instead of 0.5 (see electronic supplementary material, table S1 for hypotheses). Power was calculated as the proportion of simulations in which the chi-square difference test between the configural invariant and metric invariant model was significant. Results indicated 675 participants per country would be sufficient for 95% power to detect a violation.

*Scalar invariance*. To simulate a violation of scalar invariance, one item intercept from factor 1 was set to 4 instead of 3. Power was calculated as the proportion of simulations in which the chi-square difference test between the metric invariant and scalar invariant model was significant. Results indicated 675 participants per country would be sufficient for 95% power to detect a violation.

*Latent mean differences*. To simulate latent mean differences, the latent mean of the second group was set to be 0.3—a small to medium effect size [37]. Power was calculated as the proportion of simulations in which a chi-square difference test was obtained when testing between a model in which latent means were restricted to be the same and another model in which latent means were allowed to be estimated freely. Results indicated 675 participants per country would be sufficient for 95% power to detect a difference of 0.3.

### 2.7.3. Multi-group analysis

We simulated the analyses that will be used to address whether there are country-level differences in the correlation between the main variables by modelling a multi-group analysis. Using the simple correlation model above, we stipulated a model with the same number of participants in each country. Specifically, we aimed to detect between-country differences in the intercorrelation. The comparisons were carried out pairwise using the MODEL CONSTRAINT command in Mplus. To calculate power, we simulated a multi-group analysis with two countries, and set to detect a difference in correlation of 0.2—a small to medium effect size [37]. Monte Carlo simulations revealed that 675 participants per country would be needed to achieve 95% power.

In sum, to achieve 95% power for all three analyses, a total of 675 participants each across the US, Canada and the UK (total sample 2025 participants) was required. For data generation code, simulated data and analysis code, see the supplementary material (https://osf.io/k4jy9/).

## 2.8. Preprocessing of data

### 2.8.1. Exclusion criteria and outlier identification

Due to the nature of the online sampling, we excluded participants who were inattentive. Two types of inattentiveness are commonly observed, one being general inattentiveness and the other being marked by frequently selecting the same answer for entire blocks and completing a survey in short periods of time [38]. As preregistered, we used several criteria to exclude inattentive participants [39]. First, we removed respondents with multiple submissions using an IP check [40]. Second, we included a seriousness check [41]. Data for participants indicating they were not serious were discarded. Third, three instructed response items adapted from Barends and de Vries [42] were asked on different steps of the survey. For a list of the three items, see supplementary material (https://osf.io/qpdnt/). Data for participants who respond incorrectly to more than one of these three questions were discarded. Fourth, participants with more than 25% missing data from selecting 'prefer not to answer' were discarded. Fifth, participants whose survey completion time was less than half of the 5% trimmed mean of the complete sample of participants were flagged as potentially not paying attention, and responses were examined for inconsistencies [43].

Two additional exclusion criteria were added in light of the results in the preliminary measurement study (as part of stage 1 manuscript revisions). First, in an examination of participants' responses for the number line task, we found some participants answered the number line questions by placing their answers on the same point on the number line irrespective of the question prompt. To remove these non-responsive participants, we calculated the median absolute deviation of each participants' responses and excluded participants if the median absolute deviation of their responses was below 0.1. Finally, participants who had received a COVID-19 vaccine or had indicated that they participated in a COVID-19 trial were excluded.

Univariate outliers were detected using the absolute deviation around the median, where continuous values ±3 median absolute deviation (MAD) were considered outliers [44]. Multivariate outliers were detected using Mahalanobis distance test with a $p < 0.001$ [45]. Most questions in the survey required rating-scale responses that had well-defined ranges, and the removal of outliers for these questions may have underestimated actual variability in the population. Therefore, for rating-scale questions, identified outliers were examined for obvious errors or participant inattention. When no evidence for errors or inattention was found, the value was retained in the final dataset. For questions without a well-defined range (i.e. the number line questions in the basic numeracy section), identified outliers were examined for errors. If no errors were found, the outliers were treated as missing data. Finally, for the set of questions in the number line task, we calculated the median absolute deviation of each participant's responses and excluded participants if the median absolute deviation of their responses was below 0.1 (please see the electronic supplementary material, appendix A for more details).

A total of 2124 participants (710 from US, 703 from Canada and 711 from the UK) were recruited for the current study. In total, 93 participants were removed from the reported data analyses based on the outlier exclusion criteria. Specifically, two participants were removed based on the IP address criterion, one was removed based on the seriousness criterion, four were removed due to the missing data criterion, 27 were removed based on the median absolute deviation criterion, 54 were removed based on the vaccination criterion and two were removed due to the Mahalanobis distance criterion. The final sample consisted of 680 participants from the US, 679 participants from Canada and 675 from the UK.

### 2.8.2. Missing data

As preregistered, we conducted a missing values analysis to identify patterns of missing data in the items. We anticipated two patterns that often occur in online research. The first consists of missing sporadic responses throughout the questionnaire, due either to inattention, or to participants not wanting to answer specific items. We analysed frequency of non-response for each item and identified items with higher non-response frequency. These items are not missing at random, and we determined whether they could be explained by covariates included in the current study, at which point, we included the covariates either in the model or as auxiliary variables that can be used to improve the estimation of missing data in full information maximum-likelihood (FIML).

Total percentage missingness was calculated for each participant. Generally, missingness was low ($M = 0.2\%$, s.d. = 0.95). Total percentage missingness was regressed onto demographic and other

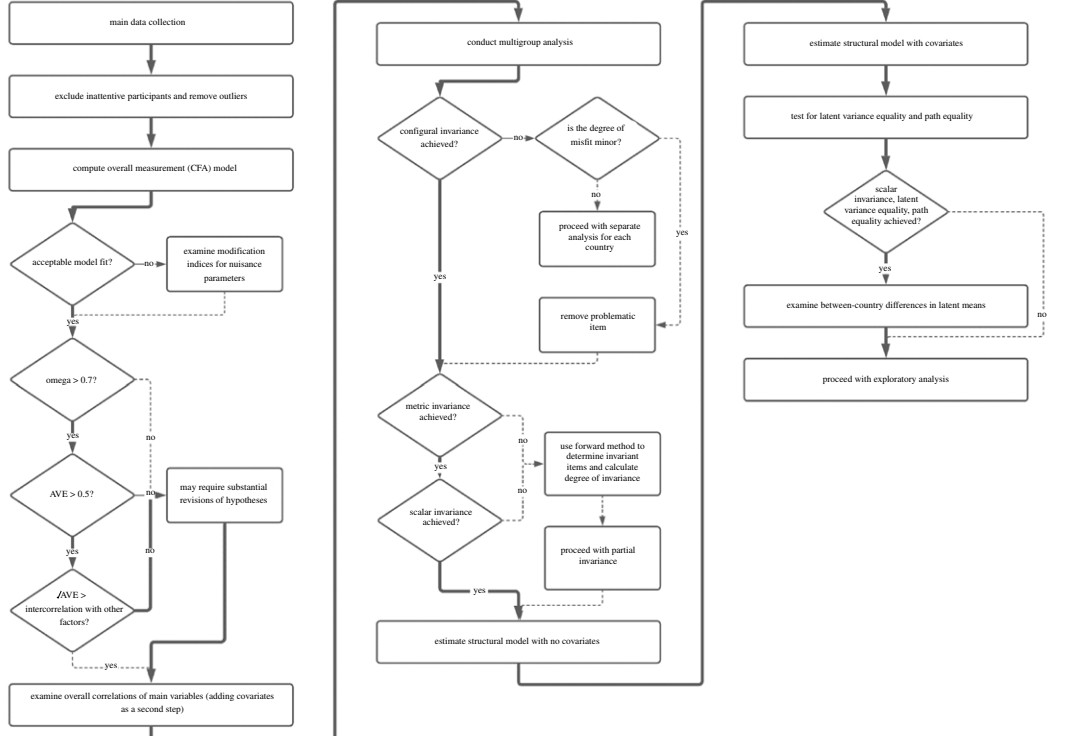

**Figure 2.** Decision tree of the main analysis.

covariates to ascertain whether missingness could be explained by demographic factors or other covariates. Results indicated that gender, frequency of obtaining information of COVID-19 via news sources, and self-proclaimed informedness about COVID-19 significantly predicted percentage missingness. As such, these covariates were added as auxiliary variables used to improve the estimation of missing data.

The second pattern of missingness that we anticipated was attrition throughout the questionnaire. Respondents who dropped off at different points along the questionnaire may differ from those who completed the survey in three potentially different ways (i.e. less motivation, having restricted amounts of uninterrupted time available to complete the survey, lower overall math skills). We analysed this particular pattern by testing whether missingness at the later points was related to performance on earlier items. Regardless of the in-depth missingness analyses, FIML [46] is still the best approach because no cases needed to be left out except when careless responding was identified.

To assess attrition, per cent missingness was calculated for each of the three main sections of the main study and each were regressed on performance in the basic numeracy and COVID-19 health numeracy tasks. Results indicated that per cent missingness was significantly related with performance in both basic numeracy and COVID-19 health numeracy. As such, missing data were processed using FIML [46].

## 2.9. Analysis

The analysis plan included a preliminary measurement study, tests of reliability and validity, main data analyses and exploratory analyses. Please figure 2 for a flowchart of the analysis plan.

### 2.9.1. Reliability and validity

Subsequent to estimating the confirmatory factor analysis (CFA) in the main study (see below), reliability—the degree to which indicators of the latent factor consistently measured the same underlying construct—was assessed using coefficient omega for each latent variable estimated [47]. We used the typical cut-off point for acceptable reliability of 0.7 for each of the three latent variables [47,48]. Once reliability was established, convergent validity—the degree to which individual items were strongly related to the hypothesized factor—was assessed using average variance extracted (AVE). We used the typical cut-off score for acceptable convergent validity, where AVE for each of the

three factors should be higher than 0.5. Finally, discriminant validity—the degree to which individual items were not related to other factors—was assessed using the Fornell & Larcker criterion [49,50]. Specifically, we adopted the convention that the square root of the AVE for each construct should be larger than the correlation of the specific construct with any of the other constructs.

### 2.9.2. Item parcels

The current study employed a large number of items in the measurement of the main variables. Operationalizing latent variables with a large number of indicator items is not ideal from a psychometric perspective. This is because individual items tend to be less statistically reliable than aggregates and are more likely to have correlated specific effects that may subject the final model to nuisance parameters (i.e. correlated residuals; [51]). To address this, we employed an item parcel approach to form a smaller set of composite indicator items to inform the latent variables.

Given that the current survey employed multiple question formats (e.g. multiple-choice, fill-in-the-blank and number line), a random parcelling approach is not ideal, as random parcelling assumes that all items are interchangeable. Results from the preliminary measurement study showed that each of the hypothesized latent variables were unidimensional. As such, parcels were assigned using a balancing approach.

To reduce the risk of overfitting, the same items composition from the preliminary measurement study was used to construct the parcels in the main study (see electronic supplementary material, table S3 for individual items in each of the parcels).

### 2.9.3. Main analyses

After ensuring data quality by implementing our participant and outlier exclusions, we pursued the research questions using a structural equation modelling approach. Basic numeracy, COVID-19 health numeracy and COVID-19 health-related attitudes and behaviours were operationalized as latent variables each with three parcels as indicators. Parcels were treated as continuous variables. All analyses were carried out using Mplus 8.3 using the maximum-likelihood estimator with robust standard errors [46]. Where applicable, goodness of fit of the models was assessed with the $\chi^2$ test statistic, the comparative fit index (CFI), the Tucker–Lewis index (TLI) and the root mean square error of approximation (RMSEA). Typical cut-off scores for excellent and adequate fit are CFI and TLI > 0.95 and > 0.90, and RMSEA < 0.06 and < 0.08, respectively [52]. In the case of inadequate fit (i.e. CFI and TLI < 0.9, RMSEA > 0.08), modification indices were consulted for correlated residuals and other sources of misfit.

*Simple correlation model*. We first examined the associations between the three main variables in a simple correlation model. As a first step, a model with only the three main (latent) variables was entered. As a second step, the robustness of the observed correlations was tested by entering covariates as predictors of the three main variables.

*Invariance testing*. The main objective of this research was to investigate differences across countries in the means and variation of the main variables in the model (basic numeracy, COVID-19 health numeracy and COVID-19 health-related attitudes and behaviours). This included multiple-group measurement invariance analyses of the individual latent measures. This procedure is usually performed as a step to establish factorial validity across groups. Once established, it becomes possible to evaluate differences in latent means, variances, covariances or specific correlations across different countries.

We used the standard procedures in the confirmatory factor analytic literature for evaluating measurement invariance across countries [48]. In short, this consisted of a number of steps testing the invariance of different parts of the factor/latent variable structure. These included (i) configural invariance—the overall factor structure (i.e. the number of factors and their indicator variables), (ii) metric invariance—the loadings of the indicator variables, and (iii) scalar invariance—the intercepts of the indicator variables [36]. Strict measurement invariance is not required for the following analyses and will not be tested. These are tested in incremental stages, and the first three stages are essential to proceed with tests of latent means. Configural invariance was assessed using model fit, where CFI and TLI > 0.90 and RMSEA < 0.08 is generally accepted as adequate model fit [52]. Metric and scalar invariance are evaluated using the chi-square difference test and the degradation of fit in CFI and RMSEA. Specifically, we used the criterion of $\Delta$RMSEA < 0.015 and $\Delta$CFI < −0.01 [53,54].

In the case where measurement invariance is not achieved, we proceeded with partial-invariance tests and identified the source of non-invariance. Some degree of researcher judgement is required to gauge the severity of the invariance between countries. Had we not met configural invariance, we planned to examine modification indices for the source of the misfit. If the degree of misfit were small, the removal of the problematic indicator item was considered a potential solution. This would necessitate the examination of between-country differences in the individual items constituting the parcel to find problematic items. If the degree of invariance is high, it would be necessary to estimate separate models for each country. In the event that we did not meet metric or scalar invariance in a given country, we planned to identify invariant items using the forward method [55]. Effect size of invariance ($d_{MACS}$) would be calculated to gauge the magnitude of invariance [56]. Identified invariant items were allowed to vary across countries. We planned to proceed with partial scalar or metric invariance if less than half of the indicators are not invariant [57,58]. Lack of invariance in the factor structure is often seen as a nuisance, but it can also advance our understanding of cross-cultural differences in how concepts are conceptualized. For example, if results indicated that the sole source of non-invariance in the latent variable COVID-19 health-related attitudes and behaviours is the factor loading of mask-wearing frequency, it could be interpreted that the amount of variance of mask-wearing frequency that COVID-19 health-related attitudes and behaviours can explain is different between countries.

Finally, in the case where scalar invariance, latent variance equality and variance-covariance equality were achieved [58], potential latent mean differences between countries were examined.

*Multi-group analysis.* Once metric or scalar invariance or partial metric or scalar invariance was achieved, between-country differences in latent correlations were examined in two steps. As a first step, we formulated a model with only the three main variables and examined the intercorrelations. As a second step, the robustness of the observed intercorrelations was tested by entering covariates as predictors of the three main variables. Between-country differences in the correlations were examined pairwise using the MODEL CONSTRAINT command in Mplus.

### 2.9.4. Exploratory analyses

Multiple exploratory analyses and variables were preregistered in light of the results from the preliminary measurement study. Please see the electronic supplementary material, appendix B for a full list.

# 3. Results

## 3.1. Descriptive statistics

To begin, descriptive statistics were computed. Descriptive statistics of the item parcels and covariates are presented in table 1.

## 3.2. Confirmatory factor analysis

As preregistered, a confirmatory factor analysis was next estimated using the same parcelling method (i.e. same strategy and contents of the parcels) as in the preliminary measurement study. Reliability as measured by coefficient omega for basic numeracy, COVID-19 health numeracy and COVID-19 attitudes and behaviours were 0.848, 0.763 and 0.945, respectively. As the factor loadings were high and the coefficient omega for each latent variable was above 0.7, this suggests that the latent variables had high internal consistency. AVEs for basic numeracy, COVID-19 health numeracy and COVID-19 attitudes and behaviours were 0.653, 0.522 and 0.851, respectively. As the AVEs of the three latent variables were above 0.5, this suggests that convergent validity was established for all three latent variables. Finally, $\sqrt{AVE}$ for COVID-19 attitudes and behaviours were larger than the correlation of the specific construct with any other construct. This suggests that discriminant validity was established for COVID-19 attitudes and behaviours. However, the $\sqrt{AVE}$ for COVID-19 health numeracy (0.72) was smaller than its correlation with basic numeracy (0.86). Similarly, $\sqrt{AVE}$ for basic numeracy (0.81) was smaller than its correlation with COVID-19 health numeracy (0.86). This suggests that COVID-19 health numeracy and basic numeracy cannot be reliability discriminated from one another.

**Table 1.** Descriptive statistics for all item parcels and covariates. Note: item membership in each parcel is reported in electronic supplementary material, table S4. See electronic supplementary material, table S5 for descriptive statistics stratified by country.

| continuous variables | mean | variance | skew | kurtosis |
|---|---|---|---|---|
| basic numeracy parcel 1 | −0.034 | 0.115 | −0.354 | −0.198 |
| basic numeracy parcel 2 | −0.062 | 0.126 | −0.08 | −0.634 |
| basic numeracy parcel 3 | 0.088 | 0.147 | −0.638 | −0.338 |
| COVID-19 health numeracy parcel 1 | 0.743 | 0.246 | −0.923 | 0.368 |
| COVID-19 health numeracy parcel 2 | 0.657 | 0.280 | −0.489 | −0.641 |
| COVID-19 health numeracy parcel 3 | 0.548 | 0.261 | 0.079 | −0.862 |
| attitudes and behaviours parcel 1 | 8.790 | 1.402 | −2.209 | 6.254 |
| attitudes and behaviours parcel 2 | 8.590 | 1.616 | −2.055 | 5.127 |
| attitudes and behaviours parcel 3 | 8.810 | 1.471 | −2.095 | 5.578 |
| years of education | 14.872 | 13.547 | −0.376 | 1.504 |
| age | 59.765 | 208.581 | −0.696 | −0.287 |
| socio-economic status | 6.188 | 3.327 | −0.417 | 0.117 |
| people in household | 2.214 | 1.437 | 3.984 | 54.804 |
| general anxiety | 3.699 | 5.841 | 0.809 | −0.290 |
| COVID-19 anxiety | 5.916 | 6.884 | −0.246 | −0.968 |
| math anxiety | 3.807 | 6.922 | 0.716 | −0.599 |
| political affiliation (higher right leaning) | 0.519 | 0.070 | −0.065 | −0.596 |
| frequency of obtaining information of COVID-19 via news sources | 4.394 | 1.144 | −1.979 | 3.199 |
| frequency of obtaining information of COVID-19 via social sources | 3.153 | 2.419 | −0.276 | −1.430 |
| self-proclaimed informedness of COVID-19 | 3.419 | 0.364 | −0.673 | 0.352 |

(Continued.)

**Table 1.** (*Continued.*)

| continuous variables | mean | variance | skew | kurtosis |
|---|---|---|---|---|
| *categorical variables* | | | | |
| gender | 46% female | | | |
| | 54% male | | | |
| | <1% non-binary | | | |
| income change since the pandemic | 73% unchanged | | | |
| | 23% decreased | | | |
| | 4% increased | | | |
| employment status before the pandemic | 15% unemployed | | | |
| | 40% unemployed by choice (retired/homemaker/student) | | | |
| | 44% employed | | | |
| employment status change since the pandemic | 15% employment decreased | | | |
| | 84% unchanged employment | | | |
| | 1% employment increased | | | |
| community type | 20% rural | | | |
| | 51% suburban | | | |
| | 29% urban | | | |
| minority group | 88% non-minority | | | |
| | 11% minority | | | |

**Table 2.** Zero-order correlations between the main variables.

|  | 1 | 2 | 3 |
|---|---|---|---|
| 1. basic numeracy | — | 0.862*** | −0.039 |
| 2. COVID-19 health numeracy |  | — | 0.029 |
| 3. COVID-19 attitudes and behaviours |  |  | — |

***p < 0.001, **p < 0.01, *p < 0.05.

**Table 3.** Partial correlations between the residualized main variables.

|  | 1 | 2 | 33 |
|---|---|---|---|
| 1. basic numeracy | — | 0.834*** | −0.038 |
| 2. COVID-19 health numeracy |  | — | 0.021 |
| 3. COVID-19 attitudes and behaviours |  |  | — |

***p < 0.001, **p < 0.01, *p < 0.05.

Given the results above suggest that COVID-19 health numeracy and basic numeracy may not be distinguishable, we next explored whether a two-factor model—where COVID-19 health numeracy and basic numeracy were combined into a single general numeracy factor—better described the data. To this end, the preregistered three-factor model was compared with an alternative two-factor model where the six indicator variables for basic numeracy and COVID-19 health numeracy were instead loaded onto one general numeracy factor. Results indicated that the two-factor model had significantly worse model fit, $\chi^2 = 184.11$, $p < 0.001$, $\Delta$CFI = 0.016, $\Delta$RMSEA = 0.033. Accordingly, the decision was made to continue analysis with the preregistered three-factor model with the caveat that COVID-19 health numeracy and basic numeracy as operationalized in the current study may not be highly distinguishable. The same analyses were performed on the alternative two-factor model and reported in the electronic supplementary material, appendix C.

## 3.3. Baseline correlation model

As preregistered, a baseline model—in which it was assumed that there were no between-country differences in the relations between the three variables—was estimated. The relations between the variables were represented by zero-order correlations. For the baseline three-factor model, model fit was good, ($\chi^2_{24} = 84.566$, $p < 0.001$, CFI = 0.995, TLI = 0.992, RMSEA = 0.035). The zero-order correlations between the main variables of interest are presented in table 2.

## 3.4. Baseline correlation model with covariates

As preregistered, to test the robustness of the correlations between the three main variables of interest, we estimated a baseline correlation model in which each of the main variables were residualized by all covariates listed in table 1. Model fit was good, ($\chi^2_{150} = 402.199$, $p < 0.001$, CFI = 0.980, TLI = 0.971, RMSEA = 0.029). The partial correlations between the main variables of interest are presented in table 3.

## 3.5. Invariance testing

As preregistered, the main variables are operationalized as latent variables, we first conducted invariance testing to establish factorial validity across countries (table 4).

Configural invariance was assessed through model fit. Results indicated that model fit was good, ($\chi^2_{72} = 147.899$, $p < 0.001$, CFI = 0.993, TLI = 0.990, RMSEA = 0.039). This suggests that configural invariance was achieved.

Metric invariance was assessed through the chi-square difference test and change in model fit when compared with the configural invariant model. Model fit for the metric invariant model was good, ($\chi^2_{984} = 176.361$, $p < 0.001$, CFI = 0.992, TLI = 0.990, RMSEA = 0.040). Comparison between models

**Table 4.** Tests of measurement and structural invariance. Note: CFI = comparative fit index; TLI = Tucker–Lewis index; RMSEA = root mean square error of approximation; SRMR = standardized root mean square residual.

| model | $\chi^2$ | d.f. | CFI | TLI | RMSEA | SRMR | ΔCFI | ΔRMSEA |
|---|---|---|---|---|---|---|---|---|
| 1. configural | 147.899 | 72 | 0.993 | 0.990 | 0.039 | 0.026 | — | — |
| 2. metric | 176.361 | 84 | 0.992 | 0.990 | 0.040 | 0.036 | 0.001 | 0.001 |
| 3. scalar | 293.796 | 96 | 0.983 | 0.981 | 0.055 | 0.047 | 0.009 | 0.015 |
| 4. strict | 358.758 | 114 | 0.979 | 0.980 | 0.056 | 0.065 | 0.004 | 0.001 |
| 5. variance-covariance equality | 367.285 | 120 | 0.978 | 0.981 | 0.055 | 0.070 | 0.001 | 0.001 |

revealed a significant deterioration of model fit via chi-square difference test, ($\chi^2_{12} = 28.46$, $p < 0.047$), but no evidence for deterioration of model fit in CFI and RMSEA, (ΔCFI = 0.001, ΔRMSEA = 0.001). Given the sample size of the current study and that the chi-square test tends to be oversensitive with large sample sizes, it is likely that the significant chi-square difference test reflected minor model misfit. In sum, results indicated that metric invariance was achieved.

Scalar invariance was assessed through the chi-square difference test and change in model fit when compared with the metric invariant model. Model fit for the Scalar invariant model was good, ($\chi^2_{94} = 293.796$, $p < 0.001$, CFI = 0.983, TLI = 0.981, RMSEA = 0.055). Comparison between models revealed a significant deterioration of model fit via chi-square difference test, ($\chi^2_{12} = 117.435$, $p < 0.001$), but no evidence for deterioration of model fit in CFI and RMSEA, (ΔCFI = 0.009, ΔRMSEA = 0.015). As such, results indicated that scalar invariance was achieved.

## 3.6. Multi-group analysis

As preregistered, having established scalar invariance, we next examined between-country differences in the relations between the main variables. Model fit was good, ($\chi^2_{94} = 293.796$, $p < 0.001$, CFI = 0.983, TLI = 0.981, RMSEA = 0.055). Results indicated basic numeracy and COVID-19 health numeracy were significantly correlated for participants in Canada ($r = 0.839$, $p < 0.001$), the US ($r = 0.882$, $p < 0.001$) and the UK ($r = 0.876$, $p < 0.001$), suggesting that in all three countries, participants who had better basic numeracy skills also had better COVID-19-related health numeracy skills. However, basic numeracy and COVID-19 attitudes and behaviours were not correlated for participants in Canada ($r = -0.060$, $p = 0.165$), the US ($r = -0.085$, $p = 0.237$) or the UK ($r = 0.032$, $p = 0.454$). Finally, COVID-19 health numeracy and COVID-19 attitudes and behaviours were not correlated for participants in Canada ($r = 0.064$, $p = 0.151$), the US ($r = -0.053$, $p = 0.237$) or the UK ($r = 0.082$, $p = 0.071$).

Covariates were next added to the model to test the robustness of the correlations. Model fit was good, ($\chi^2_{474} = 770.762$, $p < 0.001$, CFI = 0.969, TLI = 0.957, RMSEA = 0.035). Results indicated basic numeracy and COVID-19 health numeracy were significantly correlated for participants in Canada ($r = 0.819$, $p < 0.001$), the US ($r = 0.863$, $p < 0.001$) and the UK ($r = 0.859$, $p < 0.001$). Basic numeracy and COVID-19 attitudes and behaviours were not correlated for participants in Canada ($r = -0.081$, $p = 0.137$), the UK ($r = -0.035$, $p = 0.516$), but was negatively correlated for the US ($r = -0.129$, $p = 0.008$).[1] Finally, COVID-19 health numeracy and COVID-19 attitudes and behaviours were not correlated for participants in Canada ($r = -0.016$, $p = 0.787$), the US ($r = -0.067$, $p = 0.198$) or the UK ($r = 0.004$, $p = 0.949$).

As the correlations between COVID-19 health numeracy and COVID-19 attitudes and behaviours were not statistically significant in any of the countries, pairwise comparisons of between-country differences in these correlations were not conducted. Similarly, the correlation between basic numeracy and COVID-19 attitudes and behaviours was only statistically significant in the US, therefore pairwise comparisons were not conducted. Finally, pairwise comparison of the relation between basic numeracy and COVID-19 health numeracy revealed that none of the correlations differed between countries (all $ps > 0.20$).

[1]The relation between basic numeracy and COVID-19 attitudes and behaviours was statistically insignificant when only examining the three main variables but was significant in the US when covariates are included. This suggests that one or more of the covariates are acting as a negative confounder. However, a *post hoc* analysis of negative confounders not based on theory would not be prudent, as the identification of one or multiple confounders would entail a large number of comparisons.

## 3.7. Latent mean differences

As preregistered, having established that there were no between-country differences in the relations between the main variables, we next explored whether there were latent mean differences between countries. To this end, we first tested whether strict invariance and variance-covariance equality could be established. Test for latent strict invariance and variance-covariance equality was assessed through the chi-square difference test and change in model fit when compared with the scalar invariant model. Model fit for the strict invariance model was good, ($\chi^2_{114} = 386.041$, $p < 0.001$, CFI = 0.979, TLI = 0.980, RMSEA = 0.056). Comparison between models revealed a significant chi-square difference test, ($\chi^2_{18} = 64.962$, $p < 0.001$), but no significant change in CFI and RMSEA, (ΔCFI = 0.004, ΔRMSEA = 0.001). As such, results indicated that strict invariance was achieved.

Model fit for the variance-covariance equality model was good, ($\chi^2_{120} = 367.285$, $p < 0.001$, CFI = 0.978, TLI = 0.981, RMSEA = 0.055). Comparison between models revealed an insignificant chi-square difference test, ($\chi^2_6 = 8.527$, $p = 0.202$) and no significant change in CFI and RMSEA, (ΔCFI = 0.001, ΔRMSEA = 0.001). As such, results indicated that variance-covariance equality was achieved.

As strict invariance and variance-covariance equality was established, we proceeded with comparisons of latent means across countries. First, we compared a model where latent means are constrained to be equal versus a model where latent means are free to vary. Results suggest that model fit does not significantly deteriorate when latent means are constrained to be equal ($\chi^2_3 = 8.425$, $p = 0.037$, ΔCFI < 0.001, ΔRMSEA = 0.001), suggesting that there are no latent mean differences between countries.

## 3.8. Exploratory analysis

### 3.8.1. Sub-dimensions of basic numeracy

The current study used parcels in order to ascertain the relations between the three main variables. However, the relations between sub-dimensions of basic numeracy and the main variables COVID-19 health numeracy and COVID-19 attitudes and behaviours were also of interest. To examine this, composite (mean) scores for the sub-dimensions of the number line task (whole numbers, fractions, percentages, large numbers and non-symbolic numbers) were computed and were correlated with the main variables. Results are presented in table 5. Zero-order correlations underscore the findings of the primary analysis and add some additional nuanced descriptions. However, caution is warranted in their interpretation given the lack of covariate presented here. First, all number line estimation types show a moderate correlation with COVID-19 health numeracy ($r = 0.419$ to $r = 0.538$), even the non-symbolic NLE ($r = 0.425$), which is expected to be the most distal numerical estimation task to our measure of COVID-19 health numeracy. The strongest correlation between a sub-dimension of basic numeracy and COVID-19 health numeracy was our section of basic numeracy containing word problems ($r = 0.723$), probably because the measures shared method variance (i.e. COVID-19 health numeracy was also tested with word problems).

## 4. Discussion

After conducting a measurement pilot with 525 adults, the current study surveyed 2032 adult participants across the UK, the US and Canada during the month of December 2020 in the first year of the COVID-19 pandemic. We aimed to understand the relations between (i) basic numeracy, (ii) COVID-19 health numeracy, and (iii) COVID-19 health-related attitudes and behaviours. Importantly, the current study differs from most studies of the influence of numeracy on health-related attitudes and behaviours by the inclusion of a wide range of basic numeracy items, including symbolic and non-symbolic number line judgements (i.e. placing whole numbers, fractions, percentages, large numbers and visual representations of proportions on a number line) alongside mathematical word problems more traditionally related to health numeracy.

Our measure of COVID-19 health-related attitudes and behaviours was designed to capture the extent to which people followed public health advice and understood COVID-19 to be a global health threat. Based on previous findings relating higher numeracy to better health literacy (e.g. [23]) and better decision-making generally (e.g. [59]), we expected to observe a positive relation among our three constructs.

**Table 5.** Zero-order correlations between components of basic numeracy and the main variables. Note: upper triangle consists of zero-order correlations. NLE represents number line estimation task.

|  | 1 | 2 | 3 | 4 | 5 | 6 | 7 | 8 |
|---|---|---|---|---|---|---|---|---|
| 1. COVID-19 health numeracy | — | 0.032 | 0.510*** | 0.473*** | 0.419*** | 0.538*** | 0.425*** | 0.723*** |
| 2. COVID-19 attitudes and behaviours |  | — | −0.005 | −0.023 | −0.009 | −0.042 | 0.015 | −0.034 |
| 3. NLE whole numbers |  |  | — | 0.251*** | 0.370*** | 0.410*** | 0.252*** | 0.465*** |
| 4. NLE fractions |  |  |  | — | 0.264*** | 0.343*** | 0.285*** | 0.483*** |
| 5. NLE percentages |  |  |  |  | — | 0.302*** | 0.253*** | 0.384*** |
| 6. NLE large numbers |  |  |  |  |  | — | 0.291*** | 0.474*** |
| 7. NLE non-symbolic numbers |  |  |  |  |  |  | — | 0.324*** |
| 8. word problems |  |  |  |  |  |  |  | — |

***$p < 0.001$, **$p < 0.01$, *$p < 0.05$.

In line with our predictions, basic numeracy was positively related to health numeracy. However, contrary to our predictions, measures of both basic numeracy and COVID-19 health numeracy were not related to COVID-19 health-related attitudes and behaviours in all three countries. In other words, participants' numeracy skills were not related to their adherence to public health advice. Thus far, studies relating numeracy to personal behaviours surrounding COVID-19 have been mixed. For example, a recent study by Roozenbeek et al. [60] reported that higher numeracy skills (mostly tapping participants' understanding of percentages and probabilities) made participants less susceptible to misinformation surrounding COVID-19. Numeracy also had a small and positive relation to compliance with health guidance, though not within all countries surveyed. However, Roozenbeek et al. [60] also reported that participants' numeracy did not relate to their likelihood of getting vaccinated when it was available. In another study, Thompson et al. [2,3], implemented an online health numeracy intervention and reported that adopting protective health behaviour was not influenced by an improvement in health numeracy even though protective behaviours and measures of numeracy were correlated, both findings that are convergent with the current study. Given these findings and those of the current study, there does not appear to be a robust and strong relation between numeracy and health-related attitudes and behaviours pertaining to COVID-19.

A number of factors can help explain the lack of association between basic and health numeracy and COVID-19 health-related attitudes and behaviour. It is possible that there was insufficient variability in the attitudes and behaviour measure due to a ceiling effect, which made it difficult to find associations between this measure and the numeracy measures. Data collection for the current study began in December 2020. By that time, public health advice had been established for many months and adherence may have been less dependent on people's estimation of the severity of the situation and more dependent on established local norms. Furthermore, the countries in which participants were recruited for the Roozenbeek et al. [60] study differed from ours (i.e. UK, US, Ireland, Spain and Mexico versus UK, US and Canada). Public health advice varied across countries and restrictions were implemented at different times, which could help explain the presence or absence of certain associations with behaviour and attitudes.

The lack of an association between numeracy and COVID-19 health-related attitudes and behaviour may also parallel a trend that has recently been observed in attitudes about climate change. In a study of numeracy, science comprehension and climate change risk perception, Kahan et al. [61] compared two competing hypotheses regarding risk perception. The science comprehension thesis suggests that risk perception is guided by one's understanding of the complex scientific concepts underlying climate change—a better grasp of the science results in a better understanding of the global threat. By contrast, the cultural cognition thesis suggests that individuals form perceptions of societal risks that cohere with like-minded cultural groups. People adapt the data they receive to fit culturally established interpretations. In this study, Kahan et al. [61] observed small, negative correlations between both science-literacy and numeracy and perceived climate change risk. In other words, better knowledge of

the science and the numerical concepts of risk did not enhance risk perception. Instead, the relation was significant in the opposite direction, supporting the idea that it was cultural cognition, rather than scientific comprehension, that drove perceived climate change risk. There is some data already to suggest that political worldview, such as prosocial tendencies and individualistic values [62] or even political party affiliation [63] are strongly related to COVID-19 attitudes and behaviours. Therefore, as the cultural cognition thesis suggests, high numeracy and scientific dissemination about the risks and spread of disease alone may not be enough to influence a population's health-related attitudes and behaviours. Directly addressing cultural values that influence cultural cognition may be a more effective strategy for influencing health behaviours in combination with improving health numeracy. Similar strategies have been suggested for influencing perspectives and behaviours related to climate change where public outreach would target both science literacy and cultural factors [64]. More work on this line of enquiry, including further analysis of the current data, is needed to substantiate this speculation.

Another major aim of the current study was to investigate the relation between basic numeracy (unrelated to medical or health-related information) and health numeracy. Before conducting analyses on the structural equation model, we evaluated the measurement model by conducting a confirmatory factor analysis of the three latent variables. Results suggest that all three variables are reliable and show convergent validity. However, in contrast to the results yielded in the preliminary measurement study, we failed to show divergent validity between basic numeracy and COVID-19 health numeracy. This suggests that these two latent variables may be measuring the same, or at least a largely overlapping, underlying construct. Despite this lack of divergent validity, model fit was significantly worse when the two numeracy constructs were grouped into one latent variable. This suggests that the basic numeracy and COVID-19 health-related numeracy may theoretically be separable constructs. In particular, the number line items show lower bi-directional correlations with the health numeracy construct (ranging from $r = 0.419$ to $0.538$) than the word problems ($r = 0.723$), showing some aspects of the basic numeracy construct are more independent than others. Still, most items measuring health numeracy were more similar in format to the word problem portion of the basic numeracy measure, so question formatting may contribute to the higher correlation in addition to the content domain of the items. Future studies that employ more sensitive designs may be able to reliably distinguish the two constructs.

These results may help guide future studies investigating the role of numeracy in health-related decisions and behaviours. Since measures of basic numeracy and topic-specific measures of numeracy are highly correlated, researchers interested in relations between numeracy and non-numeracy-related outcome measures, such as decision making or compliance, may not need to incorporate separate measures for basic numeracy and topic-specific numeracy.

In addition to the limitations already mentioned, other factors should be considered when interpreting the current study's results. People's attitudes and behaviours have continued to shift based on the current status of the pandemic, the constant updating of local public health guidelines, and scientific advances in understanding and combating the disease. The current study survey was conducted as the initial stages of vaccine roll-out were underway in all three countries, though none of the participants in the current study had yet been vaccinated. Adherence to public health advice, and public health advice itself, may continue to increase in variability as the COVID-19 pandemic continues, which may influence the relations explored in the current study.

Finally, while our study has used existing measures of basic numeracy, measurements of COVID-19 health-related numeracy and COVID-19 attitudes and behaviours have been adapted from related sources. As such, it may be the case that the measurement of COVID-19 health-related numeracy and COVID-19 attitudes and behaviours may not match the intended underlying concepts. However, this risk may be reduced as the questions, at face value, do clearly communicate to readers the intent of the survey. Nevertheless, future studies would be required to assess construct validity, such as examining our measures with other established measures of numeracy and health numeracy.

Overall, the current study suggests that basic numeracy is highly related to one's understanding of data about COVID-19. However, one's level of understanding of the numerical information about the disease does not predict their attitudes about its severity or their likelihood of following public health advice. Therefore, while numeracy appears to be a necessary part of understanding data about COVID-19, numeracy alone may not be sufficient for predicting public opinion or adoption of recommended health practices.

Ethics. The study was approved by the non-medical research ethics board at the University of Western Ontario and was conducted according to their guidelines. Participants were presented with a letter of information and implied consent was acquired before starting the survey. Qualtrics panelists join from a variety of sources. They may be airline

customers who chose to join in reward for SkyMiles, retail customers who opted in to get points at their favourite retail outlet, or general consumers who participate for cash or gift cards, etc. When participants are invited to take a survey, they are informed how they will be compensated. All procedures and methods were approved by the Non-medical Ethics Review Board of the University of Western Ontario. The approval number is REB# 115879.

Data accessibility. All the code and data can be found on the project's Open Science Framework (OSF) page: https://osf.io/qpdnt/.

Authors' contributions. N.T.T.L.: conceptualization, data curation, formal analysis, methodology, validation, visualization, writing—original draft, writing—review and editing; E.D.W.: conceptualization, data curation, methodology, project administration, resources, software, supervision, validation, visualization, writing—original draft, writing—review and editing; M.S.: conceptualization, data curation, formal analysis, investigation, methodology, project administration, writing—original draft, writing—review and editing; R.L.C.: conceptualization, methodology, project administration, resources, supervision, validation, writing—original draft, writing—review and editing; L.P.: conceptualization, methodology, project administration, resources, supervision, validation, writing—original draft, writing—review and editing; P.T.: data curation, formal analysis, methodology, writing—original draft; C.G.: conceptualization, methodology, project administration, resources, supervision, validation, writing—review and editing; I.S.A.: methodology, resources, validation, visualization, writing—review and editing; A.D.R.: conceptualization, formal analysis, methodology, project administration, writing—review and editing; C.T.: formal analysis, methodology, resources, validation, writing—review and editing; J.V.H.: methodology, resources, validation, writing—original draft, writing—review and editing; J.B.: conceptualization, methodology, resources, writing—original draft, writing—review and editing; A.A.: conceptualization, methodology, resources, validation, writing—review and editing; E.B.: methodology, resources, writing—original draft, writing—review and editing; I.C.: methodology, validation, writing—review and editing; F.O.: methodology, resources, writing—review and editing; D.A.: conceptualization, funding acquisition, methodology, project administration, supervision, validation, writing—original draft, writing—review and editing.

All authors gave final approval for publication and agreed to be held accountable for the work performed therein.

Competing interests. We declare we have no competing interests.

Funding. This research was supported by funding from an Advanced Research Fellowship from the Klaus J. Jacobs Foundation to D.A. at Western University; by the U.S. Department of Education Institute of Education Sciences Grants R305A160295 and R305U200004 to C.T. at Kent State University; NIH NICHD F32 Grant HD102106 to A.D.R.; BrainsCAN Postdoctoral Fellowship funded by the Canada First Research Excellence Fund (CFREF) to M.S.; CAPES (Doc-Pleno, 88881.128282/2016-01) to I.S.A.; Post-doctoral fellowship PDM/20/057 to E.B. and a Banting Postdoctoral Fellowship (NSERC) and BrainsCAN Postdoctoral Fellowship at Western University, funded by the Canada First Research Excellence Fund (CFREF) to E.D.W.

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
