## [Peer Review File · Royal Society Open Science]

Review History

RSOS-201303.R0 (Original submission)

Review form: Reviewer 1 (Sanjay Srivastava)

Do you have any ethical concerns with this paper?

No

Recommendation?

Accept with minor revision

Comments to the Author(s)

This is an interesting and timely proposal. In broad terms, the case for investigating the relationship between numeracy and COVID attitudes and behavior was compelling to me on

both scientific and practical grounds. The methods seemed broadly sound to me, and were described at a good level of detail and with clear language (which is hard to do with these kinds of analyses, so it is a big credit to the authors for that). Below I detail some comments and concerns which I hope will be helpful:

1. My largest concern is about the focus on mediation, which is a complex causal process. The introduction did not elaborate on the theoretical or conceptual case for investigating mediation. Furthermore, the design, a cross-sectional survey, is not an appropriate one for testing hypotheses about mediation (e.g., Bullock et al., 2010). I believe it would be more appropriate to reframe this research as descriptive, and it would still be compelling. To what extent is numeracy associated with COVID attitudes and behavior? And how much is that association about general numeracy versus domain-specific numeracy? I think this reframing would not be very far from the general research questions set up in the introduction, and the research design could remain unchanged. It would lead to some changes in the analytic strategy, but not a complete overhaul.

2. Figure 1 shows arrows from observed indicators to latent variables, suggesting they are formative. However, the text suggests that the latent variables will be reflective (and I do not believe a model with formative indicators would be identified). Assuming the text is right, the figure should probably be changed.

3. The statement that English “minimizes potential linguistic influences on numerical processing” does not appear to be supported by the cited reference, which discusses differences among languages but does not present English as having fewer influences. It makes sense that the researchers are interested in studying the countries they are from and the single, most commonly spoken language in them. But it is not necessary to argue that there is some scientific advantage to what seems to me to be more of a practical decision.

4. I reviewed the materials and had some questions and comments about a few of the items:

- a. COVID numeracy item 11 is: “In which country do you have a higher chance of being tested for COVID-19.” Doesn’t the answer depend on how each country decides who to test? It is possible that the researchers are thinking of random testing, or mean “you” generically, but I am not sure the respondents will too.
- b. COVID numeracy item 12: Would the answer depend on whether testing is done by random sampling vs. purposive (e.g., testing people with symptoms)?
- c. COVID numeracy item 13: Would the answer depends on whether confirmed cases are retested (e.g., to see if they have “cleared” the virus)?

5. As described, the “pilot” study is not a pilot study (Leon et al., 2011), and smaller-N studies are generally not a good basis for a power analysis (Kraemer et al., 2006). I would describe it more as a preliminary psychometric study. The authors should address whether the sample size will be adequate for the planned analyses. (I am not confident that it is, but that’s just a gut response and I may well be wrong.) I also did not understand the sentence, “Third, mean scores of items for each of the main variables will be used as a rough guide to estimate the intercorrelations of the main variables” on page 14.

6. I do not think outlier removal is necessary for rating-scale responses that have well-defined ranges. It may be better to use robust estimation and inference methods, which I believe are straightforward to implement in Mplus.

7. On p. 16 the authors write that there may be a problem with spurious correlations. The issue isn’t spuriousness – residual correlations may be a real and reliable feature of these measures. Regardless of the terminology, I think parceling is a wise choice here. I encourage the authors to look at Little et al (2013), which cautions against random parceling in many situations. They

review pros and cons of other strategies, including the strategy proposed here for subdimensions, as well as alternatives for dealing with subdimensions. (It may be that a domain-representative approach is better than the facet-representative one described here, so I'd encourage the authors to consider whether that's the case.)

8. What if the proposed model does not fit? It is very common that structural equation models do not fit well as planned, an issue the authors seem to be anticipating when they discuss residual correlations. What criteria (fit thresholds etc.) will be used to decide if there is good-enough fit to proceed with interpreting the substantive results? And if the fit is not adequate, and the authors proceed to make modifications in the same data they will conduct main analyses in, there is a risk of overfitting. It may be that the preliminary psychometric study could be increased in sample size and treated like a training dataset, with modifications recorded in an interim preregistration, and then the main study run as if it's a holdout dataset. I recognize that this would increase the cost and resources required and therefore is not a trivial suggestion. So I will add that some concessions to practicality may be appropriate here.

9. Will political leaning be analyzed across countries, or separately within each country? "Left" vs. "right" have different meanings, and are associated with different policy positions, in different countries.

10. Finally, a general comment. There are several places where the authors have left room for human judgment during the data analyses. For example they discuss running exploratory factor analyses to make decisions about parceling. And they will have to make decisions about whether measurement invariance is established, and when it is not, what to do about it. (The analyses plan here is underspecified.) I have preregistered SEMs before and I'm currently in the middle of a registered report of a measurement invariance study, which has given me a deep appreciation for how hard it is to preregister a complete decision tree in this kind of work. I think some room for researcher degrees of freedom is unavoidable in this kind of work and should not be an impediment to its acceptance as a registered report. A training-holdout plan as I described earlier might address some of these other issues as well. Alternatively, the authors may consider whether some kind of blinded analysis strategy (MacCoun & Perlmutter, 2017) could be implemented here to shield those decisions from any potential biasing influence on the substantive results.

Sanjay Srivastava
(This is a signed review.)

References

Bullock, J. G., Green, D. P., & Ha, S. E. (2010). Yes, but what's the mechanism?(don't expect an easy answer). *Journal of personality and social psychology*, 98(4), 550.

Kraemer, H. C., Mintz, J., Noda, A., Tinklenberg, J., & Yesavage, J. A. (2006). Caution regarding the use of pilot studies to guide power calculations for study proposals. *Archives of general psychiatry*, 63(5), 484-489.

Leon, A. C., Davis, L. L., & Kraemer, H. C. (2011). The role and interpretation of pilot studies in clinical research. *Journal of psychiatric research*, 45(5), 626-629.

Little, T. D., Rhemtulla, M., Gibson, K., & Schoemann, A. M. (2013). Why the items versus parcels controversy needn't be one. *Psychological methods*, 18(3), 285.

MacCoun, R. J., & Perlmutter, S. (2017). Blind analysis as a correction for confirmatory bias in physics and in psychology. *Psychological science under scrutiny: Recent challenges and proposed solutions*, 297-322.

Review form: Reviewer 2

Do you have any ethical concerns with this paper?

No

Recommendation?

Accept with minor revision

Comments to the Author(s)

The Authors are planning to conduct a large survey across three countries (UK, USA and Canada) to test the relation between basic numeracy, health numeracy (about the COVID-19 pandemic) and self-reported attitudes/behaviours in relation to health (with reference to the COVID-19 pandemic). Basic numeracy is measured with tasks assessing number magnitude processing (analogue number line tasks) and conceptual knowledge of fractions and decimals. Health numeracy is measured with a (very nice) series of multiple-choice questions on graph literacy, proportions and odd ratios. Health-related attitudes/behaviours are measured with a series of questions measuring compliance and the perception of impact and severity of the pandemic. A positive relation is expected between basic numeracy and health-related attitudes/behaviours (whereby higher numeracy originates more compliant attitudes/behaviours), mediated by health numeracy. Differences between countries will be analysed and potential moderating variables are included in the design.

1. This is a well-presented and thorough proposal that aims to address an interesting research question. However, in its current version it includes a plan for a pilot that could perhaps have been conducted before submission and there are a few unclear points. The most limiting aspect of the current design is that the mediation analysis may show a relation between the latent constructs without controlling for general ability.

2. Abstract.

It is clear and concise, and mostly focuses on the primary goal of the study - apart from the final statement. Why are political views and anxiety tested? This is not the strongest component of the study and the variables (one measured with a single item) are included for exploratory purposes. I think that only if robust evidence of moderation effects was found after data collection they could deserve a mention here.

3. Participants.

More information is needed about these prospective market research participants and whether Qualtrics panels are representative of the general population in the three countries of interest. Could they live in the same household and share the same information channels, behavioural changes etc.?

What compensation is offered?

4. Procedure and materials.

What information will be available about screen size and how will this be taken into account? At the end of the survey, participants are asked which type of device they are using but screen sizes may vary greatly even within the same type of device. Further, across devices there may be a systematic variation of response modality (mouse vs. touchpad vs. pen vs. finger).

5. Power analysis.

The Authors provide a power analysis for their mediation model, multigroup and moderation analyses and choose their sample size accordingly. What are the α -levels, and should correction for multiple testing be applied? Have missing data been taken into account? These may increase the sample size requirements by a rather large amount. In the mediation analysis, how likely is it that the size of the X & Y effect is the same as that of X & M and M & Y effects?

Given that several of the measures included in this study have been created ad hoc, planning a pilot study before conducting the larger study is very appropriate. I wonder, however, whether the pilot study should not have been conducted before submitting an IPA request for the larger study. Depending on the results of the pilot study, changes to the power analysis and to the study materials may be opportune (i.e. these would not be in definitive form at the time of IPA).

6. Number magnitude processing.

More than any other tests, the number line tasks and their derived PAE indices seem to depend heavily on hand-eye coordination/tool precision (e.g. pen, mouse, finger, touchpad), in addition to basic numeracy skills. Could individual scores in the number line tasks be confounded by response means? Inclusion of a few calibration items may be useful.

7. Demographics and other potential moderators.

Please define SES before using the acronym.

8. Ending questions.

Typo: Devince --& device

I am not convinced that somebody who has not been responding seriously that far will be reliable at the "seriousness check".

Review form: Reviewer 3 (Katherine Corker)

Do you have any ethical concerns with this paper?

No

Recommendation?

Major revision

Comments to the Author(s)

The current paper plans to test a mediational hypothesis linking basic numeracy with health attitudes and behavior through COVID-19 health numeracy. There are a number of strengths in the proposal, along with some important weaknesses. On the strength side, the team investigates an interesting question using a rigorous analytic plan, including extensive measurement invariance testing. I examined the materials on OSF, and things appear to be well organized (although there is not much there yet, beyond copies of the instruments). Key weaknesses and issues to address are outlined below:

1. Power analyses: The authors use Mplus to simulate a set of power analyses for their model. In the majority of analyses, the conclusion points to a sample size of 675 for each of their three planned countries. This is peculiar to see such consistency across several different analytic strategies. There are at least two possible explanations. First, the authors may have done a sensitivity analysis, working backward from sample size and desired power to effect size. There is nothing wrong with such an approach, but if this is the technique that was used, the paper should be explicit about it. The effect sizes vary across the different specifications while sample

size stays constant, so this may be what was done. On the other hand, this consistency could point to an error in the analyses (perhaps some kind of limit in Mplus or some computational limit). 675 per country does seem lower than I would have expected, but the authors do not plan to model any of their variables at the country level (nor could they, with only three nations), so this might explain the final number. I would have looked at the power simulation code, but it was not provided on OSF so I could not. I recommend sharing that code at some point. Regarding expectations for power in the multi-site context, the authors might consult Arend and Schäfer (2019, *Psych Methods*). The multi-level approach that they discuss is complementary to the multi-group SEM approach employed here.

2. There is a bit of a mismatch between what the authors say they want to do in their introduction, and what they can do with just three countries. For instance, they state that they will be “investigating whether country-level factors in these three countries influence the mediation model.” What they can do with their multi-group approach is look at whether model loadings, intercepts, variances/covariances (paths), and latent means differ between countries. They cannot investigate country level variables as moderators. They do not actually propose to do so in the planned analyses (or design table), but some of the introduction reads as though this is what they want to do. They can look at individual level moderators (as they propose to do using their ‘additional analyses.’ The moderators just cannot be at the level of the country. Ultimately, if they wanted to do such tests, they would be very underpowered, because power would stem in a large way from the number of countries in the test.

3. Concerns about measurement: The authors have an extensive plan to investigate the measurement properties of their variables. This is good, but ultimately, I am concerned about whether it is actually possible to do all of this in one project. The basic numeracy variable is a composite of pieces of existing measures. As such, it probably has the highest chance of successfully approximating a unidimensional factor. The COVID-19 health numeracy variable is constructed ad hoc, but modeled after existing measures, and the attitude/behavior measures appear to be constructed entirely ad hoc. The attitude/behavior measure, in particular, is unlikely to approximate a unidimensional factor (just based on my experience with these things – you almost certainly aren’t going to get it right straight out of the chute). The authors plan to collect 200 independent cases as a pilot study to check the functioning of their measures, but it’s not clear that this will be enough data to do a decent job at this, nor is it apparent that we’d expect a generalizable solution to be born out of this pilot data. The authors discuss plans to investigate factor structure, but do not mention reliability and validity. Validity of the ad hoc measures appears to be assumed. Reliability could and should be tested. Furthermore, a backup plan is needed should reliability prove insufficient. With dichotomously scored items, reliability is often quite low.

4. Role of time: The health situation with the virus is unfolding rapidly and, as you note, on different time scales in the different countries under investigation. In what way do you plan to account for time, if at all? I would imagine the numeracy constructs are thought to be relatively temporally stable, but the DV (attitudes and behaviors) might be changing at a country level or local geographic level over time. For instance, if local rules and regulations change, increased compliance with mask wearing (for example) might be expected, but these changes would have nothing to do with numeracy. Likewise, if the virus spreads, objective risk goes up, which likely has a fairly direct link to attitudes/behaviors. If my country has basically eradicated the virus (haha, I wish), I will act differently than if it is widespread (even if I’m not particularly numerate). The correlation between numeracy and attitudes/behaviors might vary in strength over time, perhaps based on some unknowable combination of these national/local behavior shifts. There could also be issues with ceiling/floor effects that vary over time (which would again affect the strength of correlations). For example, if masks are the law and no mask carries a steep fine, you’ll have little variation in mask wearing.

5. There are several examples throughout the paper where sensible plans are described, but details are lacking. For instance, you say that device information (phone vs. computer) will be taken into account in the model, but you don't say exactly how. In general, in a registered report, it is fairly important to have details like this locked down to the extent possible.

6. The authors state that all cases will be analyzed using FIML, even if there is some limited missing data. This is a good choice and appropriate. What is missing is consideration of how exclusions (for data quality or for high levels of missing data) will play into the obtained sample size. I recommend that the authors consider collecting extra data (beyond 675 per country) to ensure that the final sample size will be above the target. Just make sure to specify the stopping rule in the next draft (e.g., We will sample 10% extra cases – 745 per country – to allow for exclusions and attrition. All available cases will be analyzed.)

7. You propose to do EFA then CFA on the same data, but this is not ideal. You might be able to do an EFA in your pilot data and then a CFA in the main data. I would recommend constructing parcels randomly regardless. How many parcels do you plan to specify per factor?

8. You mention that you will use the “standard procedures” for measurement invariance. I would cite a specific source on this topic (perhaps the Kline SEM textbook, if that is the approach you will follow). Likewise, you mention nested model tests and chi-square comparisons, but do not mention any other fit statistics. It is typical in this area to report CFI and RMSEA, at a minimum, and authors often interpret change in CFI and RMSEA between steps in the model. If you plan to do this, you should name your cutoffs and cite your source. Finally, you might consider interpreting effect sizes for invariance alongside your significance tests (see Nye & Drasgow, 2011 and their metric dMACs).

9. I don't believe latent mean differences are discussed in the text (I only see them in the table). You'll need scalar invariance, then latent variance and covariance (i.e., path) equality before testing differences in latent means. A lot of hurdles to clear before you can test this.

10. I'm not sure how the additional analyses should be handled. It might be better to explore and add them later as “exploratory,” rather than try to fully delineate here. Just one thing I considered: political ideology surely cannot be modelled consistently across countries, right? Do you have a way to measure ideology that you believe would be cross-nationally invariant?

Additional points:

- Hypotheses could be more clearly expressed (p. 7). The diagram was fairly clear, but the text was ambiguously worded.
- In addition to the .qsf file, post a copy of the .docx from Qualtrics so that people can see the questionnaire without loading it into Qualtrics.

Signed,
Katherine S. Corker

Decision letter (RSOS-201303.R0)

Dear Dr Ansari,

The Editors assigned to your stage one Registered Report ("Numeracy and COVID-19: examining interrelationships between numeracy, health numeracy and behaviour") have now received comments from reviewers. We would like you to revise your paper in accordance with the referee and editors suggestions which can be found below (not including confidential reports to the Editor). Please note this decision does not guarantee eventual acceptance.

When submitting your revised manuscript, you must respond to the comments made by the referees and upload a file "Response to Referees" in "Section 2 - File Upload". Please use this to document how you have responded to the comments, and the adjustments you have made. In order to expedite the processing of the revised manuscript, please be as specific as possible in your response.

Kind regards,
Andrew Dunn
Royal Society Open Science
openscience@royalsociety.org

on behalf of Professor Chris Chambers (Registered Reports Editor, Royal Society Open Science)
openscience@royalsociety.org

Associate Editor Comments to Author (Professor Chris Chambers):

Associate Editor: 1

Comments to the Author:

We now have a set of three very detailed and constructive reviews from a combination of field- and methodological experts (and on behalf of the journal I would like to convey my sincere thanks to the reviewers for the very helpful and rapid assessments). On the positive side, the reviewers are unanimous in noting the timeliness and value of the research question and general rigour of the proposed plans. On the other hand, the reviewers identify a range of issues that will need to be addressed to achieve IPA. Major concerns to address include justification of mediation analysis (R1.1), a lack of sufficient clarity/detail and risk of bias in the analysis plans (R1.2, R1.10, R2.5, R3.5, R3.6, R3.8, R3.9), a significant point about possible error or confusion in the power calculations (R3.10; and in a revision please share the code used for the power analysis as requested by the reviewer), and accounting for time and changing status of pandemic and associated health behaviours on a per-country basis (R3.4).

All of the above points falls within the scope of a major Stage 1 revision, but resting above all of these issues is what I see as an even weightier concern identified by all 3 reviewers: the reliability and robustness of measurement and the level of risk-planning associated with the preliminary/pilot study. As proposed, there appears to be a significant risk that main study will not be possible (or will need to be heavily deviated) due to the preliminary/pilot turning out to be unreliable or failing to validate assumptions. There are a number of ways this risk could be

addressed. One approach, suggested by Rev 1 would be to increase the size of the preliminary study, treating it as a training dataset and the main study as a holdout dataset. An alternative approach, suggested by Rev 2 would be to conduct the preliminary study now and include it in a revised Stage 1 submission.

From an editorial point of view, I think both suggestions have merit, but it may make the most sense -- and resolve the most uncertainties about measurement especially (see Rev 3) -- to conduct the preliminary study first and include it in a revised Stage 1 RR. If the authors decide to go down this route, I would suggest initially revising the manuscript with this plan in mind before running the preliminary study (settling as many of the other issues as possible), at which point, with the reviewers in agreement, the authors could activate the preliminary study. Once this is complete, you could return with a final Stage 1 revision that includes the pilot data to provide the necessary proof of concept. From this, if reviews are positive, a full Stage 1 IPA decision could be forthcoming. Normally this kind of process might be considered too slow for authors but I believe the rapid pace of the COVID-19 RR review process would make it feasible. I will leave this with the authors to consider as a potential way forward.

Comments to Author:

Reviewer: 1

Comments to the Author(s)

This is an interesting and timely proposal. In broad terms, the case for investigating the relationship between numeracy and COVID attitudes and behavior was compelling to me on both scientific and practical grounds. The methods seemed broadly sound to me, and were described at a good level of detail and with clear language (which is hard to do with these kinds of analyses, so it is a big credit to the authors for that). Below I detail some comments and concerns which I hope will be helpful:

1. My largest concern is about the focus on mediation, which is a complex causal process. The introduction did not elaborate on the theoretical or conceptual case for investigating mediation. Furthermore, the design, a cross-sectional survey, is not an appropriate one for testing hypotheses about mediation (e.g., Bullock et al., 2010). I believe it would be more appropriate to reframe this research as descriptive, and it would still be compelling. To what extent is numeracy associated with COVID attitudes and behavior? And how much is that association about general numeracy versus domain-specific numeracy? I think this reframing would not be very far from the general research questions set up in the introduction, and the research design could remain unchanged. It would lead to some changes in the analytic strategy, but not a complete overhaul.
2. Figure 1 shows arrows from observed indicators to latent variables, suggesting they are formative. However, the text suggests that the latent variables will be reflective (and I do not believe a model with formative indicators would be identified). Assuming the text is right, the figure should probably be changed.
3. The statement that English “minimizes potential linguistic influences on numerical processing” does not appear to be supported by the cited reference, which discusses differences among languages but does not present English as having fewer influences. It makes sense that the researchers are interested in studying the countries they are from and the single, most commonly spoken language in them. But it is not necessary to argue that there is some scientific advantage to what seems to me to be more of a practical decision.
4. I reviewed the materials and had some questions and comments about a few of the items:
 - a. COVID numeracy item 11 is: “In which country do you have a higher chance of being tested for COVID-19.” Doesn’t the answer depend on how each country decides who to test? It is possible

that the researchers are thinking of random testing, or mean “you” generically, but I am not sure the respondents will too.

b. COVID numeracy item 12: Would the answer depend on whether testing is done by random sampling vs. purposive (e.g., testing people with symptoms)?

c. COVID numeracy item 13: Would the answer depends on whether confirmed cases are retested (e.g., to see if they have “cleared” the virus)?

5. As described, the “pilot” study is not a pilot study (Leon et al., 2011), and smaller-N studies are generally not a good basis for a power analysis (Kraemer et al., 2006). I would describe it more as a preliminary psychometric study. The authors should address whether the sample size will be adequate for the planned analyses. (I am not confident that it is, but that’s just a gut response and I may well be wrong.) I also did not understand the sentence, “Third, mean scores of items for each of the main variables will be used as a rough guide to estimate the intercorrelations of the main variables” on page 14.

6. I do not think outlier removal is necessary for rating-scale responses that have well-defined ranges. It may be better to use robust estimation and inference methods, which I believe are straightforward to implement in Mplus.

7. On p. 16 the authors write that there may be a problem with spurious correlations. The issue isn’t spuriousness – residual correlations may be a real and reliable feature of these measures. Regardless of the terminology, I think parceling is a wise choice here. I encourage the authors to look at Little et al (2013), which cautions against random parceling in many situations. They review pros and cons of other strategies, including the strategy proposed here for subdimensions, as well as alternatives for dealing with subdimensions. (It may be that a domain-representative approach is better than the facet-representative one described here, so I’d encourage the authors to consider whether that’s the case.)

8. What if the proposed model does not fit? It is very common that structural equation models do not fit well as planned, an issue the authors seem to be anticipating when they discuss residual correlations. What criteria (fit thresholds etc.) will be used to decide if there is good-enough fit to proceed with interpreting the substantive results? And if the fit is not adequate, and the authors proceed to make modifications in the same data they will conduct main analyses in, there is a risk of overfitting. It may be that the preliminary psychometric study could be increased in sample size and treated like a training dataset, with modifications recorded in an interim preregistration, and then the main study run as if it’s a holdout dataset. I recognize that this would increase the cost and resources required and therefore is not a trivial suggestion. So I will add that some concessions to practicality may be appropriate here.

9. Will political leaning be analyzed across countries, or separately within each country? “Left” vs. “right” have different meanings, and are associated with different policy positions, in different countries.

10. Finally, a general comment. There are several places where the authors have left room for human judgment during the data analyses. For example they discuss running exploratory factor analyses to make decisions about parceling. And they will have to make decisions about whether measurement invariance is established, and when it is not, what to do about it. (The analyses plan here is underspecified.) I have preregistered SEMs before and I’m currently in the middle of a registered report of a measurement invariance study, which has given me a deep appreciation for how hard it is to preregister a complete decision tree in this kind of work. I think some room for researcher degrees of freedom is unavoidable in this kind of work and should not be an impediment to its acceptance as a registered report. A training-holdout plan as I described earlier might address some of these other issues as well. Alternatively, the authors may consider

whether some kind of blinded analysis strategy (MacCoun & Perlmutter, 2017) could be implemented here to shield those decisions from any potential biasing influence on the substantive results.

Sanjay Srivastava
(This is a signed review.)

References

Bullock, J. G., Green, D. P., & Ha, S. E. (2010). Yes, but what's the mechanism?(don't expect an easy answer). *Journal of personality and social psychology*, 98(4), 550.

Kraemer, H. C., Mintz, J., Noda, A., Tinklenberg, J., & Yesavage, J. A. (2006). Caution regarding the use of pilot studies to guide power calculations for study proposals. *Archives of general psychiatry*, 63(5), 484-489.

Leon, A. C., Davis, L. L., & Kraemer, H. C. (2011). The role and interpretation of pilot studies in clinical research. *Journal of psychiatric research*, 45(5), 626-629.

Little, T. D., Rhemtulla, M., Gibson, K., & Schoemann, A. M. (2013). Why the items versus parcels controversy needn't be one. *Psychological methods*, 18(3), 285.

MacCoun, R. J., & Perlmutter, S. (2017). Blind analysis as a correction for confirmatory bias in physics and in psychology. *Psychological science under scrutiny: Recent challenges and proposed solutions*, 297-322.

Reviewer: 2

Comments to the Author(s)

The Authors are planning to conduct a large survey across three countries (UK, USA and Canada) to test the relation between basic numeracy, health numeracy (about the COVID-19 pandemic) and self-reported attitudes/behaviours in relation to health (with reference to the COVID-19 pandemic). Basic numeracy is measured with tasks assessing number magnitude processing (analogue number line tasks) and conceptual knowledge of fractions and decimals. Health numeracy is measured with a (very nice) series of multiple-choice questions on graph literacy, proportions and odd ratios. Health-related attitudes/behaviours are measured with a series of questions measuring compliance and the perception of impact and severity of the pandemic. A positive relation is expected between basic numeracy and health-related attitudes/behaviours (whereby higher numeracy originates more compliant attitudes/behaviours), mediated by health numeracy. Differences between countries will be analysed and potential moderating variables are included in the design.

1. This is a well-presented and thorough proposal that aims to address an interesting research question. However, in its current version it includes a plan for a pilot that could perhaps have been conducted before submission and there are a few unclear points. The most limiting aspect of the current design is that the mediation analysis may show a relation between the latent constructs without controlling for general ability.

2. Abstract.

It is clear and concise, and mostly focuses on the primary goal of the study - apart from the final statement. Why are political views and anxiety tested? This is not the strongest component of the study and the variables (one measured with a single item) are included for exploratory purposes.

I think that only if robust evidence of moderation effects was found after data collection they could deserve a mention here.

3. Participants.

More information is needed about these prospective market research participants and whether Qualtrics panels are representative of the general population in the three countries of interest. Could they live in the same household and share the same information channels, behavioural changes etc.?

What compensation is offered?

4. Procedure and materials.

What information will be available about screen size and how will this be taken into account? At the end of the survey, participants are asked which type of device they are using but screen sizes may vary greatly even within the same type of device. Further, across devices there may be a systematic variation of response modality (mouse vs. touchpad vs. pen vs. finger).

5. Power analysis.

The Authors provide a power analysis for their mediation model, multigroup and moderation analyses and choose their sample size accordingly. What are the α -levels, and should correction for multiple testing be applied? Have missing data been taken into account? These may increase the sample size requirements by a rather large amount. In the mediation analysis, how likely is it that the size of the $X > Y$ effect is the same as that of $X > M$ and $M > Y$ effects?

Given that several of the measures included in this study have been created ad hoc, planning a pilot study before conducting the larger study is very appropriate. I wonder, however, whether the pilot study should not have been conducted before submitting an IPA request for the larger study. Depending on the results of the pilot study, changes to the power analysis and to the study materials may be opportune (i.e. these would not be in definitive form at the time of IPA).

6. Number magnitude processing.

More than any other tests, the number line tasks and their derived PAE indices seem to depend heavily on hand-eye coordination/tool precision (e.g. pen, mouse, finger, touchpad), in addition to basic numeracy skills. Could individual scores in the number line tasks be confounded by response means? Inclusion of a few calibration items may be useful.

7. Demographics and other potential moderators.

Please define SES before using the acronym.

8. Ending questions.

Typo: Devince  device

I am not convinced that somebody who has not been responding seriously that far will be reliable at the "seriousness check".

Reviewer: 3

Comments to the Author(s)

The current paper plans to test a mediational hypothesis linking basic numeracy with health attitudes and behavior through COVID-19 health numeracy. There are a number of strengths in the proposal, along with some important weaknesses. On the strength side, the team investigates an interesting question using a rigorous analytic plan, including extensive measurement invariance testing. I examined the materials on OSF, and things appear to be well organized (although there is not much there yet, beyond copies of the instruments). Key weaknesses and issues to address are outlined below:

1. Power analyses: The authors use Mplus to simulate a set of power analyses for their model. In the majority of analyses, the conclusion points to a sample size of 675 for each of their three planned countries. This is peculiar to see such consistency across several different analytic strategies. There are at least two possible explanations. First, the authors may have done a sensitivity analysis, working backward from sample size and desired power to effect size. There is nothing wrong with such an approach, but if this is the technique that was used, the paper should be explicit about it. The effect sizes vary across the different specifications while sample size stays constant, so this may be what was done. On the other hand, this consistency could point to an error in the analyses (perhaps some kind of limit in Mplus or some computational limit). 675 per country does seem lower than I would have expected, but the authors do not plan to model any of their variables at the country level (nor could they, with only three nations), so this might explain the final number. I would have looked at the power simulation code, but it was not provided on OSF so I could not. I recommend sharing that code at some point. Regarding expectations for power in the multi-site context, the authors might consult Arend and Schäfer (2019, *Psych Methods*). The multi-level approach that they discuss is complementary to the multi-group SEM approach employed here.

2. There is a bit of a mismatch between what the authors say they want to do in their introduction, and what they can do with just three countries. For instance, they state that they will be “investigating whether country-level factors in these three countries influence the mediation model.” What they can do with their multi-group approach is look at whether model loadings, intercepts, variances/covariances (paths), and latent means differ between countries. They cannot investigate country level variables as moderators. They do not actually propose to do so in the planned analyses (or design table), but some of the introduction reads as though this is what they want to do. They can look at individual level moderators (as they propose to do using their ‘additional analyses.’ The moderators just cannot be at the level of the country. Ultimately, if they wanted to do such tests, they would be very underpowered, because power would stem in a large way from the number of countries in the test.

3. Concerns about measurement: The authors have an extensive plan to investigate the measurement properties of their variables. This is good, but ultimately, I am concerned about whether it is actually possible to do all of this in one project. The basic numeracy variable is a composite of pieces of existing measures. As such, it probably has the highest chance of successfully approximating a unidimensional factor. The COVID-19 health numeracy variable is constructed ad hoc, but modeled after existing measures, and the attitude/behavior measures appear to be constructed entirely ad hoc. The attitude/behavior measure, in particular, is unlikely to approximate a unidimensional factor (just based on my experience with these things – you almost certainly aren’t going to get it right straight out of the chute). The authors plan to collect 200 independent cases as a pilot study to check the functioning of their measures, but it’s not clear that this will be enough data to do a decent job at this, nor is it apparent that we’d expect a generalizable solution to be born out of this pilot data. The authors discuss plans to investigate factor structure, but do not mention reliability and validity. Validity of the ad hoc measures appears to be assumed. Reliability could and should be tested. Furthermore, a backup plan is needed should reliability prove insufficient. With dichotomously scored items, reliability is often quite low.

4. Role of time: The health situation with the virus is unfolding rapidly and, as you note, on different time scales in the different countries under investigation. In what way do you plan to account for time, if at all? I would imagine the numeracy constructs are thought to be relatively temporally stable, but the DV (attitudes and behaviors) might be changing at a country level or local geographic level over time. For instance, if local rules and regulations change, increased compliance with mask wearing (for example) might be expected, but these changes would have nothing to do with numeracy. Likewise, if the virus spreads, objective risk goes up, which likely

has a fairly direct link to attitudes/behaviors. If my country has basically eradicated the virus (haha, I wish), I will act differently than if it is widespread (even if I'm not particularly numerate). The correlation between numeracy and attitudes/behaviors might vary in strength over time, perhaps based on some unknowable combination of these national/local behavior shifts. There could also be issues with ceiling/floor effects that vary over time (which would again affect the strength of correlations). For example, if masks are the law and no mask carries a steep fine, you'll have little variation in mask wearing.

5. There are several examples throughout the paper where sensible plans are described, but details are lacking. For instance, you say that device information (phone vs. computer) will be taken into account in the model, but you don't say exactly how. In general, in a registered report, it is fairly important to have details like this locked down to the extent possible.

6. The authors state that all cases will be analyzed using FIML, even if there is some limited missing data. This is a good choice and appropriate. What is missing is consideration of how exclusions (for data quality or for high levels of missing data) will play into the obtained sample size. I recommend that the authors consider collecting extra data (beyond 675 per country) to ensure that the final sample size will be above the target. Just make sure to specify the stopping rule in the next draft (e.g., We will sample 10% extra cases - 745 per country - to allow for exclusions and attrition. All available cases will be analyzed.)

7. You propose to do EFA then CFA on the same data, but this is not ideal. You might be able to do an EFA in your pilot data and then a CFA in the main data. I would recommend constructing parcels randomly regardless. How many parcels do you plan to specify per factor?

8. You mention that you will use the "standard procedures" for measurement invariance. I would cite a specific source on this topic (perhaps the Kline SEM textbook, if that is the approach you will follow). Likewise, you mention nested model tests and chi-square comparisons, but do not mention any other fit statistics. It is typical in this area to report CFI and RMSEA, at a minimum, and authors often interpret change in CFI and RMSEA between steps in the model. If you plan to do this, you should name your cutoffs and cite your source. Finally, you might consider interpreting effect sizes for invariance alongside your significance tests (see Nye & Drasgow, 2011 and their metric dMACs).

9. I don't believe latent mean differences are discussed in the text (I only see them in the table). You'll need scalar invariance, then latent variance and covariance (i.e., path) equality before testing differences in latent means. A lot of hurdles to clear before you can test this.

10. I'm not sure how the additional analyses should be handled. It might be better to explore and add them later as "exploratory," rather than try to fully delineate here. Just one thing I considered: political ideology surely cannot be modelled consistently across countries, right? Do you have a way to measure ideology that you believe would be cross-nationally invariant?

Additional points:

- Hypotheses could be more clearly expressed (p. 7). The diagram was fairly clear, but the text was ambiguously worded.
- In addition to the .qsif file, post a copy of the .docx from Qualtrics so that people can see the questionnaire without loading it into Qualtrics.

Signed,
Katherine S. Corker

Author's Response to Decision Letter for (RSOS-201303.R0)

See Appendix A.

RSOS-201303.R1 (Revision)

Review form: Reviewer 2

Do you have any ethical concerns with this paper?

No

Recommendation?

Accept in principle

Comments to the Author(s)

The Authors have responded satisfactorily to my points.

Review form: Reviewer 3 (Katherine Corker)

Do you have any ethical concerns with this paper?

No

Recommendation?

Accept with minor revision

Comments to the Author(s)

I revisited my previous points; the numbers below match my initial numbering scheme.

1. This point has been addressed.
2. This point has been addressed.
3. The plan to increase the pilot study sample size to 500 is a good one. Regarding whether you should sample from one country or all three, I have the same intuition as you: to collect from all three countries, but to ignore between country differences at the pilot stage. I don't have a strong justification for that intuition. Unresolved issues: You say you will examine convergent and discriminant validity, but you don't say how (well the revised text says a little more, but it is still missing important details). What variables will you use to test these validities? Or is the proposal only to test the validity of the internal structure of the measure (which does not really make sense to me)? Finally, it is typically not advisable to conduct an EFA and a CFA on the same data (as it will result in overfitting). I'm not clear why you propose to do so here (but maybe I misunderstand the plan).
4. This point has been addressed.
5. I'm not sure this is an ideal way to handle covariates (i.e., including them only if they are statistically significant; a data dependent choice). I guess if it were me, I would plan for my primary analysis to be covariate free, and then present the analysis with covariates all included as a robustness check. Or, perhaps you could commit to including or excluding the covariates in the

pilot stage, so that in the final test, the choice to include a covariate is not data dependent. I will look for some details on this at the next stage, when the pilot data are in.

6. OK, this should be noted in the registration document somewhere.

7. OK, for the main study, but I remain confused as to why it appears that an EFA and a CFA on the same data in the pilot study is proposed.

8. This point has been addressed.

9. This point has been addressed.

10. To clarify the point about exploratory analyses: Authors in RRs are always allowed to provide new exploratory results in sections that are subsequent to, and clearly delineated from, the pre-registered results sections. Planned explorations can, but are not required to be, disclosed at stage

1. Stage 1, rightly, focuses more on the confirmatory tests. It sounds like you have found an OK middle ground here, but just know that if something appears in the Stage 1 plan (even as exploratory), it must appear in the final report. So sometimes it's better to let the explorations be true explorations and show up later after you've (transparently) explored.

New point: Figure 4 is helpful to see the planned process for the main analysis (Figure 3 is nice too). I see that not all parameters in the final model are set a priori (e.g., modification indices, a data dependent feature, are invoked early in the tree). I also see that invariance testing happens after mediation testing, rather than before (which does not make sense to me). The structural model is needed before you can do the mediation test, right? You can't test your mediation model across countries unless you have invariance first. I will look for this modification in the next stage of the draft.

To summarize: I support the plan to conduct the revised pilot/preliminary study with N=500 and to use this study as the basis for a revised Stage 1 RR prior to the collection of the main dataset, but I remain skeptical that the measurement model will work out as proposed.. It might be useful to consider the circumstances that would cause the authors to abandon the plan to proceed to a second/final study - for instance if the measurement model in the pilot proves to need more changes than can be fruitfully examined in a single preliminary study. Figure 3 sheds some light on this ("substantial modifications" may be needed). Ultimately it may be hard to say ahead of time exactly how substantial the changes would need to be before more measurement work would need to be done ahead of a final hypothesis test.

Signed,

Katherine S. Corker

Decision letter (RSOS-201303.R1)

Dear Dr Ansari,

The Editors assigned to your revised Stage 1 Registered Report ("Numeracy and COVID-19: examining interrelationships between numeracy, health numeracy and behaviour") have now received comments from reviewers. We would like you to revise your paper in accordance with the referee and editors suggestions which can be found below (not including confidential reports to the Editor).

Please submit a copy of your revised paper within three weeks (i.e. by the 23-Sep-2020).

When submitting your revised manuscript, you must respond to the comments made by the referees and upload a file "Response to Referees". Please use this to document how you have responded to the comments, and the adjustments you have made. In order to expedite the processing of the revised manuscript, please be as specific as possible in your response.

on behalf of Professor Chris Chambers (Registered Reports Editor, Royal Society Open Science)
openscience@royalsociety.org

Associate Editor Comments to Author (Professor Chris Chambers):

The revised manuscript was re-assessed by the three original reviewers. All judge the manuscript to be improved, and are positive about the plan to perform an initial (larger) preliminary study as part of the Stage 1 process, prior to IPA.

There are however some remaining issues to address in the Stage 1 manuscript before progressing with this preliminary study. Foremost is the major concern of Reviewer 1 about untestable (and potentially unsafe) assumptions in the mediation model, with the reviewer (quite reasonably in my view) suggesting a more circumspect approach in which the focus is on association and conclusions regarding causality are caveated and speculative. Reviewer 3 also notes a number of details requiring further clarification.

Given the overall positive assessments, I would like to suggest the following way forward: that the authors submit a final Stage 1 revision addressing these points. In the interests of efficiency, I will assess this at desk without going back to the reviewers (just yet), and if I judge the remaining issues to be adequately settled (and especially Reviewer 3 point 3 + the comment regarding Figure 4, and the main concern of Reviewer 1) then I will issue an interim decision to keep the Stage 1 process active while the authors undertake the preliminary study. Then, once the preliminary study is complete, and the authors submit a further revised Stage 1 manuscript, I will reinvoke the reviewers, and if their assessments are positive then we can proceed to full IPA.

I remain at the authors' disposal if you have any questions about this process moving forward.

Comments to Author:

Reviewer: 1

Comments to the Author(s)

This is an improved manuscript. I continue to be impressed by the attention to methodological rigor and the clarity of the writing about some very complicated methods. And of course, I still find the research topic compelling.

I thank the authors for taking everything I said seriously and responding to everything. The authors' responses have put to rest most of the concerns I raised previously. Below I outline one objection I continue to have, and I offer some other comments on the revised plan.

1. While I appreciate that the authors took care in responding to my concern about mediation, that concern has not been put to rest for me. The authors have added justifications for the paths in the mediation model. Those justifications address the theoretical plausibility of those paths. But they do not address the conditions necessary to make inferences about them from data. That depends on the rest of the model being correct, and that is the source of my concern.

To be more explicit, for the model to be correctly specified and the mediation correctly estimated, every one of the following untestable assumptions needs to be true:

- a. The effect of COVID-19 health numeracy on basic numeracy is zero.
- b. No confounding variable has an effect on basic numeracy and health numeracy.
- c. The effect of health-related attitudes/behaviors on health numeracy is zero.
- d. No confounding third variable has an effect on health numeracy and health attitudes/behaviors.
- e. The effect of health attitudes/behaviors on basic numeracy is zero.
- f. No confounding third variable affects basic numeracy and health attitudes/behaviors.

I am not persuaded that all of these assumptions are safe to make. It seems plausible that there could be unmeasured confounding variables (SES, intelligence, access to health care, access to education, on and on). Some of the reverse paths also seem plausible, for example, that people can generalize from domain-specific numeracy skills, or that taking COVID seriously (in one's attitudes or behaviors) might motivate someone to read and learn about COVID statistics and testing.

While I understand the argument that an imperfect study can motivate a better one in the future, I also often see the opposite: caveats and hedges get lost after a study comes out and gets translated from journal article to press release to media (or from journal article to citation in future journal article). I would rather see this presented as a study of associations, with the causal story clearly marked as speculation only and not "supported" by a mediation analysis.

2. I really like the expansion of the preliminary study into more of a full-blown measurement study. I think it creates a greater chance of success for the ultimate main study. Below I have some comments on it.

a. I'm nitpicking here, but this is not a "pilot" study (see the references in my previous review). I'd call it a measurement study, or just Study 1.

b. In the cover letter, the authors mentioned they were weighing collecting all the data in one country versus in all 3. I can see merits to both and think this should be the authors' call. If it were me, I would lean toward the 3-country plan. It matches what the eventual main study will look like, and it gives the authors a chance to do some exploration of measurement invariance in that dataset.

c. The measurement study plan has some risk of overfitting. Ideally in a programmatic measurement investigation, you would do EFA and item trimming/retooling in one (or more) samples, and then do your CFA in a new sample. Otherwise, the CFA can be overfit because of capitalizing on chance in the EFA phase. Of note, this will not create a risk of overfitting in the second, main study. However, it does create a possibility that the measurement indices (fit, loadings, etc.) in the main study could look worse than they did in the overfit Study 1. Whether

that'll be a little or a lot is hard to say: speaking heuristically, the more changes the authors make in the EFA stage, the more I would worry about overfitting in the CFA. With all that being said, there is also a very important practical question of how much measurement research the authors should do before getting to the very-timely main question. The perfect should not be the enemy of the good. So again, I'm sharing some thoughts that I hope are helpful, but in the end I think this should be the authors' decision. It's a judgment call, and they will be in a good position to make it after they see how the Study 1 data looks.

d. Small thing, but should the "item parcels" section be moved up to Study 1?

I wish the authors the best of luck in this work.

Sanjay Srivastava
(This is a signed review.)

Reviewer: 2
Comments to the Author(s)
The Authors have responded satisfactorily to my points.

Reviewer: 3
Comments to the Author(s)
I revisited my previous points; the numbers below match my initial numbering scheme.

1. This point has been addressed.
2. This point has been addressed.
3. The plan to increase the pilot study sample size to 500 is a good one. Regarding whether you should sample from one country or all three, I have the same intuition as you: to collect from all three countries, but to ignore between country differences at the pilot stage. I don't have a strong justification for that intuition. Unresolved issues: You say you will examine convergent and discriminant validity, but you don't say how (well the revised text says a little more, but it is still missing important details). What variables will you use to test these validities? Or is the proposal only to test the validity of the internal structure of the measure (which does not really make sense to me)? Finally, it is typically not advisable to conduct an EFA and a CFA on the same data (as it will result in overfitting). I'm not clear why you propose to do so here (but maybe I misunderstand the plan).
4. This point has been addressed.
5. I'm not sure this is an ideal way to handle covariates (i.e., including them only if they are statistically significant; a data dependent choice). I guess if it were me, I would plan for my primary analysis to be covariate free, and then present the analysis with covariates all included as a robustness check. Or, perhaps you could commit to including or excluding the covariates in the pilot stage, so that in the final test, the choice to include a covariate is not data dependent. I will look for some details on this at the next stage, when the pilot data are in.
6. OK, this should be noted in the registration document somewhere.
7. OK, for the main study, but I remain confused as to why it appears that an EFA and a CFA on the same data in the pilot study is proposed.
8. This point has been addressed.
9. This point has been addressed.
10. To clarify the point about exploratory analyses: Authors in RRs are always allowed to provide new exploratory results in sections that are subsequent to, and clearly delineated from, the pre-registered results sections. Planned explorations can, but are not required to be, disclosed at stage 1. Stage 1, rightly, focuses more on the confirmatory tests. It sounds like you have found an OK

middle ground here, but just know that if something appears in the Stage 1 plan (even as exploratory), it must appear in the final report. So sometimes it's better to let the explorations be true explorations and show up later after you've (transparently) explored.

New point: Figure 4 is helpful to see the planned process for the main analysis (Figure 3 is nice too). I see that not all parameters in the final model are set a priori (e.g., modification indices, a data dependent feature, are invoked early in the tree). I also see that invariance testing happens after mediation testing, rather than before (which does not make sense to me). The structural model is needed before you can do the mediation test, right? You can't test your mediation model across countries unless you have invariance first. I will look for this modification in the next stage of the draft.

To summarize: I support the plan to conduct the revised pilot/preliminary study with N=500 and to use this study as the basis for a revised Stage 1 RR prior to the collection of the main dataset, but I remain skeptical that the measurement model will work out as proposed.. It might be useful to consider the circumstances that would cause the authors to abandon the plan to proceed to a second/final study - for instance if the measurement model in the pilot proves to need more changes than can be fruitfully examined in a single preliminary study. Figure 3 sheds some light on this ("substantial modifications" may be needed). Ultimately it may be hard to say ahead of time exactly how substantial the changes would need to be before more measurement work would need to be done ahead of a final hypothesis test.

Signed,
Katherine S. Corker

Author's Response to Decision Letter for (RSOS-201303.R1)

See Appendix B.

Decision letter (RSOS-201303.R2)

Dear Dr Ansari,

The Editors assigned to your Stage 1 Registered Report ("Numeracy and COVID-19: examining interrelationships between numeracy, health numeracy and behaviour") have now reached an interim Stage 1 decision based on your revised manuscript.

Please submit a copy of your revised paper including the results of the preliminary measurement study within three months (i.e. by the 22 December 2020).

To revise your manuscript, log into <http://mc.manuscriptcentral.com/rsos> and enter your Author Centre, where you will find your manuscript title listed under "Manuscripts with Decisions." Under "Actions," click on "Create a Revision." Your manuscript number has been

appended to denote a revision. Revise your manuscript and upload a new version through your Author Centre.

When submitting your revised manuscript, you must respond to the comments made by the referees and upload a file "Response to Referees". Please use this to document how you have responded to the comments, and the adjustments you have made. In order to expedite the processing of the revised manuscript, please be as specific as possible in your response.

on behalf of Professor Chris Chambers (Registered Reports Editor, Royal Society Open Science)
openscience@royalsociety.org

Associate Editor Comments to Author (Professor Chris Chambers):

Associate Editor

Comments to the Author:

Thank you for responding so thoroughly to the reviewers' comments from the last round. As promised, based on my assessment of the response and the revised manuscript, I am now issuing an interim desk decision without further review at this time. The decision is that the authors can proceed with the preliminary measurement study as described. I am noting the decision on the system as a Major Revision with a default completion window of 3 months (of course just let the journal office know if you need longer). Once the preliminary measurement study is complete, please then submit a further revision including the results and finalised protocol for the main study. This will then be returned to the Stage 1 reviewers for reappraisal.

Author's Response to Decision Letter for (RSOS-201303.R2)

See Appendix C.

RSOS-201303.R3

Review form: Reviewer 1 (Sanjay Srivastava)

Do you have any ethical concerns with this paper?

No

Recommendation?

Accept in principle

Comments to the Author(s)

I have enjoyed seeing this project develop, and I am excited for the researchers to be able to go forward with the main study.

The preliminary measurement study is well done, the decisions make sense to me, and I think proceeding as described in this manuscript is justified. The preliminary results surprised me a little bit – the correlations of health attitudes/behaviors with numeracy were small in magnitude and negative. That’s not at all a reason not to proceed! Just an observation. It will be interesting to see what happens in the main data, collection as well as to see whether or how country moderates the effects. It may be that individual differences matter more when you are not getting clear messages from authorities, as is the case here in the U.S. (versus Canada, where my impression is that there is much more consistent official messaging, so maybe it’s not up to each individual to figure things out).

I have only some minor comments:

1. The heading on page 9 is “Demographics and Other Potential Moderators,” and the paragraph below it refers both to mediation and moderation. But the mediation analysis was removed in response to my previous comments, and it appears that aside from country, the moderation analyses have also been removed in response to Reviewer 3. Table 1 also seems to still refer to mediation analyses. So these sections may need to be lightly cleaned up.

Also, at the risk of some pointless editorializing: it looks like the “exploratory” moderator analyses were taken out because Reviewer 3 stated that if they’re in the stage 1 proposal, they must be reported no matter what. I personally would be fine with a stage 1 manuscript that says certain things will be explored and only reported if the researchers see something interesting going on. That’s still being transparent, and it is probably what is going to happen anyway. But I am fine with it either way.

2. On p. 21, is there a typo or missing word in the first sentence at the top of the page? (“Subsequent to estimating...”)

3. Also on p. 21: Is it possible that some people did not answer the number line task because they do not understand number lines? If so, could that be correlated with low numeracy? Also, it appears that all of the data from those participants was excluded – what are the pros and cons of that vs. just excluding their number-line data? (in other words, is there any reason to think those participants might have given valid responses to other items) I just wanted to raise these questions for consideration – I’ll leave it to the authors to decide whether to change anything or not.

In closing I’ll restate what I hope I’ve said in previous reviews: I continue to be impressed with the researchers’ rigor, transparency, careful attention to detail, clear writing, and thoughtful approach to an important set of questions. I wish the best of luck to the researchers in moving forward!

Sanjay Srivastava
(This is a signed review.)

Decision letter (RSOS-201303.R3)

Dear Dr Ansari,

On behalf of the Editors, I am pleased to inform you that your Manuscript RSOS-201303.R3 entitled "Numeracy and COVID-19: examining interrelationships between numeracy, health numeracy and behaviour" has been accepted in principle for publication in Royal Society Open Science subject to minor revision in accordance with the referee and editor suggestions. Please find their comments at the end of this email.

The reviewers and handling editors have recommended publication, but also suggest some minor revisions to your manuscript. Therefore, I invite you to respond to the comments and revise your manuscript.

Please you submit the revised version of your manuscript within 7 days (i.e. by the 05-Dec-2020). If you do not think you will be able to meet this date please let me know immediately.

When submitting your revised manuscript, you will be able to respond to the comments made by the referees and you should upload a file "Response to Referees". You can use this to document any changes you make to the original manuscript. In order to expedite the processing of the revised manuscript, please be as specific as possible in your response to the referees.

Full author guidelines can be found here <https://royalsocietypublishing.org/rsos/registered-reports>.

on behalf of Professor Chris Chambers (Subject Editor, Royal Society Open Science)
openscience@royalsociety.org

Associate Editor Comments to Author (Professor Chris Chambers):

Associate Editor: 1

Comments to the Author:

The revised manuscript was returned to one of the previous Stage 1 reviewers (Sanjay Srivastava) who I am happy to report is now satisfied with the submission (as am I) and recommends IPA. As you will see there are a couple of minor points remaining to address in his review to ensure that the Stage 1 manuscript that receives IPA is complete and accurate.

In revising, please submit a clean version of the final Stage 1 manuscript, together with brief cover letter documenting any final changes to the manuscript. For this COVID19 RR initiative, RSOS registers approved Stage 1 protocols on behalf of authors. To do this we need your

permission and some information, so please also submit with your revision the attached preregistration checklist (or email the checklist to chambersc1@cardiff.ac.uk).

Reviewer comments to Author:

Reviewer: 1

Comments to the Author(s)

I have enjoyed seeing this project develop, and I am excited for the researchers to be able to go forward with the main study.

The preliminary measurement study is well done, the decisions make sense to me, and I think proceeding as described in this manuscript is justified. The preliminary results surprised me a little bit – the correlations of health attitudes/behaviors with numeracy were small in magnitude and negative. That’s not at all a reason not to proceed! Just an observation. It will be interesting to see what happens in the main data, collection as well as to see whether or how country moderates the effects. It may be that individual differences matter more when you are not getting clear messages from authorities, as is the case here in the U.S. (versus Canada, where my impression is that there is much more consistent official messaging, so maybe it’s not up to each individual to figure things out).

I have only some minor comments:

1. The heading on page 9 is “Demographics and Other Potential Moderators,” and the paragraph below it refers both to mediation and moderation. But the mediation analysis was removed in response to my previous comments, and it appears that aside from country, the moderation analyses have also been removed in response to Reviewer 3. Table 1 also seems to still refer to mediation analyses. So these sections may need to be lightly cleaned up.

Also, at the risk of some pointless editorializing: it looks like the “exploratory” moderator analyses were taken out because Reviewer 3 stated that if they’re in the stage 1 proposal, they must be reported no matter what. I personally would be fine with a stage 1 manuscript that says certain things will be explored and only reported if the researchers see something interesting going on. That’s still being transparent, and it is probably what is going to happen anyway. But I am fine with it either way.

2. On p. 21, is there a typo or missing word in the first sentence at the top of the page? (“Subsequent to estimating...”)

3. Also on p. 21: Is it possible that some people did not answer the number line task because they do not understand number lines? If so, could that be correlated with low numeracy? Also, it appears that all of the data from those participants was excluded – what are the pros and cons of that vs. just excluding their number-line data? (in other words, is there any reason to think those participants might have given valid responses to other items) I just wanted to raise these questions for consideration – I’ll leave it to the authors to decide whether to change anything or not.

In closing I’ll restate what I hope I’ve said in previous reviews: I continue to be impressed with the researchers’ rigor, transparency, careful attention to detail, clear writing, and thoughtful approach to an important set of questions. I wish the best of luck to the researchers in moving forward!

Sanjay Srivastava

(This is a signed review.)

Author's Response to Decision Letter for (RSOS-201303.R3)

See Appendix D.

Decision letter (RSOS-201303.R4)

Dear Dr Ansari

On behalf of the Editor, I am pleased to inform you that your Manuscript RSOS-201303.R4 entitled "Numeracy and COVID-19: examining interrelationships between numeracy, health numeracy and behaviour" has been accepted in principle for publication in Royal Society Open Science.

You may now progress to Stage 2 and complete the study as approved.

Please read the following email carefully

Your accepted Stage 1 manuscript has been publicly registered at: <https://doi.org/10.17605/OSF.IO/FG57X>

Following completion of your study, we invite you to resubmit your paper for peer review as a Stage 2 Registered Report. Please note that your manuscript can still be rejected for publication at Stage 2 if the Editors consider any of the following conditions to be met:

- The results were unable to test the authors' proposed hypotheses by failing to meet the approved outcome-neutral criteria.
- The authors altered the Introduction, rationale, or hypotheses, as approved in the Stage 1 submission.
- The authors failed to adhere closely to the registered experimental procedures. Please note that any deviations from the approved experimental procedures must be communicated to the editor immediately for approval, and prior to the completion of data collection. Failure to do so can result in revocation of in-principle acceptance and rejection at Stage 2 (see complete guidelines for further information).
- Any post-hoc (unregistered) analyses were either unjustified, insufficiently caveated, or overly dominant in shaping the authors' conclusions.
- The authors' conclusions were not justified given the data obtained.

We encourage you to read the complete guidelines for authors concerning Stage 2 submissions at <https://royalsocietypublishing.org/rsos/registered-reports#ReviewerGuideRegRep>. Please especially note the requirements for data sharing, reporting the URL of the independently registered protocol, and that withdrawing your manuscript will result in publication of a Withdrawn Registration.

Once again, thank you for submitting your manuscript to Royal Society Open Science and we look forward to receiving your Stage 2 submission. If you have any questions at all, please do not hesitate to get in touch. We look forward to hearing from you shortly with the anticipated submission date for your stage two manuscript.

on behalf of Professor Chris Chambers (Registered Reports Editor, Royal Society Open Science)
openscience@royalsociety.org

Author's Response to Decision Letter for (RSOS-201303.R4)

See Appendix E.

RSOS-201303.R5

Review form: Reviewer 1 (Sanjay Srivastava)

Is the manuscript scientifically sound in its present form?

Yes

Are the interpretations and conclusions justified by the results?

Yes

Is the language acceptable?

Yes

Do you have any ethical concerns with this paper?

No

Have you any concerns about statistical analyses in this paper?

No

Recommendation?

Accept with minor revision

Comments to the Author(s)

SPECIFIC QUERIES

"Whether the data are able to test the authors' proposed hypotheses by passing the approved outcome-neutral criteria (such as absence of floor and ceiling effects or success of positive controls)"

The data seem to be of high quality and able to answer the substantive questions. In the discussion, the authors do raise the possibility of a ceiling effect on COVID attitudes and behaviors. I had a little trouble interpreting the distributional statistics in Table 1; plots of distributions would have helped me more. My preliminary sense is that any ceiling effect is not strong enough to undermine the analysis, but perhaps the authors could address this.

"Whether the Introduction, rationale and stated hypotheses are the same as the approved Stage 1 submission"

I did not conduct a word-for-word comparison, but as best as I can tell, yes.

"Whether the authors adhered precisely to the registered experimental procedures"

Yes.

Some details about the preliminary study were moved to a supplement, but this was appropriate.

I did pick up that in the exclusion section, there was a sentence in the Stage 1 manuscript explaining an exclusion rule for people who had been vaccinated or participated in vaccine trials. The Stage 2 manuscript dropped this explanatory sentence, but it appears that the rule was still followed.

"Where applicable, whether any unregistered exploratory statistical analyses are justified, methodologically sound, and informative"

Yes, the additional analyses were sensible and informative.

Regarding the cultural cognition explanation in the discussion: Could the political leaning variable speak to this (at least in the U.S., which I am familiar with)?

"Whether the authors' conclusions are justified given the data"

Yes, the conclusions were justified. The data did not support a major hypothesis, and the authors are quite clear about this and thoughtful about what it might mean.

GENERAL COMMENT

This is a terrific manuscript. Important question, rigorous analyses, thoughtful discussion. And I can only imagine that it must have been a lot of work. Truly impressive!

Sanjay Srivastava
(signed review)

Review form: Reviewer 2

Is the manuscript scientifically sound in its present form?

Yes

Are the interpretations and conclusions justified by the results?

Yes

Is the language acceptable?

Yes

Do you have any ethical concerns with this paper?

No

Have you any concerns about statistical analyses in this paper?

No

Recommendation?

Accept with minor revision

Comments to the Author(s)

The authors adhered to the preregistered protocol. The null result for the relation between numeracy and COVID-19-related attitudes/behavior has been commented on with scientific rigour and the results of the analyses do not seem to have been overinterpreted.

A couple of points to address:

There is some ambiguity in the abstract, whereby the UK and Canada are said to show higher numeracy skills than the US. However this applies to only one of the two numeracy measures (COVID-related numeracy). For basic numeracy, no significant difference was found between the UK and Canada or between the US and Canada.

On page 41/69 the authors define and discuss the cultural cognition hypothesis in relation to the lack of association between numeracy and attitude/behavior. Any results concerning other variables included in the design, such as SES, political leaning, etc. would be of relevance and may help support/discard this claim.

Review form: Reviewer 3 (Katherine Corker)**Is the manuscript scientifically sound in its present form?**

No

Are the interpretations and conclusions justified by the results?

No

Is the language acceptable?

Yes

Do you have any ethical concerns with this paper?

No

Have you any concerns about statistical analyses in this paper?

Yes

Recommendation?

Major revision

Comments to the Author(s)

I have re-read the stage 2 manuscript in full. I have briefly reviewed the materials on OSF (note: the view only links are broken, but the unblinded link to the full project worked fine). Here are

things I noticed in no particular order. Overarching notes: Points 1 and 2 (below) need to be addressed for clarity of methods, but I doubt study conclusions will change. Point 3 might change study conclusions because standardization prior to parceling changes the variance structure of the items (because responses are nested within countries). Point 9 is the most significant flaw, reflecting an error in analysis, which will need to be corrected prior to moving forward. Point 12 also requires serious consideration in the discussion section.

1. Regarding participant exclusions: please confirm that what you did here is what was pre-registered. If there was a deviation, please label it as a deviation. The final text should say something like, "As pre-registered...[what you did]. Additionally, we later decided to ... [how you deviated]." If there are extensive deviations from the pre-registered plan, then report the main analyses following the pre-registered plan alongside analysis based on the subset of participants with the authors' preferred deviation as a robustness check.

2. Treatment of missing data: I do not believe that analysis of missing data was part of the pre-registered plan (but my memory might be poor). If I am correct, then the decision to include additional covariates in the analysis is a deviation from the pre-registered plan. If I am correct, then it should be marked as a deviation, and as with participant exclusions, results should be presented following the pre-registered plan with the deviation provided as a robustness check.

3. Item parcels: You made the choice (seemingly post hoc, because the text is in red) to standardize items prior to parcel construction. Was this standardization done within each country separately? Or in the sample as a whole? In general, standardization when doing analyses with nested data (here, participants within countries) is a no-no because it changes the variance structure of the data. Relatedly, the descriptive statistics in Table 1 apparently reflect this standardization, obscuring relevant information like how accurate participants actually were in their numeracy knowledge. Unlike so much other psychology research, the scale of measurement here is not arbitrary, so the raw means and standard deviations are of interest. Therefore, I recommend a supplemental table with raw descriptive statistics, separated by country. I also recommend reconsidering the standardization choice prior to parceling for the main analysis. Is there another way to score the items before combining them that respects the multi-level data structure? Perhaps POMP (percentage of maximum possible)? This critique also applies to the measurement pilot study (Table S2).

4. Related to the footnote in Table 1, are these statistics (means/variances/skew/kurtosis) raw values from the data, or estimates from a model? If from a model, provide the raw values somewhere, preferably stratified by country.

5. I like the way the two-factor vs. three-factor model for numeracy was handled (page 32). Make sure to correct the invariance analyses in Appendix C per the notes in point 9 below.

6. I'm not sure why correlations are tested before invariance testing is done. This might not be a big deal, but it confused me.

7. Table 4 duplicates information from the text. I prefer the table and am not sure the repeat of the fit statistics is needed in both spots.

8. On p. 18, you report that strict invariance (i.e., equivalent residuals) will not be tested, yet strict invariance appears in Table 4. Then on p. 36, you use "strict invariance" in a distinct way, to refer to latent variance and path equality. This analysis should be relabeled, because as far as I know, most people use "strict" invariance to refer to equivalent residuals, and not to equal latent variances and path equality.

9. Degrees of freedom in Table 4. The change in df from configural to metric is 12, whereas the change from metric to scalar is 18. I believe this reflects a specification error, because the change in df for both of these steps should be the same. Because you identified your factors with latent mean = 0 and variance = 1, only one country's latent variance (and then mean) should be so fixed when the equality constraint for loadings (and then intercepts) are added. So the invariance analyses will need to be re-run and re-reported. I didn't check, but this may affect the pilot study as well. This issue may also have downstream effects on the main hypothesis tests, but I didn't look into things sufficiently to determine for sure. Basically if the scalar model was misspecified, then the subsequent analyses that come after this point might also be misspecified. You have to count the change in df from one model to the next to be sure. What you report doing in the last paragraph on p. 36 is what should be done in the scalar model. Latent means are relaxed as equality constraints for intercepts are added, then later when you want to test the equality of latent means, an equality constraint for the latent means is re-added. That test - the difference between the model with the latent means forced to equality vs. one indicator country - is missing, meaning that there is currently no test of your claim that "participants in both Canada and the UK had higher COVID-19 Health Numeracy Skills when compared with participants in the US. While the UK had higher Basic Numeracy Skills than participants in the US but not Canada" (p. 36).

10. Exploration begins on page 38 (per the paper's headings), but several references to exploration appear in the previous sections of the paper. I believe it would be cleaner/clearer if all of the exploratory analyses were separated from the confirmatory analyses. As it stands, it is hard to find the confirmatory tests, given that they are mixed in with other analyses.

11. Page 42 refers to a "negative correlation ... between basic numeracy and COVID-19 attitudes and behaviours" but the correlation you found was not statistically different from zero, making this discussion confusing and possibly misleading.

12. An additional unmentioned limitation is shortcomings related to the COVID attitudes and behavior measure. If that measure is unreliable, we would not expect correlations with it to be very strong. I might have missed it, but is there discussion of the reliability/validity of that measure somewhere? By validity, I do not mean distinctiveness from the numeracy measures, but rather accuracy of the measure as established (perhaps in other research) by relations with other measures. The ceiling effect on this measure could also substantially limit its correlations with other measures.

13. Page 23: "missing random responses throughout the questionnaire" - consider changing to "missing sporadic responses throughout the questionnaire" because random has specific connotations in this context.

Signed,
Katherine S. Corker

Decision letter (RSOS-201303.R5)

Dear Dr Ansari:

On behalf of the Editor, I am pleased to inform you that your Stage 2 Registered Report RSOS-201303.R5 entitled "Numeracy and COVID-19: examining interrelationships between numeracy, health numeracy and behaviour" has been deemed suitable for publication in Royal Society Open Science subject to minor revision in accordance with the referee suggestions. Please find the referees' comments at the end of this email.

The reviewers and Subject Editor have recommended publication, but also suggest some minor revisions to your manuscript. Therefore, I invite you to respond to the comments and revise your manuscript.

Please also ensure that all the below editorial sections are included where appropriate -- if any section is not applicable to your manuscript, please can we ask you to nevertheless include the heading, but explicitly state that the heading is inapplicable. An example of these sections is attached with this email.

- Ethics statement

- Data accessibility

If you wish to submit your supporting data or code to Dryad (<http://datadryad.org/>), or modify your current submission to dryad, please use the following link:
[http://datadryad.org/submit?journalID=RSOS&manu=\(Document not available\)](http://datadryad.org/submit?journalID=RSOS&manu=(Document not available))

- Competing interests

- Authors' contributions

AB carried out the molecular lab work, participated in data analysis, carried out sequence alignments, participated in the design of the study and drafted the manuscript; CD carried out

the statistical analyses; EF collected field data; GH conceived of the study, designed the study, coordinated the study and helped draft the manuscript. All authors gave final approval for publication.

- Acknowledgements

- Funding statement

Because the schedule for publication is very tight, it is a condition of publication that you submit the revised version of your manuscript within 7 days (i.e. by the 19-Oct-2021). If you do not think you will be able to meet this date please let me know immediately.

on behalf of Professor Chris Chambers
 (Registered Reports Editor, Royal Society Open Science)
 openscience@royalsociety.org

Associate Editor Comments to Author (Professor Chris Chambers):

Associate Editor: 1

Comments to the Author:

The three reviewers who assessed the manuscript at Stage 1 kindly returned to evaluate the Stage 2 manuscript. As you will see, the assessments are broadly positive but with some issues to address to achieve Stage 2 acceptance, including clarity of deviations from protocol, distinction of confirmatory and exploratory outcomes, correction of errors, clarity of the presentation of the results, and consideration of study limitations. Reviewer 3 provides a particularly detailed assessment. Please revise and respond thoroughly to all points. I will consider a revised submission carefully at desk and will only return to the reviewers if I feel it is necessary.

Reviewer: 1

Comments to the Author(s)

SPECIFIC QUERIES

"Whether the data are able to test the authors' proposed hypotheses by passing the approved outcome-neutral criteria (such as absence of floor and ceiling effects or success of positive controls)"

The data seem to be of high quality and able to answer the substantive questions. In the discussion, the authors do raise the possibility of a ceiling effect on COVID attitudes and behaviors. I had a little trouble interpreting the distributional statistics in Table 1; plots of distributions would have helped me more. My preliminary sense is that any ceiling effect is not strong enough to undermine the analysis, but perhaps the authors could address this.

"Whether the Introduction, rationale and stated hypotheses are the same as the approved Stage 1 submission"

I did not conduct a word-for-word comparison, but as best as I can tell, yes.

"Whether the authors adhered precisely to the registered experimental procedures"

Yes.

Some details about the preliminary study were moved to a supplement, but this was appropriate.

I did pick up that in the exclusion section, there was a sentence in the Stage 1 manuscript explaining an exclusion rule for people who had been vaccinated or participated in vaccine trials.

The Stage 2 manuscript dropped this explanatory sentence, but it appears that the rule was still followed.

"Where applicable, whether any unregistered exploratory statistical analyses are justified, methodologically sound, and informative"

Yes, the additional analyses were sensible and informative.

Regarding the cultural cognition explanation in the discussion: Could the political leaning variable speak to this (at least in the U.S., which I am familiar with)?

"Whether the authors' conclusions are justified given the data"

Yes, the conclusions were justified. The data did not support a major hypothesis, and the authors are quite clear about this and thoughtful about what it might mean.

GENERAL COMMENT

This is a terrific manuscript. Important question, rigorous analyses, thoughtful discussion. And I can only imagine that it must have been a lot of work. Truly impressive!

Sanjay Srivastava
(signed review)

Comments to Author:

Reviewer: 2

Comments to the Author(s)

The authors adhered to the preregistered protocol. The null result for the relation between numeracy and COVID-19-related attitudes/behavior has been commented on with scientific rigour and the results of the analyses do not seem to have been overinterpreted.

A couple of points to address:

There is some ambiguity in the abstract, whereby the UK and Canada are said to show higher numeracy skills than the US. However this applies to only one of the two numeracy measures (COVID-related numeracy). For basic numeracy, no significant difference was found between the UK and Canada or between the US and Canada.

On page 41/69 the authors define and discuss the cultural cognition hypothesis in relation to the lack of association between numeracy and attitude/behavior. Any results concerning other variables included in the design, such as SES, political leaning, etc. would be of relevance and may help support/discard this claim.

Reviewer: 3

Comments to the Author(s)

I have re-read the stage 2 manuscript in full. I have briefly reviewed the materials on OSF (note: the view only links are broken, but the unblinded link to the full project worked fine). Here are things I noticed in no particular order. Overarching notes: Points 1 and 2 (below) need to be addressed for clarity of methods, but I doubt study conclusions will change. Point 3 might change study conclusions because standardization prior to parceling changes the variance structure of the items (because responses are nested within countries). Point 9 is the most significant flaw,

reflecting an error in analysis, which will need to be corrected prior to moving forward. Point 12 also requires serious consideration in the discussion section.

1. Regarding participant exclusions: please confirm that what you did here is what was pre-registered. If there was a deviation, please label it as a deviation. The final text should say something like, "As pre-registered....[what you did]. Additionally, we later decided to ... [how you deviated]." If there are extensive deviations from the pre-registered plan, then report the main analyses following the pre-registered plan alongside analysis based on the subset of participants with the authors' preferred deviation as a robustness check.
2. Treatment of missing data: I do not believe that analysis of missing data was part of the pre-registered plan (but my memory might be poor). If I am correct, then the decision to include additional covariates in the analysis is a deviation from the pre-registered plan. If I am correct, then it should be marked as a deviation, and as with participant exclusions, results should be presented following the pre-registered plan with the deviation provided as a robustness check.
3. Item parcels: You made the choice (seemingly post hoc, because the text is in red) to standardize items prior to parcel construction. Was this standardization done within each country separately? Or in the sample as a whole? In general, standardization when doing analyses with nested data (here, participants within countries) is a no-no because it changes the variance structure of the data. Relatedly, the descriptive statistics in Table 1 apparently reflect this standardization, obscuring relevant information like how accurate participants actually were in their numeracy knowledge. Unlike so much other psychology research, the scale of measurement here is not arbitrary, so the raw means and standard deviations are of interest. Therefore, I recommend a supplemental table with raw descriptive statistics, separated by country. I also recommend reconsidering the standardization choice prior to parceling for the main analysis. Is there another way to score the items before combining them that respects the multi-level data structure? Perhaps POMP (percentage of maximum possible)? This critique also applies to the measurement pilot study (Table S2).
4. Related to the footnote in Table 1, are these statistics (means/variances/skew/kurtosis) raw values from the data, or estimates from a model? If from a model, provide the raw values somewhere, preferably stratified by country.
5. I like the way the two-factor vs. three-factor model for numeracy was handled (page 32). Make sure to correct the invariance analyses in Appendix C per the notes in point 9 below.
6. I'm not sure why correlations are tested before invariance testing is done. This might not be a big deal, but it confused me.
7. Table 4 duplicates information from the text. I prefer the table and am not sure the repeat of the fit statistics is needed in both spots.
8. On p. 18, you report that strict invariance (i.e., equivalent residuals) will not be tested, yet strict invariance appears in Table 4. Then on p. 36, you use "strict invariance" in a distinct way, to refer to latent variance and path equality. This analysis should be relabeled, because as far as I know, most people use "strict" invariance to refer to equivalent residuals, and not to equal latent variances and path equality.
9. Degrees of freedom in Table 4. The change in df from configural to metric is 12, whereas the change from metric to scalar is 18. I believe this reflects a specification error, because the change in df for both of these steps should be the same. Because you identified your factors with latent mean = 0 and variance = 1, only one country's latent variance (and then mean) should be so fixed

when the equality constraint for loadings (and then intercepts) are added. So the invariance analyses will need to be re-run and re-reported. I didn't check, but this may affect the pilot study as well. This issue may also have downstream effects on the main hypothesis tests, but I didn't look into things sufficiently to determine for sure. Basically if the scalar model was misspecified, then the subsequent analyses that come after this point might also be misspecified. You have to count the change in df from one model to the next to be sure. What you report doing in the last paragraph on p. 36 is what should be done in the scalar model. Latent means are relaxed as equality constraints for intercepts are added, then later when you want to test the equality of latent means, an equality constraint for the latent means is re-added. That test – the difference between the model with the latent means forced to equality vs. one indicator country – is missing, meaning that there is currently no test of your claim that “participants in both Canada and the UK had higher COVID-19 Health Numeracy Skills when compared with participants in the US. While the UK had higher Basic Numeracy Skills than participants in the US but not Canada” (p. 36).

10. Exploration begins on page 38 (per the paper's headings), but several references to exploration appear in the previous sections of the paper. I believe it would be cleaner/clearer if all of the exploratory analyses were separated from the confirmatory analyses. As it stands, it is hard to find the confirmatory tests, given that they are mixed in with other analyses.

11. Page 42 refers to a “negative correlation ... between basic numeracy and COVID-19 attitudes and behaviours” but the correlation you found was not statistically different from zero, making this discussion confusing and possibly misleading.

12. An additional unmentioned limitation is shortcomings related to the COVID attitudes and behavior measure. If that measure is unreliable, we would not expect correlations with it to be very strong. I might have missed it, but is there discussion of the reliability/validity of that measure somewhere? By validity, I do not mean distinctiveness from the numeracy measures, but rather accuracy of the measure as established (perhaps in other research) by relations with other measures. The ceiling effect on this measure could also substantially limit its correlations with other measures.

13. Page 23: “missing random responses throughout the questionnaire” – consider changing to “missing sporadic responses throughout the questionnaire” because random has specific connotations in this context.

Signed,
Katherine S. Corker

Author's Response to Decision Letter for (RSOS-201303.R5)

See Appendix F.

Decision letter (RSOS-201303.R6)

Dear Dr Ansari:

It is a pleasure to accept your manuscript entitled "Numeracy and COVID-19: examining interrelationships between numeracy, health numeracy and behaviour" in its current form for publication in Royal Society Open Science. There is a final comment below from the action editor concerning an error that can be corrected at the proof stage.

COVID-19 rapid publication process:

We are taking steps to expedite the publication of research relevant to the pandemic. If you wish, you can opt to have your paper published as soon as it is ready, rather than waiting for it to be published the scheduled Wednesday.

This means your paper will not be included in the weekly media round-up which the Society sends to journalists ahead of publication. However, it will still appear in the COVID-19 Publishing Collection which journalists will be directed to each week (<https://royalsocietypublishing.org/topic/special-collections/novel-coronavirus-outbreak>).

If you wish to have your paper considered for immediate publication, or to discuss further, please notify openscience_proofs@royalsociety.org and press@royalsociety.org when you respond to this email.

Thank you for your fine contribution. On behalf of the Editors of Royal Society Open Science, we look forward to your continued contributions to the journal.

on behalf of Professor Chris Chambers (Subject Editor)
openscience@royalsociety.org

Associate Editor Comments to Author (Professor Chris Chambers):

Associate Editor

Comments to the Author:

The response and revisions are sufficient I am happy to accept the manuscript as-is. Just one thing I picked up on a final read: the following sentence in the Abstract has a typo: "However, predictions, neither basic numeracy nor COVID-19 health numeracy was related to COVID-19 health-related attitudes and behaviours (e.g., follow experts' recommendations on social distancing, wearing masks etc.)" I guess this should read "contrary to predictions" or "predictions" itself could be deleted. This can be corrected at the proof stage and there is no need for a further revision. Congratulations on a timely and important piece of work.

RSOS-201303

Point-by-point Responses to Editor and Reviewers

Dear Editor and Reviewers:

Thank you very much for your rapid and incredibly constructive reviews of our Stage 1 Manuscript. We have carefully considered all of your comments. In what follows below we provide a point-by-point response to all of the comments made. Our responses are in blue.

Associate Editor Comments to Author (Professor Chris Chambers):

Associate Editor: 1

Comments to the Author:

We now have a set of three very detailed and constructive reviews from a combination of field- and methodological experts (and on behalf of the journal I would like to convey my sincere thanks to the reviewers for the very helpful and rapid assessments). On the positive side, the reviewers are unanimous in noting the timeliness and value of the research question and general rigour of the proposed plans. On the other hand, the reviewers identify a range of issues that will need to be addressed to achieve IPA. Major concerns to address include justification of mediation analysis (R1.1), a lack of sufficient clarity/detail and risk of bias in the analysis plans (R1.2, R1.10, R2.5, R3.5, R3.6, R3.8, R3.9), a significant point about possible error or confusion in the power calculations (R3.10; and in a revision please share the code used for the power analysis as requested by the reviewer), and accounting for time and changing status of pandemic and associated health behaviours on a per-country basis (R3.4).

All of the above points falls within the scope of a major Stage 1 revision, but resting above all of these issues is what I see as an even weightier concern identified by all 3 reviewers: the reliability and robustness of measurement and the level of risk-planning associated with the preliminary/pilot study. As proposed, there appears to be a significant risk that main study will not be possible (or will need to be heavily deviated) due to the preliminary/pilot turning out to be unreliable or failing to validate assumptions. There are a number of ways this risk could be addressed. One approach, suggested by Rev 1 would be to increase the size of the preliminary study, treating it as a training dataset and the main study as a holdout dataset. And alternative approach, suggest by Rev 2 would be to conduct the preliminary study now and include in a revised Stage 1 submission.

From an editorial point of view, I think both suggestions have merit, but it may make the most sense -- and resolve the most uncertainties about measurement especially (see Rev 3) -- to conduct the preliminary study first and include it in a revised Stage 1 RR. If the authors decide to go down this route, I would suggest initially revising the manuscript with this plan in+ mind before running the preliminary study (settling as many of the other issues as possible), at which point, with the reviewers in agreement, the authors could activate the preliminary study. Once this is complete, you could return with a final Stage 1 revision that includes the pilot data to

provide the necessary proof of concept. From this, if reviews are positive, a full Stage 1 IPA decision could be forthcoming. Normally this kind of process might be considered too slow for authors but I believe the rapid pace of the COVID-19 RR review process would make it feasible. I will leave this with the authors to consider as potential way forward.

Author Response: We thank the editor for their time in reviewing the manuscript and the expert reviewers comments. We believe this feedback has enabled us to put together an improved Stage 1 proposal that is both more comprehensive and more robust to the potentialities of the planned research. We agree with the editor's assessment that the most sensible way forward is to (1) submit our current revisions to the planned study (and preliminary study), (2) receive feedback and eventual approval to move forward, and then (3) conduct the preliminary study and include this analysis in another Stage 1 revision. With this goal in mind, each reviewer's comments have been separated and responded to systematically. Reviewer's comments are in black and the authors' responses are in blue.

Comments to Author:

Reviewer: 1

Comments to the Author(s)

This is an interesting and timely proposal. In broad terms, the case for investigating the relationship between numeracy and COVID attitudes and behavior was compelling to me on both scientific and practical grounds. The methods seemed broadly sound to me, and were described at a good level of detail and with clear language (which is hard to do with these kinds of analyses, so it is a big credit to the authors for that). Below I detail some comments and concerns which I hope will be helpful:

1. My largest concern is about the focus on mediation, which is a complex causal process. The introduction did not elaborate on the theoretical or conceptual case for investigating mediation. Furthermore, the design, a cross-sectional survey, is not an appropriate one for testing hypotheses about mediation (e.g., Bullock et al., 2010). I believe it would be more appropriate to reframe this research as descriptive, and it would still be compelling. To what extent is numeracy associated with COVID attitudes and behavior? And how much is that association about general numeracy versus domain-specific numeracy? I think this reframing would not be very far from the general research questions set up in the introduction, and the research design could remain unchanged. It would lead to some changes in the analytic strategy, but not a complete overhaul.

Author Response:

We agree that a cross-sectional design is not optimal to examine a mediation relationship, as a cross-sectional design presumes that the temporal ordering of the variables in the causal chain is correct. Further, cross-sectional mediation models produce unstable estimates of longitudinal effects, and may over- or underestimate the indirect effect. There is currently no empirical evidence specific to COVID-19 to justify the causal relationships posited in the mediation

analysis. However, we argue that there is extant evidence in the literature that is highly suggestive of the order of the causal chain, as well as the sign of these relationships. In light of this, we believe that the risks of using cross-sectional designs are lessened to some extent (Fairchild & McDaniel, 2017; MacKinnon, 2008; P. Shrout et al., 2011; P. E. Shrout, 2011), and it is reasonable that these prospective results can be taken as preliminary findings that can be further expanded upon in future studies.

For path a, we would point to literature that explores the relationship between basic mathematical/numerical abilities and mathematical achievement. For instance, it has been found that students with stronger basic numerical abilities tend to have stronger performance when applying this knowledge to different contexts (Ballard & Johnson, 2004; Geary, 2011; Ludewig et al., 2019). As the understanding of the numerical facts related to COVID-19 can be considered an instance of application of numerical knowledge, we would expect those with stronger mathematical abilities to have a better grasp of these facts.

For path b, we refer to the health numeracy literature that explores the relationship between applied understanding of health-related numerical facts and health-related outcomes. It has been found that those patients with stronger health numeracy tend to interpret novel health-related information correctly and tend to have better health outcomes (e.g., Brust-Renck et al., 2017; Weinfurt et al., 2003). We would argue that COVID-19 health numeracy, and COVID-19 health-related attitudes and behaviours are specific instances of health numeracy and health outcomes. As such, we would expect those with stronger COVID-19 health numeracy to have better health outcomes.

Finally, we have mentioned in the literature review that few extant studies have examined the direct relationship between basic numeracy and health outcomes. Aside from the potential of an indirect effect as described above, a direct effect of COVID-19 health numeracy on COVID-19 health outcomes may be more tenuous, but not implausible. Those with stronger basic numerical abilities may be more familiar with the underlying mathematical concepts used in COVID-19 health information, and would have a smaller likelihood to fall into common misconceptions perpetuated in the media, intentionally or otherwise. This may cause those with stronger basic numeracy backgrounds to make good initial choices rather than making poor initial choices and correcting those choices afterwards. For example, a person who is not familiar with fractions may initially assess COVID-19 as a smaller threat due to poor representation of COVID-19 rates in the media, whereas a person with such knowledge would question how the media represents the information and have a better assessment of the threat of COVID-19.

We have amended the literature review to go over the reasons why we believe there is a casual relationship between the variables as outlined above (pg 6):

“First, there is a wealth of evidence that suggests that those with stronger basic numeracy are better equipped to apply this knowledge to diverse and novel contexts (Ballard & Johnson, 2004; Geary, 2011; Ludewig et al., 2019). Given that public information about COVID-19

disseminated in the media requires the understanding of numerous mathematical concepts that are typically not encountered in the day-to-day, it is likely that those in the population that possess stronger basic numeracy would have a stronger grasp of the risks relating to the pandemic. Second, there exist many studies on health literacy, which have examined the association between applied mathematical skills, such as health- and non-health-related word problems, and health-related decisions and outcomes. Those who perform better at these problems tend to have better health outcomes in the future (Brust-Renck et al., 2017; Weinfurt et al., 2003). Finally, stronger basic numeracy may have an independent effect on health outcomes controlling for health numeracy. At an equal level of health numeracy, those with stronger basic numeracy may be more capable of understanding the underlying numerical concepts of COVID-19 health information. As such, they would be better equipped in spotting common pitfalls to which those less familiar with the underlying numerical concepts may fall pray.”

Further, we completely agree that the individual paths of the mediation are interesting research questions on their own. As such, we have amended our analysis plan to assess each of these paths in the case where a mediation model is not supported by the data (pg 23):

“Regression Analysis

The current study focuses on the interrelations between the three main variables, basic numeracy, COVID-19 health numeracy, and COVID-19 health-related attitudes and behaviors. Specifically, we hypothesized that COVID-19 health numeracy would be predicted by basic numeracy, that COVID-19 health-related attitudes and behaviors would be predicted by COVID-19 health numeracy, and a possibility that there may be a direct effect between basic numeracy and COVID-19 health-related attitudes and behaviors. These hypotheses, when taken together, constitute a mediation analysis. However, as these hypotheses are interesting research by their own merits, we will evaluate these regression coefficients independently if an overall mediation was not supported by the data.”

2. Figure 1 shows arrows from observed indicators to latent variables, suggesting they are formative. However, the text suggests that the latent variables will be reflective (and I do not believe a model with formative indicators would be identified). Assuming the text is right, the figure should probably be changed.

Author Response: Thank you for pointing out that error. We have amended the diagram to represent reflective latent variables.

3. The statement that English “minimizes potential linguistic influences on numerical processing” does not appear to be supported by the cited reference, which discusses differences among languages but does not present English as having fewer influences. It makes sense that the researchers are interested in studying the countries they are from and the single, most commonly spoken language in them. But it is not necessary to argue that there is some scientific advantage to what seems to me to be more of a practical decision.

Author Response: We have modified the passage in the manuscript to clarify the reason for sampling only English speakers. Although we agree that the English language does not have specifically fewer linguistic influences on number representation, we thought it important to explain why a study across three nations was conducted in only one language, especially given that at least one other language is spoken by a large portion of the population in some of these countries (e.g., French in Canada). In addition to the practical advantage, conducting the study in one language also allows us to control for potential confounds represented by linguistic differences in numerical representation. One such example of this is in how large numbers are described: while French and Spanish languages use long scale naming, English and Arabic language use short scale naming. As described the modified text now appears on p.7:

“These countries were chosen because (1) they are all members of the G7 and thus have comparable economies and political systems, and (2) the response to COVID-19 (e.g., when lockdown and re-opening measures were implemented) as well as the case fatality and testing rates differed between the three countries. Furthermore, the majority language for all three countries is the same (i.e., English), thereby limiting potential confounds driven by linguistic differences in number representation (Dowker & Nuerk, 2016).”

4. I reviewed the materials and had some questions and comments about a few of the items:

- a. COVID numeracy item 11 is: “In which country do you have a higher chance of being tested for COVID-19.” Doesn’t the answer depend on how each country decides who to test? It is possible that the researchers are thinking of random testing, or mean “you” generically, but I am not sure the respondents will too.
- b. COVID numeracy item 12: Would the answer depend on whether testing is done by random sampling vs. purposive (e.g., testing people with symptoms)?
- c. COVID numeracy item 13: Would the answer depends on whether confirmed cases are retested (e.g., to see if they have “cleared” the virus)?

Author Response: We agree with the reviewer that testing protocols and retesting rates would affect these numbers in a real setting, and that countries differ tremendously with respect to testing procedures. However, the purpose of these COVID numeracy items is to assess participants’ abilities to work with absolute and relative numbers. We have therefore added to these items that participants should base their answers solely on the numbers presented in the table, and to assume that testing procedures are the same over Countries A, B and C.

5. As described, the “pilot” study is not a pilot study (Leon et al., 2011), and smaller-N studies are generally not a good basis for a power analysis (Kraemer et al., 2006). I would describe it more as a preliminary psychometric study. The authors should address whether the sample size will be adequate for the planned analyses. (I am not confident that it is, but that’s just a gut response and I may well be wrong.) I also did not understand the sentence, “Third, mean scores

of items for each of the main variables will be used as a rough guide to estimate the intercorrelations of the main variables” on page 14.

Author Response: On account of reviewer 1’s comments and comments made by the other reviewers, we have made substantial changes to the pilot study. Specifically, we have increased the number of participants and will analyze the pilot data to assess whether the data supports the hypothesized latent variables, and to make substantive decisions regarding the measurement model before main data collection starts. We have included the modified section of the manuscript below for your convenience.

We have increased the number of participants recruited for the pilot to be a total of 500. From consulting the CFA literature, we believe 500 would be enough to recover population parameters for the measurement model (Kline, 2015; Wolf et al., 2013).

Further, we have modified the pilot study to include a CFA model (please see decision tree for pilot data, Figure 3, on pg. 18) and we plan to use the intercorrelations between the latent variables in the CFA to inform the power analysis. We do agree that using small sample correlations would be problematic for power analysis purposes. As we have chosen rather conservative values for the power analysis, we will not use the pilot study for purposes of power calculations.

We are not planning to examine between-country differences with the pilot study, however, we are currently planning to collect equal numbers of participants from each country for the pilot study. We would appreciate any opinions on whether we should collect the pilot data from only one country or all three.

As described, the following text now appears beginning on pg 17:

"Analysis Plan

The main hypotheses of the current study are concerned with the interrelations between three hypothesized latent variables: basic numeracy, COVID-19 health numeracy, and COVID-19 health-related attitudes and behaviours. To assess these interrelations, several measures were newly developed for the current study. As a result, there is a large degree of uncertainty regarding whether the data would fit both the hypothesized measurement model (i.e., the relation between the latent factors and their indicators) and the structural model (i.e., the relation between latent factors). Given the unpredictable nature of the measures, it may be necessary to make substantial changes to the hypothesized statistical model post data collection. This is not ideal as it may increase the risk of overfitting the data. To alleviate this risk, we will conduct a pilot study of the variables described above with the primary purpose of assessing item quality and the measurement model. The data from the pilot study will be used to inform substantive decisions regarding the measurement model that will be submitted as part of the stage 1 revision of the registered report before main data collection (see figure 3 and figure 4 for a decision tree for the pilot study and main analysis).

Pilot Study

We will collect data from 167 participants from each of the three countries of interest (totaling 501 participants). The number of participants is in line with recommendations from the confirmatory factor analysis literature (Kline, 2015; Wolf et al., 2013).

First, preprocessing of data will be conducted in the same manner as proposed for the main data set (see **Preprocessing of Data**). Participant exclusion criteria will be revised if the rule removes a large portion of participants. Second, we will compute sample statistics and examine results for signs of ceiling or floor effects or a lack in variability. Any such problem would indicate the need to retool the question to capture the whole variability in the population.

Third, an exploratory factor analysis (EFA) will be conducted on all items. Items that exhibit cross-loading or poor factor loading will be retooled or removed. Further, factors will be examined to ascertain whether the observed data conform with the hypothesized factors. If the factor structure does not reflect the hypothesized factors, substantial changes to the hypotheses would be required. Information regarding factor structure from the EFA will be used to make substantive decisions regarding the number of item parcels and the strategy used to construct item parcels (see below). These decisions will be included in the revisions to stage 1 registered report and will be employed to analyze the data from the main study. Fourth, a confirmatory factor analysis (CFA) model with item parcels will be computed to assess model fit and examined for nuisance parameters. After the estimation of both the EFA and CFA, factor reliability will be assessed using coefficient omega (McDonald, 2013). Convergent validity will be assessed using average variance extracted (>0.5) and discriminant validity will be assessed using the Fornell & Larcker criterion (Fornell & Larcker, 1981; Hair et al., 2016).

It is typically ill-advised to use pilot studies to calculate power, as pilot studies are typically underpowered and yield unreliable estimates (Kraemer et al., 2006). As such, while we could calculate the structural model, we will not adjust power analysis values according to results from the pilot analysis. Similarly, no between country differences would be tested due to a lack of power. Pilot data will not be included in the analysis of the final sample."

6. I do not think outlier removal is necessary for rating-scale responses that have well-defined ranges. It may be better to use robust estimation and inference methods, which I believe are straightforward to implement in Mplus.

Author Response: We agree that for well-defined ranges of rating scale responses, outlier removal may not be necessary. We have amended the section to differentiate between rating-scale responses and non-rating scale responses (particularly the number line questions in the basic numeracy section; p.16). Specifically, for items with rating scales, we will use univariate and multivariate outlier identification methods to potentially flag participants who were careless or did not pay attention, but we will not remove these values unless there is evidence of inattentiveness. However, outlier removal for number line estimation tasks are common as participant inattention or error can heavily skew results. As such, for number line items,

identified outliers will be treated as missing data. We have also amended the manuscript to analyze the data using the maximum likelihood estimator with robust standard errors.

7. On p. 16 the authors write that there may be a problem with spurious correlations. The issue isn't spuriousness – residual correlations may be a real and reliable feature of these measures. Regardless of the terminology, I think parceling is a wise choice here. I encourage the authors to look at Little et al (2013), which cautions against random parceling in many situations. They review pros and cons of other strategies, including the strategy proposed here for subdimensions, as well as alternatives for dealing with subdimensions. (It may be that a domain-representative approach is better than the facet-representative one described here, so I'd encourage the authors to consider whether that's the case.)

Author Response: With regard to the section on spurious correlations, we have amended it to more accurately describe the benefits of parceling. Specifically, we have paraphrased Little's argument that individual items tend to be less reliable and item specific effects may subject the final model to more nuisance parameters. As described the modified text now appears on pg 21:

"The current study employs a large number of items in the measurement of the main variables. Operationalizing latent variables with a large number of indicator items is not ideal from a psychometric perspective. This is because individual items tend to be statistically less reliable than aggregates and have a larger likelihood to have correlated specific effects may subject the final model to nuisance parameters (i.e., correlated residuals; Little et al., 2002)."

We have amended the parceling strategy on your suggestions. As we have items with different response types, a random parcel strategy may not be ideal. In the case where unidimensionality is achieved, we will employ a balancing approach to form 3 parcels (for a decision tree, please refer to Figure 4 on pg. 18).

In the case where unidimensionality is not achieved, we do agree that the domain-representative approach may be advantageous from a modelling perspective, especially if each facet is expected to be related to other constructs beyond the shared variance. However, we believe putting subdimensions into the same facet may be advantageous in this situation when we are not only interested in the between-country differences in the structural model, but also the potential between-country differences in the measurement model. Specifically, with the facet-representative approach, between-country differences in factor loadings and factor intercept can be meaningfully interpreted (with some caveats). For instance, we may see between country-differences in the factor loading differences in mask wearing tendency, which may indicate populations in some countries value wearing masks more.

8. What if the proposed model does not fit? It is very common that structural equation models do not fit well as planned, an issue the authors seem to be anticipating when they discuss residual correlations. What criteria (fit thresholds etc.) will be used to decide if there is good-enough fit to proceed with interpreting the substantive results? And if the fit is not adequate, and the authors proceed to make modifications in the same data they will conduct main analyses in, there is a

risk of overfitting. It may be that the preliminary psychometric study could be increased in sample size and treated like a training dataset, with modifications recorded in an interim preregistration, and then the main study run as if it's a holdout dataset. I recognize that this would increase the cost and resources required and therefore is not a trivial suggestion. So I will add that some concessions to practicality may be appropriate here.

Author Response: We have taken two steps to help alleviate this problem. First, we will use the expanded pilot study to better understand the psychometric properties of the measurement model (please see our response to comment 5). Second, we have expanded and clarified the main analysis to include contingencies if invariance is not achieved. Specifically, we have delineated (1) the course of action if the overall mediation model does not emerge, (2) the course of action if configural invariance is not achieved, and (3) the course of action if scalar or metric invariance is not achieved.

With regard to the criterion for fit threshold, we will use CFI and TLI < 0.90 and RMSEA > 0.08 as the threshold to examine the modification indices for correlated residuals and other sources of misfit.

In light of your comments and the comments from the other reviewers, we have decided on increasing the number of participants in the pilot study to be 500. This increased sample will provide a more robust understanding of the measurement model (based on literature recommendations). Further, on your suggestion, we will make an interim preregistration after the pilot is complete. To decrease the risk of overfitting of the final dataset, we aim use the pilot study to preregister the substantive decisions regarding parceling method, number of parcels, and other modifications to the measurement model. Changes to the measurement model in the final dataset would only occur in the case of major misfits and will be reported as post-hoc changes. We believe this is a reasonable way to guard against overfitting.

9. Will political leaning be analyzed across countries, or separately within each country? "Left" vs. "right" have different meanings, and are associated with different policy positions, in different countries.

Author Response: While there are some cultural variations within each country of the conception the "left" and "right", we believe this to be the most well-tested and general question available that efficiently addresses how individuals self-identify politically. The question is modeled after a very similar question from the World Values Survey (<https://www.worldvaluessurvey.org/wvs.jsp>), a question which has been continually in use since 1980 and was included in the most recent data collection wave (2017-2020). Their direct wording is, "In political matters, people talk of 'the left' and 'the right.' How would you place your views on this scale, generally speaking?". We have added further clarification of the scale by adding "liberal" and "conservative" for reference, in case participants needed an orientation (i.e. "In political matters, people talk of "the left"(e.g., liberal) and "the right"(e.g., conservative). How would you place your views on this scale, generally speaking?"). Below is a snapshot of the breakdown from the 2005-2009 data collection wave using their online analysis tool, where

Canada, the US, and the UK were all surveyed. Given that these three countries are more similar to each other than the much broader set of countries in the full World Values Survey sample, we hope that the question as constructed represents a decent compromise between allowing flexibility for surveying across countries and obtaining sufficiently specific information about political self-identification.

	TOTAL	Country/region		
		Canada	United Kingdom	United States
Left	2.3%	2.4%	2.8%	1.7%
2	2.0%	2.1%	2.4%	1.5%
3	6.7%	7.1%	7.1%	5.7%
4	7.6%	7.5%	7.6%	7.7%
5	31.0%	27.9%	33.7%	34.0%
6	13.5%	10.7%	12.5%	19.2%
7	9.2%	9.1%	7.8%	10.7%
8	7.9%	8.8%	5.4%	8.3%
9	2.7%	2.5%	2.2%	3.5%
Right	2.4%	1.7%	2.3%	3.5%
Missing; Not asked by the interviewer	0.9%	-	1.4%	2.1%
No answer	1.3%	0.2%	2.9%	2.0%
Don't know	12.5%	20.1%	11.8%	-
(N)	(4,454)	(2,164)	(1,041)	(1,249)
Mean	5.49	5.45	5.29	5.70
Std Dev.	1.85	1.89	1.84	1.79
Base mean	(3,796)	(1,725)	(874)	(1,197)

Selected samples: Canada 2006, Great Britain 2005, United States 2006

Question wording

In political matters, people talk of "the left" and "the right."
How would you place your views on this scale, generally speaking?

Equivalences in other waves

1981/1984: **V123**
1990/1994: **V248**
1995/1999: **V123**
2000/2004: **V139**
2005/2009: **V114**
2010/2014: **V95**
2017/2020: **Q240**

Additional translations --Sel

To underscore the fact that a version of this question has been used successfully in all three countries in the current study, we have added the following sentence on pg. 9:

"The original political leaning question has been used on international surveys dating back to 1980, including surveys of Canada, the US, and the UK."

Reference:

Inglehart, R., C. Haerper, A. Moreno, C. Welzel, K. Kizilova, J. Diez-Medrano, M. Lagos, P. Norris, E. Ponarin & B. Puranen et al. (eds.). 2014. World Values Survey: All Rounds - Country-Pooled Datafile Version:
<https://www.worldvaluessurvey.org/WVSDocumentationWVL.jsp>. Madrid: JD Systems Institute.

10. Finally, a general comment. There are several places where the authors have left room for human judgment during the data analyses. For example they discuss running exploratory factor analyses to make decisions about parceling. And they will have to make decisions about whether measurement invariance is established, and when it is not, what to do about it. (The analyses plan here is underspecified.) I have preregistered SEMs before and I'm currently in the middle of a registered report of a measurement invariance study, which has given me a deep appreciation for how hard it is to preregister a complete decision tree in this kind of work. I think some room for researcher degrees of freedom is unavoidable in this kind of work and should not be an impediment to its acceptance as a registered report. A training-holdout plan as I described earlier might address some of these other issues as well. Alternatively, the authors may consider whether some kind of blinded analysis strategy (MacCoun & Perlmutter, 2017) could be implemented here to shield those decisions from any potential biasing influence on the substantive results.

Author Response: To reduce the amount of potential bias, we have revised the pilot study so as to preregister the parameters for the measurement model before data collection commences. Further, we have revised the section regarding measurement invariance to include what we would do if measurement invariance is not established. To increase transparency, we have also included a decision tree for the various parts of the analysis. We hope these steps would help alleviate the risk of overfitting the data as well as potential "researcher degrees of freedom".

Sanjay Srivastava (This is a signed review.)

References

- Bullock, J. G., Green, D. P., & Ha, S. E. (2010). Yes, but what's the mechanism?(don't expect an easy answer). *Journal of personality and social psychology*, 98(4), 550.
- Kraemer, H. C., Mintz, J., Noda, A., Tinklenberg, J., & Yesavage, J. A. (2006). Caution regarding the use of pilot studies to guide power calculations for study proposals. *Archives of general psychiatry*, 63(5), 484-489.
- Leon, A. C., Davis, L. L., & Kraemer, H. C. (2011). The role and interpretation of pilot studies in clinical research. *Journal of psychiatric research*, 45(5), 626-629.
- Little, T. D., Rhemtulla, M., Gibson, K., & Schoemann, A. M. (2013). Why the items versus parcels controversy needn't be one. *Psychological methods*, 18(3), 285.
- MacCoun, R. J., & Perlmutter, S. (2017). Blind analysis as a correction for confirmatory bias in physics and in psychology. *Psychological science under scrutiny: Recent challenges and proposed solutions*, 297-322.

Reviewer: 2

Comments to the Author(s)

The Authors are planning to conduct a large survey across three countries (UK, USA and Canada) to test the relation between basic numeracy, health numeracy (about the COVID-19 pandemic) and self-reported attitudes/behaviours in relation to health (with reference to the COVID-19 pandemic). Basic numeracy is measured with tasks assessing number magnitude processing (analogue number line tasks) and conceptual knowledge of fractions and decimals. Health numeracy is measured with a (very nice) series of multiple-choice questions on graph literacy, proportions and odd ratios. Health-related attitudes/behaviours are measured with a series of questions measuring compliance and the perception of impact and severity of the pandemic. A positive relation is expected between basic numeracy and health-related attitudes/behaviours (whereby higher numeracy originates more compliant attitudes/behaviours), mediated by health numeracy. Differences between countries will be analysed and potential moderating variables are included in the design.

1. This is a well-presented and thorough proposal that aims to address an interesting research question. However, in its current version it includes a plan for a pilot that could perhaps have been conducted before submission and there are a few unclear points. The most limiting aspect of the current design is that the mediation analysis may show a relation between the latent constructs without controlling for general ability.

Author Response: We now plan to conduct the pilot study in advance of the final Stage 1 approval, but after receiving feedback on the overall study design. Accordingly, we have indeed made substantial changes to the pilot study based on reviewer feedback that will strengthen the overall study design. In particular, we will use the preliminary data to confirm the measurement model of the CFA (please see below).

Additionally, we agree that general ability would be a limitation of the current study. And, while we do measure some proxies related to general achievement (e.g. years of education), a full measurement of general ability via an IQ measure is not feasible for the current study due to time constraints and the nature of the survey. Also, in response to the current reviewer's comment #6 (See below), we have added a calibration item that we may use to control for general motor ability. In the end, there are some factors related to general ability that we would have to accept as a limitation to the current study.

2. Abstract. It is clear and concise, and mostly focuses on the primary goal of the study - apart from the final statement. Why are political views and anxiety tested? This is not the strongest component of the study and the variables (one measured with a single item) are included for exploratory purposes. I think that only if robust evidence of moderation effects was found after data collection they could deserve a mention here.

Author Response: Thank you for pointing that out. We agree and have removed reference to these exploratory moderators from the abstract.

3. Participants. More information is needed about these prospective market research participants and whether Qualtrics panels are representative of the general population in the three countries of interest. Could they live in the same household and share the same information channels, behavioural changes etc.? What compensation is offered?

Author Response: Qualtrics has provided us with documentation relevant to this question which can be found here: <https://osf.io/evnf2/>. According to the Qualtrics representative we have been working with with the goal is: "...to have representation from all ages, genders, incomes, education and regions".

With respect to the question about same households, we are recording the IP of the survey respondents and will filter the data to remove participants with the same IP. Therefore, we will reduce the risk of a violation of independence due to participants living in the same household

We should also point out that our survey contains detailed demographic questions which will allow us to: a.) verify the representativeness of the sample and b.) to describe the demographics of the same in great detail.

With respect to compensation, Qualtrics state that: "Our panelists join from a variety of sources. They may be airline customers who chose to join in reward for SkyMiles, retail customers who opted in to get points at their favorite retail outlet, or general consumers who participate for cash or gift cards, etc. When participants are invited to take a survey, they are informed what they will be compensated."

4. Procedure and materials. What information will be available about screen size and how will this be taken into account? At the end of the survey, participants are asked which type of device they are using but screen sizes may vary greatly even within the same type of device. Further, across devices there may be a systematic variation of response modality (mouse vs. touchpad vs. pen vs. finger).

Author Response: This is a valid argument, but probably relevant for one of the numeracy measures, the number line estimation tasks. Therefore, we added three additional questions to the survey. The first hidden question collects meta information including Browser Type, Browser Version, Operating System, Screen Resolution, Flash Version, Java Support, and User Agent. The second question was "How did you respond to the survey?", which was added to the end questions of the survey. It has multiple choices including 1) By mouse, 2) By touchpad, 3) By pen on touch screen, 4) By finger on touch screen, 5) Other (please specify below): [blank]. The third question was about eye-hand coordination, as the reviewer mentioned in comment #6. The following text now appears on pg 8: "The survey will end with five final questions to check: 1) eye-hand coordination (three slider items in which participants are asked to move the slider to

number 2, 5, and 7 on a number line from 1 to 10. The difference with the numeracy number lines is that the final positions have been specified on top of the lines and do not need participants' estimations), 2) the device that they used (i.e., desktop computer, laptop, tablet, smartphone, other), 3) the response modality (i.e., mouse, touchpad, pen on touch screen, finger on touch screen, other), 4) response seriousness (where participants will be asked whether they would keep their data if they were the experimenter), and 5) any technical problem they may have been faced with. An additional hidden question collects meta information including Browser Type, Browser Version, Operating System, Screen Resolution, Flash Version, Java Support, and User Agent."

Following the suggestion by the reviewers, we will check the possible influence of screen resolution, response modality, eye-hand coordination, and the device in our pilot data. In case of any systematically significant influence of these factors on participants' responses, we will control it in our analysis models. Please note that screen resolution would be more informative than screen size in our study. The reason is that even if two responders use the same screen size, it is the screen resolution which plays the main role in accuracy of the responses in the number line estimation tasks. This is the only place for a potentially critical role of screen size/resolution in our survey.

The following text now appears on pg 10: "Finally, we will check the possible influence of screen resolution, response modality, eye-hand coordination, and the device in our pilot data. If any of these variables revealed a systematically significant influence on participants' responses, we will control it in our analysis models."

5. Power analysis. The Authors provide a power analysis for their mediation model, multigroup and moderation analyses and choose their sample size accordingly. What are the α -levels, and should correction for multiple testing be applied? Have missing data been taken into account? These may increase the sample size requirements by a rather large amount. In the mediation analysis, how likely is it that the size of the $X > Y$ effect is the same as that of $X > M$ and $M > Y$ effects? Given that several of the measures included in this study have been created ad hoc, planning a pilot study before conducting the larger study is very appropriate. I wonder, however, whether the pilot study should not have been conducted before submitting an IPA request for the larger study. Depending on the results of the pilot study, changes to the power analysis and to the study materials may be opportune (i.e. these would not be in definitive form at the time of IPA).

Author Response: The alpha level that was used in the power analysis was 0.05. We have revised the manuscript to make it clear. We have not taken into account missing data in the monte carlo generation, and we have not adjusted for multiple comparisons either.

On your suggestion, we have attempted to redo the monte carlo power analysis with missing data. However, we have concluded that it would not be feasible on practical grounds. The specification of random slopes (in the moderated mediation models) in conjunction with missing data would require monte carlo integration. Calculations of 10,000 replications for multiple sample sizes would make computation time prohibitively high. Similar concerns arise when

simulating multigroup analysis. However, specifying 10% missing data on dependent variables for the simple mediation case only lowered power by about 2-3%.

The sizes of the paths were determined to be the minimum size that we would be able to detect at the power level that we specified. It is likely that the sizes of these paths would be different, and likely higher, in reality.

On your suggestion and the suggestions of other reviewers, we have made extensive changes to the pilot study. Specifically, we have increased the sample size to be 500. This will be a large enough sample for us to have a good idea of the psychometric properties of the main variables. Given the newly proposed plan for piloting and then submitting for an stage 1 IPA thereafter, we can adjust the survey according to the new information about its psychometric properties. For your convenience, we have copied the pilot study section here.

With regard to using the pilot data for power calculations, Reviewer 1 has pointed out that using the pilot data to make changes to the power analysis would not be ideal, as the pilot data is underpowered and may not recover population parameters. As we have chosen rather conservative values for the power analysis, we will not use the pilot study for purposes of power calculations.

We are not planning to examine between-country differences with the pilot study, however, we are currently planning to collect equal numbers of participants from each country for the pilot study. We would appreciate any opinions on whether we should collect the pilot data from only one country or all three.

As described, the following text now appears beginning on pg 17:

"Analysis Plan

The main hypotheses of the current study are concerned with the interrelations between three hypothesized latent variables: basic numeracy, COVID-19 health numeracy, and COVID-19 health-related attitudes and behaviours. To assess these interrelations, several measures were newly developed for the current study. As a result, there is a large degree of uncertainty regarding whether the data would fit both the hypothesized measurement model (i.e., the relation between the latent factors and their indicators) and the structural model (i.e., the relation between latent factors). Given the unpredictable nature of the measures, it may be necessary to make substantial changes to the hypothesized statistical model post data collection. This is not ideal as it may increase the risk of overfitting the data. To alleviate this risk, we will conduct a pilot study of the variables described above with the primary purpose of assessing item quality and the measurement model. The data from the pilot study will be used to inform substantive decisions regarding the measurement model that will be submitted as part of the stage 1 revision of the registered report before main data collection (see figure 3 and figure 4 for a decision tree for the pilot study and main analysis).

Pilot Study

We will collect data from 167 participants from each of the three countries of interest (totaling 501 participants). The number of participants is in line with recommendations from the confirmatory factor analysis literature (Kline, 2015; Wolf et al., 2013).

First, preprocessing of data will be conducted in the same manner as proposed for the main data set (see **Preprocessing of Data**). Participant exclusion criteria will be revised if the rule removes a large portion of participants. Second, we will compute sample statistics and examine results for signs of ceiling or floor effects or a lack in variability. Any such problem would indicate the need to retool the question to capture the whole variability in the population.

Third, an exploratory factor analysis (EFA) will be conducted on all items. Items that exhibit cross-loading or poor factor loading will be retooled or removed. Further, factors will be examined to ascertain whether the observed data conform with the hypothesized factors. If the factor structure does not reflect the hypothesized factors, substantial changes to the hypotheses would be required. Information regarding factor structure from the EFA will be used to make substantive decisions regarding the number of item parcels and the strategy used to construct item parcels (see below). These decisions will be included in the revisions to stage 1 registered report and will be employed to analyze the data from the main study. Fourth, a confirmatory factor analysis (CFA) model with item parcels will be computed to assess model fit and examined for nuisance parameters. After the estimation of both the EFA and CFA, factor reliability will be assessed using coefficient omega (McDonald, 2013). Convergent validity will be assessed using average variance extracted (>0.5) and discriminant validity will be assessed using the Fornell & Larcker criterion (Fornell & Larcker, 1981; Hair et al., 2016).

It is typically ill-advised to use pilot studies to calculate power, as pilot studies are typically underpowered and yield unreliable estimates (Kraemer et al., 2006). As such, while we could calculate the structural model, we will not adjust power analysis values according to results from the pilot analysis. Similarly, no between country differences would be tested due to a lack of power. Pilot data will not be included in the analysis of the final sample."

6. Number magnitude processing. More than any other tests, the number line tasks and their derived PAE indices seem to depend heavily on hand-eye coordination/tool precision (e.g. pen, mouse, finger, touchpad), in addition to basic numeracy skills. Could individual scores in the number line tasks be confounded by response means? Inclusion of a few calibration items may be useful.

Author Response: This is a good point. As we mentioned in response to comment #4, we will check the tool precision in our pilot data and will make the decision based on that (please see our response above).

For the eye-hand coordination check, we added a slider item with three questions to the end questions. These questions are very similar to the number estimation tasks, in which we ask the responders to move the slider to move the slider to 2, 5, and 7 in a 1-10 number line, while the

exact positions have been shown on a grid space. Similar to the other three possible confounds that we explained above (i.e., screen resolution, device, response modality), we will check the responders' mean PAE in these three questions to see whether they are correlated with the PAE in the number line estimation tasks. If the association is significant, we will add them to the analysis model as an extra covariate.

The following text now appears on pg 8: “The survey will end with five final questions to check: 1) eye-hand coordination (three slider items in which participants are asked to move the slider to number 2, 5, and 7 on a number line from 1 to 10. The difference with the numeracy number lines is that the final positions have been specified on top of the lines and do not need participants' estimations), 2) the device that they used (i.e., desktop computer, laptop, tablet, smartphone, other), 3) the response modality (i.e., mouse, touchpad, pen on touch screen, finger on touch screen, other), 4) response seriousness (where participants will be asked whether they would keep their data if they were the experimenter), and 5) any technical problem they may have been faced with. An additional hidden question collects meta information including Browser Type, Browser Version, Operating System, Screen Resolution, Flash Version, Java Support, and User Agent.”

We additionally added another text on pg 10: “Finally, we will check the possible influence of screen resolution, response modality, eye-hand coordination, and the device in our pilot data. If any of these variables revealed a systematically significant influence on participants' responses, we will control it in our analysis models.”

7. Demographics and other potential moderators. Please define SES before using the acronym.

Author Response: Thank you, we have amended the manuscript (pg. 9).

8. Ending questions. Typo: Devince  device. I am not convinced that somebody who has not been responding seriously that far will be reliable at the “seriousness check”.

Author Response: Thank you for pointing out this typo. It has been fixed. In regards to the seriousness check, previous literature of online research (e.g., Aust, et al., 2013; Bayram, 2018; Reips, 2000) strongly recommends that a seriousness check of responders is a great tool to improve data quality in online studies. We have observed this improvement of data quality in our recent studies in numerical cognition (Cipora, Soltanlou, Reips, Nuerk, 2019; Huber, Nuerk, Reips, Soltanlou, 2019). Please note that we also explicitly mentioned that “Your response will not influence compensation” in this question. Therefore, we are convinced that while only about 1-2% of participants might be discarded based on their response to this question, taking this extra step would lead to a better data quality in the current study. Still, we agree that this seriousness check will not work for all participants and so it serves as just one of several checks (e.g. outlier analysis) that we will use to assure the quality of our data.

Reference:

- Aust, F., Diedenhofen, B., Ullrich, S., & Musch, J. (2013). Seriousness checks are useful to improve data validity in online research. *Behavior Research Methods*, 45, 527–535. doi.org/10.3758/s13428-012-0265-2
- Bayram, A. B. (2018). Serious subjects: A test of the seriousness technique to increase participant motivation in political science experiments. *Research & Politics*, 5(2), 2053168018767453. doi.org/10.1177/2053168018767453
- Cipora, K., Soltanlou, M., Reips, U.-D., Nuerk, H.-C. (2019). The SNARC and MARC effects measured online: Large-scale assessment methods in flexible cognitive effects. *Behavioral Research Methods*, 1-17. doi:10.3758/s13428-019-01213-5
- Huber, S., Nuerk, H. C., Reips, U. D., Soltanlou, M. (2019). Individual differences influence two-digit number processing, but not their analog magnitude processing: a large-scale online study. *Psychological Research*, 83(7), 1444-1464. doi:10.1007/s00426-017-0964-5
- Reips, U.-D. (2000). The web experiment method: Advantages, disadvantages, and solutions. In M. H. Birnbaum (Ed.), *Psychological experiments on the Internet* (pp. 89–118). San Diego: Academic Press. doi.org/10.5167/uzh-19760

Reviewer: 3

Comments to the Author(s)

The current paper plans to test a mediational hypothesis linking basic numeracy with health attitudes and behavior through COVID-19 health numeracy. There are a number of strengths in the proposal, along with some important weaknesses. On the strength side, the team investigates an interesting question using a rigorous analytic plan, including extensive measurement invariance testing. I examined the materials on OSF, and things appear to be well organized (although there is not much there yet, beyond copies of the instruments). Key weaknesses and issues to address are outlined below:

1. Power analyses: The authors use Mplus to simulate a set of power analyses for their model. In the majority of analyses, the conclusion points to a sample size of 675 for each of their three planned countries. This is peculiar to see such consistency across several different analytic strategies. There are at least two possible explanations. First, the authors may have done a sensitivity analysis, working backward from sample size and desired power to effect size. There is nothing wrong with such an approach, but if this is the technique that was used, the paper should be explicit about it. The effect sizes vary across the different specifications while sample size stays constant, so this may be what was done. On the other hand, this consistency could point to an error in the analyses (perhaps some kind of limit in Mplus or some computational limit). 675 per country does seem lower than I would have expected, but the authors do not plan to model any of their variables at the country level (nor could they, with only three nations), so this might explain the final number. I would have looked at the power simulation code, but it was not provided on OSF so I could not. I recommend sharing that code at some point. Regarding expectations for power in the multi-site context, the authors might consult Arend and Schäfer (2019, *Psych Methods*). The multi-level approach that they discuss is complementary to the multi-group SEM approach employed here.

Author Response: One of the reasons that we have reported largely consistent sample sizes across multiple monte carlo simulations is that some of the analyses (for example invariance testing) were computationally intensive. As such we opted not to find exact values for 95% power. Still, in many cases, power was higher than 95%. Most of the values chosen were done so either because we thought it was reasonable or because it is the smallest effect size of interest to the study. The only value that we revised due to feasibility was the minimum regression difference to detect between countries, which was set to 0.2. We have included how we selected the values in the power analysis in the manuscript (p.13). Further, we have uploaded the power analysis files on OSF (https://osf.io/k4iy9/?view_only=08c5b15057694297ba037c94ee41c50a).

2. There is a bit of a mismatch between what the authors say they want to do in their introduction, and what they can do with just three countries. For instance, they state that they will be “investigating whether country-level factors in these three countries influence the mediation model.” What they can do with their multi-group approach is look at whether model loadings, intercepts, variances/covariances (paths), and latent means differ between countries. They cannot investigate country level variables as moderators. They do not actually propose to do so in the planned analyses (or design table), but some of the introduction reads as though this is what they want to do. They can look at individual level moderators (as they propose to do using their ‘additional analyses.’ The moderators just cannot be at the level of the country. Ultimately, if they wanted to do such tests, they would be very underpowered, because power would stem in a large way from the number of countries in the test.

Author Response: Thank you for pointing that out. As you say, our analysis plans are mainly concerned with testing the between-country differences in the measurement and structural model and we do not have any country-level variables. We have amended the introduction to be more in line with the analysis plan p.7:

“Investigating the between-country differences in the strength of the mediation relationship and between-country differences in how covariates may influence this relationship will allow us to comment on whether and how the strength of the relations among the variables differ across countries, which can inform short-term government policies related to COVID-19. ”

3. Concerns about measurement: The authors have an extensive plan to investigate the measurement properties of their variables. This is good, but ultimately, I am concerned about whether it is actually possible to do all of this in one project. The basic numeracy variable is a composite of pieces of existing measures. As such, it probably has the highest chance of successfully approximating a unidimensional factor. The COVID-19 health numeracy variable is constructed ad hoc, but modeled after existing measures, and the attitude/behavior measures appear to be constructed entirely ad hoc. The attitude/behavior measure, in particular, is unlikely to approximate a unidimensional factor (just based on my experience with these things – you almost certainly aren’t going to get it right straight out of the chute). The authors plan to collect 200 independent cases as a pilot study to check the functioning of their measures, but

it's not clear that this will be enough data to do a decent job at this, nor is it apparent that we'd expect a generalizable solution to be born out of this pilot data.

The authors discuss plans to investigate factor structure, but do not mention reliability and validity. Validity of the ad hoc measures appears to be assumed. Reliability could and should be tested. Furthermore, a backup plan is needed should reliability prove insufficient. With dichotomously scored items, reliability is often quite low.

Author Response: Based on your comments and the comments of the other reviewers, we have decided to increase the number of participants to 500 in the pilot study. The main goal of the pilot study is to verify the measurement property of the three main variables of interest. We have copied the revised section below for your convenience.

It is a fair concern that the latent variables that we have hypothesized may not be unidimensional. Given the number of participants in the pilot dataset, we can be reasonably certain to have a good idea of the measurement model (Kline, 2015; Wolf et al., 2013). With the pilot study, we can hopefully make any substantial changes to the hypotheses and methods before data collection starts. This has the added benefit that we will also preregister the parceling technique and number of parcels before data collection starts, which would reduce the risk of overfitting and reduce researcher "degrees of freedom".

We agree that reliability and validity should be tested. To this end, we will compute coefficient omega to assess reliability, and assess convergent and discriminant validity of the EFA and CFA model for both the pilot and main analysis.

We are not planning to examine between country differences with the pilot study, however, we are currently planning to collect equal numbers of participants from each country for the pilot study. We would appreciate any opinions on whether we should collect the pilot data from only one country or all three.

As described, the following text now appears beginning on pg 17:

"Analysis Plan

The main hypotheses of the current study are concerned with the interrelations between three hypothesized latent variables: basic numeracy, COVID-19 health numeracy, and COVID-19 health-related attitudes and behaviours. To assess these interrelations, several measures were newly developed for the current study. As a result, there is a large degree of uncertainty regarding whether the data would fit both the hypothesized measurement model (i.e., the relation between the latent factors and their indicators) and the structural model (i.e., the relation between latent factors). Given the unpredictable nature of the measures, it may be necessary to make substantial changes to the hypothesized statistical model post data collection. This is not ideal as it may increase the risk of overfitting the data. To alleviate this risk, we will conduct a pilot study of the variables described above with the primary purpose of assessing item quality and the

measurement model. The data from the pilot study will be used to inform substantive decisions regarding the measurement model that will be submitted as part of the stage 1 revision of the registered report before main data collection (see figure 3 and figure 4 for a decision tree for the pilot study and main analysis).

Pilot Study

We will collect data from 167 participants from each of the three countries of interest (totaling 501 participants). The number of participants is in line with recommendations from the confirmatory factor analysis literature (Kline, 2015; Wolf et al., 2013).

First, preprocessing of data will be conducted in the same manner as proposed for the main data set (see **Preprocessing of Data**). Participant exclusion criteria will be revised if the rule removes a large portion of participants. Second, we will compute sample statistics and examine results for signs of ceiling or floor effects or a lack in variability. Any such problem would indicate the need to retool the question to capture the whole variability in the population.

Third, an exploratory factor analysis (EFA) will be conducted on all items. Items that exhibit cross-loading or poor factor loading will be retooled or removed. Further, factors will be examined to ascertain whether the observed data conform with the hypothesized factors. If the factor structure does not reflect the hypothesized factors, substantial changes to the hypotheses would be required. Information regarding factor structure from the EFA will be used to make substantive decisions regarding the number of item parcels and the strategy used to construct item parcels (see below). These decisions will be included in the revisions to stage 1 registered report and will be employed to analyze the data from the main study. Fourth, a confirmatory factor analysis (CFA) model with item parcels will be computed to assess model fit and examined for nuisance parameters. After the estimation of both the EFA and CFA, factor reliability will be assessed using coefficient omega (McDonald, 2013). Convergent validity will be assessed using average variance extracted (>0.5) and discriminant validity will be assessed using the Fornell & Larcker criterion (Fornell & Larcker, 1981; Hair et al., 2016).

It is typically ill-advised to use pilot studies to calculate power, as pilot studies are typically underpowered and yield unreliable estimates (Kraemer et al., 2006). As such, while we could calculate the structural model, we will not adjust power analysis values according to results from the pilot analysis. Similarly, no between country differences would be tested due to a lack of power. Pilot data will not be included in the analysis of the final sample."

4. Role of time: The health situation with the virus is unfolding rapidly and, as you note, on different time scales in the different countries under investigation. In what way do you plan to account for time, if at all? I would imagine the numeracy constructs are thought to be relatively temporally stable, but the DV (attitudes and behaviors) might be changing at a country level or local geographic level over time. For instance, if local rules and regulations change, increased compliance with mask wearing (for example) might be expected, but these changes would have nothing to do with numeracy. Likewise, if the virus spreads, objective risk goes up, which likely

has a fairly direct link to attitudes/behaviors. If my country has basically eradicated the virus (haha, I wish), I will act differently than if it is widespread (even if I'm not particularly numerate). The correlation between numeracy and attitudes/behaviors might vary in strength over time, perhaps based on some unknowable combination of these national/local behavior shifts. There could also be issues with ceiling/floor effects that vary over time (which would again affect the strength of correlations). For example, if masks are the law and no mask carries a steep fine, you'll have little variation in mask wearing.

Author Response: It is a fair point that while basic numeracy and COVID-19 health numeracy are fairly stable, attitudes and behaviors are probably more dynamic. As you have pointed out, it is indeed true that behaviors would track with the prevalence of the disease. With qualtrics, we will be able to collect the bulk of the data within a short timespan of a few days, as such, we will be able to reduce the within-country differences in attitudes and behaviors that are due to prevalence and other national/local factors. We could further potentially reduce some of the risk of between-country comparisons by including the prevalence rate as a covariate to attitudes and behaviors. However, as you have pointed out, controlling for these national/local behavior shifts would be trying. This is certainly a limitation of the current study that cannot be easily addressed.

5. There are several examples throughout the paper where sensible plans are described, but details are lacking. For instance, you say that device information (phone vs. computer) will be taken into account in the model, but you don't say exactly how. In general, in a registered report, it is fairly important to have details like this locked down to the extent possible.

Author Response: We have added more detail regarding how we will include covariates. Specifically, we will regress the dependent variables, COVID-19 understanding and COVID-19 behaviours and attitudes, onto the covariates once we have established a measurement model.

Following the suggestion by you and reviewer 2, we will check the possible influence of screen resolution, response modality, eye-hand coordination, and the device in our pilot data. In case of any systematically significant influence of these three factors on participants' responses, we will control it in our analysis models. Please note the only place for a potential confounding effect of these variables might be the accuracy of the responses in the number line estimation tasks in our survey.

The following text now appears on pg 10: "Finally, we will check the possible influence of screen resolution, response modality, eye-hand coordination, and the device in our pilot data. If any of these variables revealed a systematically significant influence on participants' responses, we will control it in our analysis models."

6. The authors state that all cases will be analyzed using FIML, even if there is some limited missing data. This is a good choice and appropriate. What is missing is consideration of how exclusions (for data quality or for high levels of missing data) will play into the obtained sample size. I recommend that the authors consider collecting extra data (beyond 675 per country) to ensure that the final sample size will be above the target. Just make sure to specify the stopping

rule in the next draft (e.g., We will sample 10% extra cases – 745 per country – to allow for exclusions and attrition. All available cases will be analyzed.)

Author Response: We do agree that we should have 675 complete cases. On account of costs of recruitment, we believe we could first use the current exclusion criterion we have on the pilot data to get an approximate percentage of participants that would be excluded. Then we can make a more informed choice in the number of participants needed to reach 675 complete cases. We will include this information after we have completed the pilot study as part of stage 1 registered report revisions. Assuredly, we will collect enough data so as to reach 675 participants after exclusions.

7. You propose to do EFA then CFA on the same data, but this is not ideal. You might be able to do an EFA in your pilot data and then a CFA in the main data. I would recommend constructing parcels randomly regardless. How many parcels do you plan to specify per factor?

Author Response: We agree that EFA and CFA on the same dataset may be problematic. On your recommendation, we have amended the section on the pilot study. We will perform an EFA to inform parceling strategy and CFA with parcels to assess the measurement model on the pilot data. The final model with the main data will only be used for a CFA (see our decision trees in figure 3 and 4 in the manuscript for pilot and final analyses respectively).

With regard to parceling strategy, we have adopted the suggestions made by reviewer 1. We have decided that due to the variety of question types we have, random parceling may not be appropriate, as random parcels assume that questions are interchangeable. Instead, we opted to employ a balancing parcel approach if unidimensionality is achieved and use a facet-representative approach if we observe subdimensions in a construct.

We believe putting subdimensions into the same facet may be advantageous in this situation when we are not only interested in the between-country differences in the structural model, but also the potential between-country differences in the measurement model. Specifically, with the facet-representative approach, between-country differences in factor loadings and factor intercept can be meaningfully interpreted (at least as preliminary results). For instance, if we see between country-differences in the factor loading differences in mask wearing tendency, it may indicate populations in some countries value wearing masks more.

The number of parcels that we will use will ultimately depend on the EFA that we will run with the pilot data. We will submit the parceling strategy, number of parcels, and indicator item as part of the stage 1 registered report revisions.

8. You mention that you will use the “standard procedures” for measurement invariance. I would cite a specific source on this topic (perhaps the Kline SEM textbook, if that is the approach you will follow). Likewise, you mention nested model tests and chi-square comparisons, but do not mention any other fit statistics. It is typical in this area to report CFI and RMSEA, at a minimum, and authors often interpret change in CFI and RMSEA between steps in the model. If you plan

to do this, you should name your cutoffs and cite your source. Finally, you might consider interpreting effect sizes for invariance alongside your significance tests (see Nye & Drasgow, 2011 and their metric dMACs).

Author Response: Those are great points. We have clarified the manuscript to include the source of procedure that we will follow to assess measurement invariance. We have also included sources for changes in fit of CFI and RMSEA as a test for invariance. Finally, we have included the use of dMACs when invariance between items is detected.

As described the section is as follows (pg 23, also see figure 3 and figure 4 on pg 18-19):

“Multigroup Path Analysis

In addition to the overall mediation model, we will also examine the moderated mediation of the categorical variable country. We will employ multigroup path analysis to investigate differences across countries in the means and variation of the main variables in the model (basic numeracy, COVID-19 health numeracy, and COVID-19 health-related attitudes and behaviors)- This will include multiple-group measurement invariance analyses of the individual latent measures. This procedure is usually performed as a step to establish factorial validity across groups. Once established, it becomes possible to evaluate differences in latent means, variances, covariances, or specific regression coefficients including those assessing mediation and moderation across different countries.

We will use the standard procedures in the confirmatory factor analytic literature for evaluating measurement invariance across countries (Kline, 2015). In short, this consists of a number of steps testing the invariance of different parts of the factor/latent variable structure. These include (1) configural invariance - the overall factor structure (i.e., the number of factors and their indicator variables), (2) metric invariance - the loadings of the indicator variables, and (3) scalar invariance - the intercepts of the indicator variables—(Putnick & Bornstein, 2016). Strict measurement invariance is not required for the following analyses and will not be tested. These are tested in incremental stages, and the first three stages are essential to proceed with tests of latent means. Configural invariance will be assessed using model fit, where CFI and TLI > 0.90 and RMSEA < 0.08 is generally accepted as adequate model fit (Hu & Bentler, 1999). Metric and scalar invariance are evaluated using the chi-square difference test, the degradation of fit in CFI and RMSEA. Specifically, we will use the criterion of $\Delta RMSEA < .015$ and $\Delta CFI < -0.01$ (Chen, 2007; Cheung & Rensvold, 2002).

In the case where measurement invariance is not achieved, we will proceed with partial-invariance tests and identify the source of non-invariance. Some degree of researcher judgement would be required to gauge the severity of the invariance between countries. In the event that we do not meet configural invariance, modification indices will be examined for the source of the misfit. If the degree of misfit is small, the removal of the problematic indicator item could be a potential solution. This may necessitate the examination of between-country differences in the individual items constituting the parcel to find problematic items. If the degree of invariance is high, it may be necessary to estimate separate models for each country. In the event that we do not meet

metric or scalar invariance in a given country, invariant items will be identified using the forward method (Jung & Yoon, 2016), effect size of invariance (d_{MACS}) will be calculated to gauge the magnitude of invariance (Nye & Drasgow, 2011). Identified invariant items will be allowed to vary across countries. We will proceed with partial scalar or metric invariance if less than half of the indicators are not invariance (Steenkamp & Baumgartner, 1998; Vandenberg & Lance, 2000). Lack of invariance in the factor structure is often seen as a nuisance, but it can also advance our understanding of cross-cultural differences in how constructs are conceptualized. For example, if results indicate that the sole source of non-invariance in the latent variable COVID-19 health-related attitudes and behaviors is the factor loading of mask-wearing frequency, it can be interpreted that the amount of variance of mask wearing frequency that COVID-19 health-related attitudes and behaviors can explain is different between countries.

Once metric or scalar invariance or partial metric or scalar invariance is achieved, latent variance and path coefficients will be examined, and equality in latent variance and covariance will be tested. In the case where scalar invariance, latent variance equality and path equality is achieved (Vandenberg & Lance, 2000), latent mean differences between countries will be examined.”

9. I don't believe latent mean differences are discussed in the text (I only see them in the table). You'll need scalar invariance, then latent variance and covariance (i.e., path) equality before testing differences in latent means. A lot of hurdles to clear before you can test this.

Author Response: We have amended the section to also address latent mean differences. Specifically, we have specified that scalar invariance, latent variance and covariance equality would need to first be tested before latent mean differences (please see the copied section in the previous response). We agree that it is difficult to achieve these preconditions to test for latent mean differences, and that it is likely that we will not find latent variance or path equaliviance. However, we believe that differences in latent variance and path coefficients are interesting research findings by their own merits.

10. I'm not sure how the additional analyses should be handled. It might be better to explore and add them later as “exploratory,” rather than try to fully delineate here. Just one thing I considered: political ideology surely cannot be modelled consistently across countries, right? Do you have a way to measure ideology that you believe would be cross-nationally invariant? Additional points: - Hypotheses could be more clearly expressed (p. 7). The diagram was fairly clear, but the text was ambiguously worded. - In addition to the .qsf file, post a copy of the .docx from Qualtrics so that people can see the questionnaire without loading it into Qualtrics.

Author Response: In order to sufficiently distance the "additional" analyses from the main objectives of the study, we have now framed them as "exploratory analyses" and not tried to delineate their analysis further.

For a response to your query about the political ideology question, please see our response to Reviewer 1, Comment 9. In short, the political ideology question has been used over a long period of time internationally in the World Values Survey, and data is available for all three

countries (Canada, the US, and the UK) from a large sample using a close variant of this question.

To clarify our hypotheses, and in particular to justify the use of a mediation model to test these hypotheses, we have presented a more robust (and hopefully more direct) set of hypotheses in the introduction that match our presentation of the same ideas in Table 1.

added both exported .qsf and .docx files of the survey to the OSF. Please note that since the exported .docx file is not very well presented, all the items of all parts of the survey have been separately uploaded in the "Material" subfolder on OSF as well.

Signed, Katherine S. Corker

Appendix B

Comment E.1

There are however some remaining issues to address in the Stage 1 manuscript before progressing with this preliminary study. Foremost is the major concern of Reviewer 1 about untestable (and potentially unsafe) assumptions in the mediation model, with the reviewer (quite reasonably in my view) suggesting a more circumspect approach in which the focus is on association and conclusions regarding causality are caveated and speculative. Reviewer 3 also notes a number of details requiring further clarification.

Response E.1

In light of the comments from reviewer 1, we have opted to adopt reviewer 1's suggestion of employing a correlational model instead. We have made extensive edits to the manuscript to reflect this. We do believe that a mediation tested as an exploratory analysis with speculation and caveats clearly indicated would be valuable if we were to speculate on the mediating nature of the intercorrelations. Please see comment 1.1 for more details.

We have included more details regarding the procedure that we will use to test reliability, convergent validity and discriminant validity. Regarding reviewer 3's comment regarding concurrent validity of the latent factors, we appreciate that we currently have no planned analysis to test whether the constructs measure what we purport it to measure. We further agree that optimally, we would have established tests of numeracy with which we can compare our measures. For both the basic numeracy and COVID-19 health numeracy measures, however, we believe the construct validity is adequate as these measures were made by experts in the numerical cognition field. Further, the measures are adapted from existing tests of numerical ability. We are less certain regarding COVID-19 attitudes and behaviors, however, we believe this measure has reasonable face validity.

Reviewer 1

Comment 1.1

While I appreciate that the authors took care in responding to my concern about mediation, that concern has not been put to rest for me. The authors have added justifications for the paths in the mediation model. Those justifications address the theoretical plausibility of those paths. But they do not address the conditions necessary to make inferences about them from data. That depends on the rest of the model being correct, and that is the source of my concern.

To be more explicit, for the model to be correctly specified and the mediation correctly estimated, every one of the following untestable assumptions needs to be true:

- a. The effect of COVID-19 health numeracy on basic numeracy is zero.
- b. No confounding variable has an effect on basic numeracy and health numeracy.
- c. The effect of health-related attitudes/behaviors on health numeracy is zero.
- d. No confounding third variable has an effect on health numeracy and health attitudes/behaviors.
- e. The effect of health attitudes/behaviors on basic numeracy is zero.
- f. No confounding third variable affects basic numeracy and health attitudes/behaviors.

I am not persuaded that all of these assumptions are safe to make. It seems plausible that there could be unmeasured confounding variables (SES, intelligence, access to health care, access to education, on and on). Some of the reverse paths also seem plausible, for example, that people can generalize from domain-specific numeracy skills, or that taking COVID seriously (in one's attitudes or behaviors) might motivate someone to read and learn about COVID statistics and testing.

While I understand the argument that an imperfect study can motivate a better one in the future, I also often see the opposite: caveats and hedges get lost after a study comes out and gets translated from journal article to press release to media (or from journal article to citation in future journal article). I would rather see this presented as a study of associations, with the causal story clearly marked as speculation only and not "supported" by a mediation analysis.

Response 1.1

Upon further consideration, we agree with reviewer 1 that presenting the current study as a study of associations would be more prudent. To that end, we have modified our preregistration document to reflect this. First, we have removed the mention of a mediation model in the abstract and introduction. Second, we have modified our analysis to consider associations and between-country differences in those associations.

Comment 1.2

I really like the expansion of the preliminary study into more of a full-blown measurement study. I think it creates a greater chance of success for the ultimate main study. Below I

have some comments on it.

a. I'm nitpicking here, but this is not a "pilot" study (see the references in my previous review). I'd call it a measurement study, or just Study 1.

b. In the cover letter, the authors mentioned they were weighing collecting all the data in one country versus in all 3. I can see merits to both and think this should be the authors' call. If it were me, I would lean toward the 3-country plan. It matches what the eventual main study will look like, and it gives the authors a chance to do some exploration of measurement invariance in that dataset.

c. The measurement study plan has some risk of overfitting. Ideally in a programmatic measurement investigation, you would do EFA and item trimming/retooling in one (or more) samples, and then do your CFA in a new sample. Otherwise, the CFA can be overfit because of capitalizing on chance in the EFA phase. Of note, this will not create a risk of overfitting in the second, main study. However, it does create a possibility that the measurement indices (fit, loadings, etc.) in the main study could look worse than they did in the overfit Study 1.

Whether that'll be a little or a lot is hard to say: speaking heuristically, the more changes the authors make in the EFA stage, the more I would worry about overfitting in the CFA. With all that being said, there is also a very important practical question of how much measurement research the authors should do before getting to the very-timely main question. The perfect should not be the enemy of the good. So again, I'm sharing some thoughts that I hope are helpful, but in the end I think this should be the authors' decision. It's a judgment call, and they will be in a good position to make it after they see how the Study 1 data looks.

d. Small thing, but should the "item parcels" section be moved up to Study 1?

Response 1.2

a. We agree with reviewer 1 it is not a pilot study per se; however, we somewhat hesitate to call it study 1 or measurement study. This is because we will be conducting both an EFA and CFA on the same data (as you have pointed out in comment 1.2c). Due to the increased chance of overfitting in such a case, there may be some bias in the results. While we believe these preliminary results would be useful for purposes of informing the main data collection, calling these results as study 1 may mislead some readers less familiar with SEM regarding the reliability of these results. As such, we have opted to call it a preliminary measurement study to place less weight on the study.

b. Thank you for your opinion, we agree that there is merit in both approaches and your opinion is greatly appreciated.

c. We agree that running an EFA following by a CFA increases the chances of overfitting. Further, we agree that more optimally we would run an EFA on one sample and CFA on another. We believe the current approach is a good compromise between statistical rigor and practicality.

We see the preliminary measurement study as serving two main functions: (1) identifying problematic items, and (2) ascertaining the overall factor structure of the variables for purposes of preregistration. The first function would be fulfilled by the EFA across individual items, and the second function would be fulfilled by the CFA across formed parcels. As you have pointed out, there may be chance correlations in the preliminary measurement study that may influence the parceling strategy used in the main study. This would cause a reduction in model fit; however, we believe unless a large portion of items exhibit bias, part of the risk would be ameliorated by the use of parcels. Finally, we agree that some evaluation would be needed if major changes are required in the EFA stage. We believe that once the preliminary study is finished, we will have a better idea of the scope of changes that might be necessary.

d. We have done so in the current iteration of the manuscript.

Reviewer 3

*Please note that the pilot study has been renamed the preliminary measurement study (please see comment 1.2a). Further, we have elected to only examine the intercorrelations among the latent variables rather than doing a mediation analysis (please see comment 1.1).

Comment 3.3

The plan to increase the pilot study sample size to 500 is a good one. Regarding whether you should sample from one country or all three, I have the same intuition as you: to collect from all three countries, but to ignore between country differences at the pilot stage. I don't have a strong justification for that intuition.

Unresolved issues: You say you will examine convergent and discriminant validity, but you don't say how (well the revised text says a little more, but it is still missing important details). What

variables will you use to test these validities? Or is the proposal only to test the validity of the internal structure of the measure (which does not really make sense to me)? Finally, it is typically not advisable to conduct an EFA and a CFA on the same data (as it will result in overfitting). I'm not clear why you propose to do so here (but maybe I misunderstand the plan).

Response 3.3

We have expanded the section discussing convergent and discriminant validity and included the specific criteria that we will use to evaluate. As described the section is as follows:

“Reliability and Validity

Subsequent to estimating the EFA and CFA in both the preliminary measurement study and main study (see below). Reliability – the degree to which indicators of the latent factor consistently measure the same underlying construct – will be assessed using coefficient omega for each latent variable estimated (McDonald, 2013). We will use the typical cut-off point for acceptable reliability of 0.7 for each of the three latent variables (Kline, 2015; McDonald, 2013). Once reliability is established, convergent validity – the degree to which individual items are strongly related to the hypothesized factor – will be assessed using average variance extracted (AVE). We will use the typical cutoff score for acceptable convergent validity, where AVE for each of the three factors should be higher than 0.5. Finally, discriminant validity – the degree to which the degree to which individual items are not related to other factors – will be assessed using the Fornell & Larcker criterion (Fornell & Larcker, 1981; Hair et al., 2016). Specifically, we will adopt the convention that the square root of the AVE for each construct should be larger than the correlation of the specific construct with any of the other construct.”

Regarding concurrent validity of the latent factors, we appreciate that we currently have no planned analysis to test whether the constructs measure what we purport it to measure. We further agree that optimally, we would have established tests of numeracy with which we can compare our measures. For both the basic numeracy and COVID-19 health numeracy measures, however, we believe the construct validity is adequate as these measures were made by experts in the numerical cognition field. Further, the measures are adapted from existing tests of numerical ability. We are less certain regarding COVID-19 attitudes and behaviors, however, we believe this measure has reasonable face validity.

Regarding EFA and CFA on the same sample, reviewer 1 has voiced similar concerns in comment 1.2c. We do agree that running an EFA following by a CFA increases the chances of overfitting. Further, we agree that more optimally we would run an EFA on one sample and CFA on another. We believe the current approach is a good compromise between statistical rigor and practicality.

We see the preliminary measurement study as serving two main functions: (1) identifying problematic items, and (2) ascertaining the overall factor structure of the variables for purposes of preregistration. The first function would be fulfilled by the EFA across individual items, and the second function would be fulfilled by the CFA across formed parcels. As you have pointed out, biases in the items may lead to parceling strategies that take advantage of these biases and may result in overfitting of the CFA. However, we do not think this is particularly problematic as the preliminary measurement study is mainly to inform main data collection and this bias would not be present in the main study. Indeed, reviewer 1 has pointed out that if there is bias in the preliminary measurement study, it would cause a reduction in model fit in the main study. We believe unless a large portion of items exhibit bias, part of the risk would be ameliorated by the use of parcels. Finally, reviewer 1 has pointed out that if major changes are needed from the results of the EFA, we would have to be wary that model fit in the main study would be affected, we agree and if we do need to make major changes, some judgement would be required with regard to whether an additional measurement study may be needed.

Comment 3.5

I'm not sure this is an ideal way to handle covariates (i.e., including them only if they are statistically significant; a data dependent choice). I guess if it were me, I would plan for my primary analysis to be covariate free, and then present the analysis with covariates all included as a robustness check. Or, perhaps you could commit to including or excluding the covariates in the pilot stage, so that in the final test, the choice to include a covariate is not data dependent. I will look for some details on this at the next stage, when the pilot data are in.

Response 3.5

We agree with reviewer 3. We will include all covariates irrespective of statistical significance, and we will explicitly add covariates as a robustness check.

Comment 3.6

OK, this should be noted in the registration document somewhere.

Response 3.6

We will have added the section. As described the section is as follows:

“Finally, the preliminary measurement study will be used to inform the actual number of participants that would be sampled in the main study. As we have set multiple criteria to exclude inattentive participants, we will need sample extra cases to allow for exclusions and attrition. The proportion of participants excluded in the preliminary measurement study will be used to estimate the number of participants required to reach 675 complete cases per country. This information will be included in the revisions to stage 1 registered report.”

Comment 3.7

OK, for the main study, but I remain confused as to why it appears that an EFA and a CFA on the same data in the pilot study is proposed.

Response 3.7

Please see response 3.3.

Comment 3.10

To clarify the point about exploratory analyses: Authors in RRs are always allowed to provide new exploratory results in sections that are subsequent to, and clearly delineated from, the pre-registered results sections. Planned explorations can, but are not required to be, disclosed at stage 1. Stage 1, rightly, focuses more on the confirmatory tests. It sounds like you have found an OK middle ground here, but just know that if something appears in the Stage 1 plan (even as exploratory), it must appear in the final report. So sometimes it's better to let the explorations be true explorations and show up later after you've (transparently) explored.

Response 3.10

On consideration of reviewer 3's comments, we have decided to remove the exploratory analysis from the document for now. We may revisit the exploratory analysis once we have a better idea of how the data may look like after the preliminary study.

Comment 3.11

New point: Figure 4 is helpful to see the planned process for the main analysis (Figure 3 is nice too). I see that not all parameters in the final model are set a priori (e.g., modification indices, a data dependent feature, are invoked early in the tree). I also see that invariance testing happens after mediation testing, rather than before (which does not make sense to me). The structural model is needed before you can do the mediation test, right? You can't test your mediation model across countries unless you have invariance first. I will look for

this modification in the next stage of the draft.

Response 3.11

While we would use the parceling strategy from the preliminary measurement study for the main data analysis, we believe it would be prudent to examine for nuisance parameters in the modification indices before proceeding. Of course, if modifications are made, we will report both the original and modified model (if modifications are minor, the original model could be added to an online supplementary section).

With regards to calculating an overall mediation model (now changed to correlation model; please see comment 1.1) before invariance testing, we believe this is a reasonable first step from a model building perspective. In the case that we do not find significant differences in the intercorrelations between countries, these correlations would be a reasonable description of the whole population. We are open to removing the overall mediation model if reviewer 3 believes it is unnecessary.

Comment 3.12

To summarize: I support the plan to conduct the revised pilot/preliminary study with $N=500$ and to use this study as the basis for a revised Stage 1 RR prior to the collection of the main dataset, but I remain skeptical that the measurement model will work out as proposed.. It might be useful to consider the circumstances that would cause the authors to abandon the plan to proceed to a second/final study – for instance if the measurement model in the pilot proves to need more changes than can be fruitfully examined in a single preliminary study. Figure 3 sheds some light on this (“substantial modifications” may be needed). Ultimately it may be hard to say ahead of time exactly how substantial the changes would need to be before more measurement work would need to be done ahead of a final hypothesis test.

Response 3.12

We are reasonably confident that the numeracy and health numeracy latent variables will be as hypothesized, as these items were constructed with the help of a panel of math cognition experts. One point of departure that reviewer 3 has mentioned is that COVID-19 attitudes and COVID-19 actions may not come together as one latent variable. In such a case, we will detect this with the preliminary measurement model. With regards to other substantial changes to the hypothesized model, we will have a better idea once the preliminary measurement model is done.

Appendix C

Response to Reviewers

RSOS-201303.R3 “Numeracy and COVID-19: examining interrelationships between numeracy, health numeracy and behaviour”

Dear Reviewers,

In what follows below we have provided our responses to your comments on our 2nd revision of the above referenced manuscript. As you might recall, the decision by the editor, Dr. Chris Chambers, was that we were to conduct a measurement study before further review of our manuscript would proceed. We have now completed this measurement study. We are therefore pleased to submit a revised manuscript (3rd revision) that contains both a.) our responses to the previous reviews (marked in blue in the revised manuscript) and b.) the results of our preliminary measurement studies (marked in green in the revised manuscript). In view of the results of this preliminary measurement study, we detail how we have adjusted our survey in view of these results.

Response to Reviews of RSOS-201303.R3:

Comment E.1

There are however some remaining issues to address in the Stage 1 manuscript before progressing with this preliminary study. Foremost is the major concern of Reviewer 1 about untestable (and potentially unsafe) assumptions in the mediation model, with the reviewer (quite reasonably in my view) suggesting a more circumspect approach in which the focus is on association and conclusions regarding causality are caveated and speculative. Reviewer 3 also notes a number of details requiring further clarification.

Response E.1

In light of the comments from reviewer 1, we have opted to adopt reviewer 1’s suggestion of employing a correlational model instead. We have made extensive edits to the manuscript to reflect this. We do believe that a mediation tested as an exploratory analysis with speculation and caveats clearly indicated would be valuable if we were to speculate on the mediating nature of the intercorrelations. Please see comment 1.1 for more details.

We have included more details regarding the procedure that we will use to test reliability, convergent validity and discriminant validity. Regarding reviewer 3’s comment regarding concurrent validity of the latent factors, we appreciate that we currently have no planned analysis to test whether the constructs measure what we purport it to measure. We further agree that optimally, we would have established tests of numeracy with which we can compare our measures. For both the basic numeracy and COVID-19 health numeracy measures, however, we believe the construct validity is adequate as these measures were made by experts in the numerical cognition field. Further, the measures are adapted from

existing tests of numerical ability. We are less certain regarding COVID-19 attitudes and behaviors, however, we believe this measure has reasonable face validity.

Reviewer 1

Comment 1.1

While I appreciate that the authors took care in responding to my concern about mediation, that concern has not been put to rest for me. The authors have added justifications for the paths in the mediation model. Those justifications address the theoretical plausibility of those paths. But they do not address the conditions necessary to make inferences about them from data. That depends on the rest of the model being correct, and that is the source of my concern.

To be more explicit, for the model to be correctly specified and the mediation correctly estimated, every one of the following untestable assumptions needs to be true:

- a. The effect of COVID-19 health numeracy on basic numeracy is zero.
- b. No confounding variable has an effect on basic numeracy and health numeracy.
- c. The effect of health-related attitudes/behaviors on health numeracy is zero.
- d. No confounding third variable has an effect on health numeracy and health attitudes/behaviors.
- e. The effect of health attitudes/behaviors on basic numeracy is zero.
- f. No confounding third variable affects basic numeracy and health attitudes/behaviors.

I am not persuaded that all of these assumptions are safe to make. It seems plausible that there could be unmeasured confounding variables (SES, intelligence, access to health care, access to education, on and on). Some of the reverse paths also seem plausible, for example, that people can generalize from domain-specific numeracy skills, or that taking COVID seriously (in one's attitudes or behaviors) might motivate someone to read and learn about COVID statistics and testing.

While I understand the argument that an imperfect study can motivate a better one in the future, I also often see the opposite: caveats and hedges get lost after a study comes out and gets translated from journal article to press release to media (or from journal article to citation in future journal article). I would rather see this presented as a study of associations, with the causal story clearly marked as speculation only and not "supported" by a mediation analysis.

Response 1.1

Upon further consideration, we agree with reviewer 1 that presenting the current study as a study of associations would be more prudent. To that end, we have modified our preregistration document to reflect this. First, we have removed the mention of a mediation model in the abstract and introduction. Second, we have modified our analysis to consider associations and between-country differences in those associations.

Comment 1.2

I really like the expansion of the preliminary study into more of a full-blown measurement study. I think it creates a greater chance of success for the ultimate main study. Below I have some comments on it.

a. I'm nitpicking here, but this is not a "pilot" study (see the references in my previous review). I'd call it a measurement study, or just Study 1.

b. In the cover letter, the authors mentioned they were weighing collecting all the data in one country versus in all 3. I can see merits to both and think this should be the authors' call. If it were me, I would lean toward the 3-country plan. It matches what the eventual main study will look like, and it gives the authors a chance to do some exploration of measurement invariance in that dataset.

c. The measurement study plan has some risk of overfitting. Ideally in a programmatic measurement investigation, you would do EFA and item trimming/retooling in one (or more) samples, and then do your CFA in a new sample. Otherwise, the CFA can be overfit because of capitalizing on chance in the EFA phase. Of note, this will not create a risk of overfitting in the second, main study. However, it does create a possibility that the measurement indices (fit, loadings, etc.) in the main study could look worse than they did in the overfit Study 1.

Whether that'll be a little or a lot is hard to say: speaking heuristically, the more changes the authors make in the EFA stage, the more I would worry about overfitting in the CFA. With all that being said, there is also a very important practical question of how much measurement research the authors should do before getting to the very-timely main question. The perfect should not be the enemy of the good. So again, I'm sharing some thoughts that I hope are helpful, but in the end I think this should be the authors' decision. It's a judgment call, and they will be in a good position to make it after they see how the Study 1 data looks.

d. Small thing, but should the "item parcels" section be moved up to Study 1?

Response 1.2

a. We agree with reviewer 1 it is not a pilot study per se; however, we somewhat hesitate to call it study 1 or measurement study. This is because we will be conducting both an EFA and CFA on the same data (as you have pointed out in comment 1.2c). Due to the increased chance of overfitting in such a case, there may be some bias in the results. While we believe these preliminary results would be useful for purposes of informing the main data collection, calling these results as study 1 may mislead some readers less familiar with

SEM regarding the reliability of these results. As such, we have opted to call it a preliminary measurement study to place less weight on the study.

b. Thank you for your opinion, we agree that there is merit in both approaches and your opinion is greatly appreciated.

c. We agree that running an EFA following by a CFA increases the chances of overfitting. Further, we agree that more optimally we would run an EFA on one sample and CFA on another. We believe the current approach is a good compromise between statistical rigor and practicality.

We see the preliminary measurement study as serving two main functions: (1) identifying problematic items, and (2) ascertaining the overall factor structure of the variables for purposes of preregistration. The first function would be fulfilled by the EFA across individual items, and the second function would be fulfilled by the CFA across formed parcels. As you have pointed out, there may be chance correlations in the preliminary measurement study that may influence the parceling strategy used in the main study. This would cause a reduction in model fit; however, we believe unless a large portion of items exhibit bias, part of the risk would be ameliorated by the use of parcels. Finally, we agree that some evaluation would be needed if major changes are required in the EFA stage. We believe that once the preliminary study is finished, we will have a better idea of the scope of changes that might be necessary.

d. We have done so in the current iteration of the manuscript.

Reviewer 3

*Please note that the pilot study has been renamed the preliminary measurement study (please see comment 1.2a). Further, we have elected to only examine the intercorrelations among the latent variables rather than doing a mediation analysis (please see comment 1.1).

Comment 3.3

The plan to increase the pilot study sample size to 500 is a good one. Regarding whether you should sample from one country or all three, I have the same intuition as you: to collect from all three countries, but to ignore between country differences at the pilot stage. I don't have a strong justification for that intuition.

Unresolved issues: You say you will examine convergent and discriminant validity, but you don't say how (well the revised text says a little more, but it is still missing important details). What

variables will you use to test these validities? Or is the proposal only to test the validity of the internal structure of the measure (which does not really make sense to me)? Finally, it is typically not advisable to conduct an EFA and a CFA on the same data (as it will result in overfitting). I'm not clear why you propose to do so here (but maybe I misunderstand the plan).

Response 3.3

We have expanded the section discussing convergent and discriminant validity and included the specific criteria that we will use to evaluate. As described the section is as follows:

“Reliability and Validity

Subsequent to estimating the EFA and CFA in both the preliminary measurement study and main study (see below). Reliability – the degree to which indicators of the latent factor consistently measure the same underlying construct – will be assessed using coefficient omega for each latent variable estimated (McDonald, 2013). We will use the typical cut-off point for acceptable reliability of 0.7 for each of the three latent variables (Kline, 2015; McDonald, 2013). Once reliability is established, convergent validity – the degree to which individual items are strongly related to the hypothesized factor – will be assessed using average variance extracted (AVE). We will use the typical cutoff score for acceptable convergent validity, where AVE for each of the three factors should be higher than 0.5. Finally, discriminant validity – the degree to which the degree to which individual items are not related to other factors – will be assessed using the Fornell & Larcker criterion (Fornell & Larcker, 1981; Hair et al., 2016). Specifically, we will adopt the convention that the square root of the AVE for each construct should be larger than the correlation of the specific construct with any of the other construct.”

Regarding concurrent validity of the latent factors, we appreciate that we currently have no planned analysis to test whether the constructs measure what we purport it to measure. We further agree that optimally, we would have established tests of numeracy with which we can compare our measures. For both the basic numeracy and COVID-19 health numeracy measures, however, we believe the construct validity is adequate as these measures were made by experts in the numerical cognition field. Further, the measures are adapted from existing tests of numerical ability. We are less certain regarding COVID-19 attitudes and behaviors, however, we believe this measure has reasonable face validity.

Regarding EFA and CFA on the same sample, reviewer 1 has voiced similar concerns in comment 1.2c. We do agree that running an EFA following by a CFA increases the chances of overfitting. Further, we agree that more optimally we would run an EFA on one sample and CFA on another. We believe the current approach is a good compromise between statistical rigor and practicality.

We see the preliminary measurement study as serving two main functions: (1) identifying problematic items, and (2) ascertaining the overall factor structure of the variables for purposes of preregistration. The first function would be fulfilled by the EFA across individual items, and the second function would be fulfilled by the CFA across formed parcels. As you have pointed out, biases in the items may lead to parceling strategies that

take advantage of these biases and may result in overfitting of the CFA. However, we do not think this is particularly problematic as the preliminary measurement study is mainly to inform main data collection and this bias would not be present in the main study. Indeed, reviewer 1 has pointed out that if there is bias in the preliminary measurement study, it would cause a reduction in model fit in the main study. We believe unless a large portion of items exhibit bias, part of the risk would be ameliorated by the use of parcels. Finally, reviewer 1 has pointed out that if major changes are needed from the results of the EFA, we would have to be wary that model fit in the main study would be affected, we agree and if we do need to make major changes, some judgement would be required with regard to whether an additional measurement study may be needed.

Comment 3.5

I'm not sure this is an ideal way to handle covariates (i.e., including them only if they are statistically significant; a data dependent choice). I guess if it were me, I would plan for my primary analysis to be covariate free, and then present the analysis with covariates all included as a robustness check. Or, perhaps you could commit to including or excluding the covariates in the pilot stage, so that in the final test, the choice to include a covariate is not data dependent. I will look for some details on this at the next stage, when the pilot data are in.

Response 3.5

We agree with reviewer 3. We will include all covariates irrespective of statistical significance, and we will explicitly add covariates as a robustness check.

Comment 3.6

OK, this should be noted in the registration document somewhere.

Response 3.6

We will have added the section. As described the section is as follows:

“Finally, the preliminary measurement study will be used to inform the actual number of participants that would be sampled in the main study. As we have set multiple criterions to exclude inattentive participants, we will need sample extra cases to allow for exclusions and attrition. The proportion of participants excluded in the preliminary measurement study will be used to estimate the number of participants required to reach 675 complete cases per country. This information will be included in the revisions to stage 1 registered report.”

Comment 3.7

OK, for the main study, but I remain confused as to why it appears that an EFA and a CFA on the same data in the pilot study is proposed.

Response 3.7

Please see response 3.3.

Comment 3.10

To clarify the point about exploratory analyses: Authors in RRs are always allowed to provide new exploratory results in sections that are subsequent to, and clearly delineated from, the pre-registered results sections. Planned explorations can, but are not required to be, disclosed at stage 1. Stage 1, rightly, focuses more on the confirmatory tests. It sounds like you have found an OK middle ground here, but just know that if something appears in the Stage 1 plan (even as exploratory), it must appear in the final report. So sometimes it's better to let the explorations be true explorations and show up later after you've (transparently) explored.

Response 3.10

On consideration of reviewer 3's comments, we have decided to remove the exploratory analysis from the document for now. We may revisit the exploratory analysis once we have a better idea of how the data may look like after the preliminary study.

Comment 3.11

New point: Figure 4 is helpful to see the planned process for the main analysis (Figure 3 is nice too). I see that not all parameters in the final model are set a priori (e.g., modification indices, a data dependent feature, are invoked early in the tree). I also see that invariance testing happens after mediation testing, rather than before (which does not make sense to me). The structural model is needed before you can do the mediation test, right? You can't test your mediation model across countries unless you have invariance first. I will look for this modification in the next stage of the draft.

Response 3.11

While we would use the parceling strategy from the preliminary measurement study for the main data analysis, we believe it would be prudent to examine for nuisance parameters in the modification indices before proceeding. Of course, if modifications are made, we will report both the original and modified model (if modifications are minor, the original model could be added to an online supplementary section).

With regards to calculating an overall mediation model (now changed to correlation model; please see comment 1.1) before invariance testing, we believe this is a reasonable first step from a model building perspective. In the case that we do not find significant differences in the intercorrelations between countries, these correlations would be a reasonable description of the whole population. We are open to removing the overall mediation model if reviewer 3 believes it is unnecessary.

Comment 3.12

To summarize: I support the plan to conduct the revised pilot/preliminary study with N=500 and to use this study as the basis for a revised Stage 1 RR prior to the collection of the main dataset, but I remain skeptical that the measurement model will work out as proposed.. It might be useful to consider the circumstances that would cause the authors to abandon the plan to proceed to a second/final study – for instance if the measurement

model in the pilot proves to need more changes than can be fruitfully examined in a single preliminary study. Figure 3 sheds some light on this (“substantial modifications” may be needed). Ultimately it may be hard to say ahead of time exactly how substantial the changes would need to be before more measurement work would need to be done ahead of a final hypothesis test.

Response 3.12

We are reasonably confident that the numeracy and health numeracy latent variables will be as hypothesized, as these items were constructed with the help of a panel of math cognition experts. One point of departure that reviewer 3 has mentioned is that COVID-19 attitudes and COVID-19 actions may not come together as one latent variable. In such a case, we will detect this with the preliminary measurement model. With regards to other substantial changes to the hypothesized model, we will have a better idea once the preliminary measurement model is done.

Appendix D

Numeracy and COVID-19: examining interrelationships between numeracy, health numeracy and behaviour

Nathan T.T. Lau¹, Eric D. Wilkey¹, Mojtaba Soltanlou¹, Rebekka Lagacé Cusiac¹, Lien Peters¹, Paul Tremblay¹, Celia Goffin¹, Isabella Starling Alves², Andrew David Ribner³, Clarissa Thompson⁴, Jo Van Hoof⁵, Julia Bahnmüller⁶, Aymee Alvarez¹, Elien Bellon⁷, Ilse Coolen⁸, Fanny Ollivier⁹, Daniel Ansari¹

¹Department of Psychology, Western University, Canada

²Department of Educational Psychology, University of Wisconsin-Madison, USA

³Learning Research and Development Center, University of Pittsburgh, USA

⁴Department of Psychological Sciences, Kent State University, USA

⁵Centre for Instructional Psychology and Technology, KU Leuven, Belgium

⁶Centre for Mathematical Cognition, Loughborough University, United Kingdom

⁷Parenting and Special Education, KU Leuven, Belgium

⁸Université de Paris, LaPsyDÉ, CNRS, F-75005 Paris, France

⁹Laboratoire de Psychologie, Cognition, Comportement et Communication, Université Rennes 2, France

Corresponding Author: Daniel Ansari, Department of Psychology, Western University, daniel.ansari@uwo.ca

Abstract

The COVID-19 pandemic has exposed people across the globe to large amounts of statistical data. Previous studies have shown that individuals' mathematical understanding of health-related information affects their attitudes and behaviours. Here, we investigate the relation between a) basic numeracy, b) COVID-19 health numeracy and c) COVID-19 health-related attitudes and behaviours. To do this, an online survey measuring these three variables will be distributed in Canada, the United States, and the United Kingdom. Basic Numeracy, COVID-19 health numeracy and COVID-19 health-related attitudes and behaviours are expected to be positively correlated with each other. Multigroup analysis will be used to investigate mean differences and differences in the strength of the correlation.

Keywords: numeracy, health numeracy, COVID-19, health policy and adherence

Introduction

The outbreak of the COVID-19 pandemic represents an unprecedented event. While human history is dotted with pandemics, never before have we been able to track and model a disease with such high sophistication. One of the many ways in which COVID-19 represents a watershed moment is the way in which it has pushed people across the planet to process and interpret rapidly evolving numerical information to inform their behaviours. Websites reporting COVID-19 statistics, such as total number of infections and total number of deaths, have been created (e.g., <https://coronavirus.jhu.edu>) and are widely cited in media reports on COVID-19, leading to unprecedented levels of attention to dynamic numerical information among the general population.

In addition to the numerical information pertaining to the consequences of the virus (e.g., number of cases, number of deaths), people across the planet are being introduced to unfamiliar and challenging mathematical concepts, such as ‘flattening the curve’, ‘exponential growth’, ‘false negative rates’ of COVID-19 tests, etc. Specifically, the degree to which individuals can make sense of COVID-19 numbers may influence the way they understand these critical health-related concepts which may, in turn, influence their adherence to public health advice and their perception of the risks posed by COVID-19.

Indeed, several experts in the study of mathematical cognition have recently discussed the role that numeracy (e.g., understanding proportions, large numbers) may play in how individuals estimate the risks associated with COVID-19 (Shepherd, 2020; Thompson, Taber, Coifman, et al., 2020; Thompson, Taber, Sidney, et al., 2020). For example, an online experimental study with over 1,200 U.S. participants found that those exposed to a short educational intervention consisting of a worked example with instructions on how to calculate fatality rates were more accurate on post-intervention health decision-making problems than those participants who were randomly assigned to a control condition (Thompson, Taber, Sidney, et al., 2020). Importantly, COVID-19 risk perceptions and worry increased throughout the 10-day study for those in the intervention condition, relative to those in the control condition. Since risk perception inherently involves magnitude comparisons (i.e., higher vs. lower risk), improving proportional understanding may have increased the accuracy involved in assessing risk magnitude. However, there are still open questions pertaining to the ways in which numeracy and understanding of health information related to COVID-19 may affect people's behaviour in response to the pandemic. Therefore, the aim of the present study is to address these knowledge gaps. In what follows, we first review the mathematical concepts and misconceptions that are relevant to understanding COVID-19 related information. We then go on to discuss how we aim to study the interrelations between (A) basic numerical processing and understanding (hereafter basic numeracy), (B) health numeracy directly relevant to COVID-19 (hereafter COVID-19 health numeracy) and (C) COVID-19 health-related attitudes and behaviours.

During the COVID-19 pandemic, policy makers and public figures frequently cite numerical information in order to defend decisions and influence public action. For example, some have argued against radical actions (e.g., stay-at-home orders and business closures) on the premise that equally large numbers of people die from other causes such as automobile accidents every year without causing nationwide shutdowns (Ali, 2020). As of April 7th (nearly a month after the first COVID-19 death was recorded in the US), John Hopkins University reported 31.4 thousand COVID-19 related deaths in the US compared to the 39.4 thousand deaths reported by the National Safety Council due to car accidents in the US over the course of 2018. However, comparing the current number of COVID-19 related deaths to the number of people who died in car accidents in a given year is like comparing apples and oranges, in part because both statistics have different denominators (i.e., one is per month while the other is per year). When taking the time period into account, we can observe that 31.4 thousand people died of COVID-19 while approximately 3 thousand people died of car accidents in the same one-month time period ($39,000/12=3,250$), leading us to draw drastically different conclusions about the severity of the pandemic. This example shows how important relational information (i.e., proportions) is when making decisions surrounding COVID-19.

Despite the importance of understanding relational information, there is a wealth of empirical evidence to show that processing proportions (fractions, decimals, percentages or odds ratios) is challenging for many adults (Lusardi, 2012; Spiegelhalter et al., 2011). What underlies these difficulties in processing proportions? One of the possible sources is the whole number bias. The whole number bias refers to the inappropriate use of whole number properties (i.e., natural numbers) when processing rational numbers (e.g., fractions and decimals; Ni & Zhou, 2005). In the specific case of fractions and odds ratios, the whole number bias also manifests when individuals pay more attention to the components of the proportion (i.e., the numerator and/or denominator) rather than to the magnitude of the entire ratio (i.e., the relation between the numerator and denominator; Braithwaite & Siegler, 2018; Lewis et al., 2016). For example, an individual might perceive a chance of 1 in 10 as being smaller than 10 in 100 simply because the numbers of the first odds ratio (i.e., 1 and 10) are smaller than the numbers in the second odds ratio (i.e., 10 and 100) although they represent the same proportion (i.e., 0.1).

In applied settings, paying attention to the magnitude of components rather than to the magnitude of the entire proportion can lead to other types of biases such as denominator neglect. Denominator neglect occurs when individuals compare two or more numbers of events without taking into account the total number of opportunities for that event to occur (Spiegelhalter et al., 2011). In the context of the COVID-19 pandemic, examples of this bias are common in the media when public figures state that the United States has carried out more tests than any other country, while ignoring the fact that the number of tests per capita (i.e., the number of tests accounting for population size) is much lower than several other countries at the time of writing. In addition to these misleading claims, other factors, such as how statistics are presented in the media, can also increase the occurrence of denominator neglect in the general population. For example, the media

often reports, either directly or with the use of graphs, absolute magnitudes such as the total number of COVID-19 cases. Although this information is important for health planning (e.g., estimating the number of ventilators needed), it also puts focus on the numerator and leads individuals comparing different regions to ignore total population size.

As is evident from the above, a strong understanding of proportions in a variety of formats is necessary in order to accurately understand COVID-19 related information. Moreover, the size of the numbers that are contained within COVID-19 related data may be a barrier to understanding that information and acting accordingly. For instance, it has been shown that many individuals have misrepresentations when it comes to their understanding of large numbers (i.e., in the order of thousands, millions and billions; Landy et al., 2013, 2017). Failing to grasp important magnitude differences between large numbers could result in altered perceptions such as minimizing the impact of the pandemic on a larger scale. Furthermore, processing proportions with large numbers may create additional cognitive load, even when individuals have an accurate representation of large numbers (Deck & Jahedi, 2015). Lastly, large numbers in COVID-19 data are a result of the virus' exponential spread, which is another concept that is poorly understood since people are more often exposed to linear (or arithmetic) growth (Siegel, 2020). All of these factors may translate into difficulty digesting complex mathematical information related to large-scale numbers associated with COVID-19. As a result, poor understanding of these concepts may lead people to undermine efforts to slow the spread of the disease by not following social distancing guidelines.

The review of the literature above suggests that having difficulties understanding mathematical concepts and accurately processing numerical quantities may influence the way in which individuals understand and act upon COVID-19 related information. Indeed, it has been shown that individuals with higher numeracy are more likely to pay attention to numerical information, interpret it correctly and make decisions accordingly relative to those with low numeracy (Peters et al., 2007). In contrast, individuals with relatively low numeracy are more susceptible to bias. For example, they are more likely to ignore numerical information and instead rely on intuitions based on emotional states and other extraneous factors, such as their trust in or distrust of the information source (Peters et al., 2007; Reyna et al., 2009). Furthermore, numeracy has been shown to influence how people evaluate risk (Keller & Siegrist, 2009; Schwartz et al., 2004; Woloshin et al., 2001). For example, individuals with higher numeracy levels have been shown to respond differently to high-risk vs low-risk situations while individuals with low numeracy do not (Keller & Siegrist, 2009). Another study on numeracy and adaptive decision making found that individuals with low numeracy were more likely to choose riskier options when small losses were inevitable (Jasper et al., 2013).

To the best of our knowledge, there has, to date, been no study that has examined the role of basic numeracy alongside applied, health-related numeracy on health outcomes. However, evidence from disparate areas of psychology and health informatics suggests that the three main

variables, basic numeracy, COVID-19 health numeracy and COVID-19 health-related attitudes and behaviours are related.

First, there is a wealth of evidence that suggests that those with stronger basic numeracy are better equipped to apply this knowledge to diverse and novel contexts (Ballard & Johnson, 2004; Geary, 2011; Ludewig et al., 2019). Given that public information about COVID-19 disseminated in the media requires the understanding of numerous mathematical concepts that are typically not encountered in the day-to-day, it is likely that those in the population that possess stronger basic numeracy would have a stronger grasp of the risks relating to the pandemic. Second, there exist many studies on health literacy, which have examined the association between applied mathematical skills, such as health- and non-health-related word problems, and health-related decisions and outcomes. Those who perform better at these problems tend to have better health outcomes in the future (Brust-Renck et al., 2017; Weinfurt et al., 2003). Finally, stronger basic numeracy may have an independent effect on health outcomes controlling for health numeracy. At an equal level of health numeracy, those with stronger basic numeracy may be more capable of understanding the underlying numerical concepts of COVID-19 health information. As such, they would be better equipped in spotting common pitfalls to which those less familiar with the underlying numerical concepts may fall prey.

The current study directly addresses this critical knowledge gap by assessing the relation of both (A) basic numeracy and (B) COVID-19 health numeracy to (C) COVID-19 health-related attitudes and behaviours in a correlational study. Basic numeracy involves numerical magnitude processing and conceptual understanding of rational numbers (e.g., proportions). COVID-19 health numeracy is our measure of health numeracy relevant for understanding numerical information about COVID-19 (e.g., risk factors and flattening the curve). And lastly, COVID-19 health-related attitudes and behaviours indexes people's perception of and adherence to policies such as social distancing and handwashing. It is hypothesized that all three variables are positively related with one another. Specifically, that basic numeracy is positively correlated with COVID-19 health numeracy, that COVID-19 health numeracy is positively correlated with COVID-19 health-related attitudes and behaviours, and that basic numeracy is positively correlated with COVID-19 health-related attitudes and behaviours.

The primary aims of the current study are (1) to assess the interrelations among basic numeracy, health numeracy, and health-related attitudes and behaviours and (2) determine whether the relations among the three variables differ across three different countries: Canada, the United States, and the United Kingdom. These countries were chosen because (1) they are all members of the G7 and thus have comparable economies and political systems, and (2) the response to COVID-19 (e.g., when lockdown and re-opening measures were implemented) as well as the case fatality and testing rates differed between the three countries. Furthermore, the majority language for all three countries is the same (i.e., English), thereby limiting potential confounds driven by linguistic differences in number representation (Dowker & Nuerk, 2016). Investigating the between-country

differences in the strength of the correlations and between-country differences in how covariates may influence this relationship will allow us to comment on whether and how the strength of the relations among the variables differ across countries, which can inform short-term government policies related to COVID-19.

Methods

Ethics Statement

The study was approved by the non-medical research ethics board at the University of Western Ontario and will be conducted according to their guidelines. Participants will be presented with a letter of information and implied consent will be acquired before starting the survey. Qualtrics panelists join from a variety of sources. They may be airline customers who chose to join in reward for SkyMiles, retail customers who opted in to get points at their favorite retail outlet, or general consumers who participate for cash or gift cards, etc. When participants are invited to take a survey, they are informed what they will be compensated.

Procedure and Materials

In total, 2,025 participants will complete an online survey hosted by Qualtrics (see Power Analysis). Qualtrics will collect the data using their participant panels. The survey will take approximately 20 minutes to complete and will include four sections: demographics and other cultural variables, basic numeracy, COVID-19 health numeracy, COVID-19 health-related attitudes and behaviors. The survey will start with the consent form, followed by the demographics section for all respondents. The order of the other three sections (i.e., basic numeracy, COVID-19 health numeracy and COVID-19 health-related attitudes and behaviours) will be randomized across respondents (see Figure 2). The survey will end with a few questions about the device that they used, a seriousness check (where participants will be asked whether they would keep their data if they were the experimenter), and a question about any technical problem they may have been faced with. For a list of all items, see supplementary material (https://osf.io/qpdnt/?view_only=08ad84266beb41eaa5113d0acb2f890).

Figure 2. Overview of the flow of the survey

Demographics and Other Potential Moderators

Detailed demographic information will be collected using 14 items covering age, gender, country of residence, ethnicity, socioeconomic status (SES; Adler et al., 2000), employment status (adapted from the World Value Survey, WVS-6; Inglehart et al., 2014) and living situation. In addition to these demographic variables, a short list of potential moderating or confounding variables will be collected in order to test for their possible effect on variables of interest ~~in the mediation model~~, namely COVID-19 news consumption (three items), anxiety (three items), political leaning (one item). For behaviors related to news consumption, participants will indicate how often they access news sources (e.g., news websites, government communication) or social sources (e.g., social media, friends and family) for news on a scale of 1 ("daily") to 5 ("less than monthly") and how well informed they feel about the pandemic on a scale of 1 ("not informed") to 4 ("very informed"). The items measuring anxiety were adapted from the Single-Item Math Anxiety scale (Núñez-Peña et al., 2014) and will measure general anxiety (Davey et al., 2007), anxiety related to COVID-19, and math anxiety. Participants will rate their anxiety level on a scale of 1 ("not anxious") to 10 ("very anxious"). For political leaning, participants will be asked to rate their beliefs on a visual analog scale with "left/liberal" label at the left end and "right/conservative" label at the right end (adapted from the WVS-6; Inglehart et al., 2014). Finally, information about the type of electronic device used to complete the survey will be collected due to the nature of the response to the number line tasks. It is expected that participants will use either computers or mobile devices to respond to the questionnaire, meaning that the total length of the number lines may vary from one device to the other. Therefore, information about the device used to answer the

survey will be collected in order to test for differences in number line accuracy due to screen size. Demographics items will be presented in a fixed order as no order effect will be expected.

Basic Numeracy

The basic numeracy measure is designed to capture elements of numeracy that are relevant for understanding health information related to COVID-19 and includes a total of 30 items. It was created by adapting and combining four existing measurement tools focused on the understanding and processing of proportions and large numbers.

First, numerical magnitude processing will be measured using the number line task, a task that has been consistently found to be correlated with different mathematical skills (Schneider et al., 2018). In this task, the participant will be presented with an empty number line bound on each side. The participant will be given a number and asked to indicate its location on a number line with respect to the two endpoints. Five versions of the task will be used to assess participants' understanding of whole numbers (six items), fractions (six items), percentages (three items), large numbers (five items), and nonsymbolic ratios (three items). For all number line items, accuracy will be scored using percentage absolute error (PAE), which is the absolute difference between the estimated and the correct answer divided by the scale of the number line. Therefore, a higher PAE indicates lower accuracy (i.e., estimation is further away from true value), while a low PAE indicates higher accuracy. Numbers were selected because they showed suitable variability for capturing individual differences in previous research (Landy et al., 2013; Thompson, Taber, Sidney, et al., 2020). For all five number line versions, at least 3 of the items were matched for relative position on the number line to provide points of comparison across number line types.

Second, conceptual knowledge of fractions and decimals will be measured using two multiple-choice items and two fill-in-the-blank items selected from the Fraction-knowledge Assessment (Matthews et al., 2016) in addition to two open-ended items selected from the Rational Number Sense Test (Van Hoof et al., 2015). Whereas number line items assess the processing of specific magnitudes, these items will assess broader conceptual understanding of rational numbers such as fractions and decimals (e.g., "How many possible fractions are between $\frac{1}{4}$ and $\frac{1}{2}$?"). These six items will be scored correct (1) or incorrect (0). Lastly, the one-item version of the Berlin Numeracy test (Cokely et al., 2012) will be included in the basic numeracy measure. Although it has been proven as a valid and reliable test of risk literacy, it is often used as a measure of general numeracy and will be used to validate the novel basic numeracy measure described in this section. The item included in this survey has been shown to discriminate participants roughly into a top and bottom half of risk literacy in a number of samples across several countries (Cokely et al., 2012). The item will be scored correct (1) or incorrect (0).

Basic numeracy items will be presented pseudorandomly: the items will be randomly present within a group of similar items to reduce task switching. For example, all items consisting

of non-symbolic ratios will be presented on the same page in a random order and will not be mixed with other variants of the number line task. A complete list of items can be found at: https://osf.io/qpdnt/?view_only=08ad84266beb41ead5113d0acb2f890).

COVID-19 Health Numeracy

COVID-19 health numeracy will measure understanding of COVID-19 concepts and statistics using a custom-made questionnaire composed of 18 multiple-choice items. Items cover three concepts important to understanding COVID-19 related data in the media: graph literacy, proportions, and odds ratios. Fictional data was used in all sections in order to avoid bias due to prior COVID-19 knowledge. All items will consist of multiple choice items and be scored correct (1) or incorrect (0). Therefore, a higher score indicates high understanding of COVID-19 related concepts and statistics, whereas a lower score indicates poor understanding of COVID-19 related concepts and statistics.

Nine items will measure graph literacy. Of these, five items were conceptually modeled after the graph literacy scale proposed by Galesic and Garcia-Retamero (Galesic & Garcia-Retamero, 2011). Individuals will be shown a graph representing the total number of confirmed COVID-19 cases over time for two fictional countries, one with linear growth and the other with exponential growth. Four of these items will have one of three difficulty levels related to graph literacy. The first difficulty level consists of retrieving information, such as data points, directly from the graph (one item). The second difficulty level consists of evaluating relationships between data points, for example, by finding where two curves intersect each other (two items). The third level consists of making inferences from the information given by the graph, without the information being directly observable in the graph (one item). Also, one item will measure explicit knowledge of linear and exponential functions. The last four graph literacy items will test individuals' knowledge of "flattening the curve" and be accompanied by a diagram showing curves for the number of cases over time with and without social distancing. Questions will test participants' understanding of the effects of social distancing on various aspects of potential infection rate scenarios, such as length of outbreak and the height and latency of the peak of infection of the pandemic, based on the diagram.

Six items will measure the ability to accurately process absolute and relative magnitudes in the context of a highly infectious disease. A table containing fictional data about COVID-19 (i.e., number of confirmed cases, number of deaths, number of tests and total population) for three fictional countries will be presented. Similar to the section on graph literacy, items will have one of two difficulty levels. The first difficulty level will assess the ability to retrieve relevant data from the table (e.g., "Which country has conducted the most COVID-19 tests?"; three items). The second difficulty level will assess the ability to obtain relevant proportions from the data in order

to accurately compare country statistics (e.g., “Which country has the highest fatality rate for COVID-19?”; three items).

Finally, three items will measure participants’ ability to compare odds ratios. A situation describing fictional fatality rates for COVID-19 alone as well as for two other hypothetical risk factors will be presented using odds ratios. Items will involve comparing the odds ratios associated with the different risk factors and calculating relative risk.

Similar to items in the basic numeracy section, COVID-19 health numeracy items will be presented pseudorandomly. In other words, items within a subsection (e.g., items about flattening of the curve) will always be presented together in a random order to reduce task switching. For a list of items, see the supplementary material (https://osf.io/qpdnt/?view_only=08ad84266beb41eaa5113d0acb2f890).

Attitudes and Behaviors Towards COVID-19

Attitudes and behaviors towards COVID-19 will be measured using a total of 17 items. The first 12 items will focus on four frequently issued recommendations by health authorities (e.g., WHO): washing hands frequently and thoroughly, staying home unless the travel is essential, social distancing, and wearing a mask in public. For each recommendation, participants will be asked to indicate (1) whether or not this recommendation was issued by their local authorities, (2) when applicable, the extent to which they have followed the respective recommendation on a scale of 1 (“Never”) to 10 (“Consistently all the time”), and (3) how useful they think this recommendation is in the fight against the COVID-19 pandemic on a scale of 1 (“Completely useless”) to 10 (“Extremely useful”). Next, two items will measure participants' perceived change in (1) their own behavior and (2) people around them (e.g., family, close friends) in response to the COVID-19 pandemic. Participants will respond to both items on a scale of 1 (“Not at all”) to 10 (“To a great extent”). The last three items will assess participants’ perceptions about the severity and impact of the COVID-19 pandemic by assessing the degree to which they believe the COVID-19 pandemic is a serious global threat, COVID-19 is a serious medical condition, and that the benefits of the recommended actions to fight the COVID-19 pandemic outweigh their psychological, economic, and cultural costs. For these items, participants will respond on a scale of 1 (“Not at all”) to 10 (“To a great extent”).

Items of attitudes and behaviors towards COVID-19 will be presented in a fixed order to reduce bias in response selection as respondents might justify their responses about their own behaviors based on the local authorities' recommendations. For a complete list of items, see the supplementary material (https://osf.io/qpdnt/?view_only=08ad84266beb41eaa5113d0acb2f890).

Power Analysis

The current study focuses on the interrelations among three main constructs of interest, COVID-19 health numeracy, and COVID-19 health-related attitudes and behaviours. We are particularly interested in the robustness of these intercorrelations controlling for the covariates, and whether there may be between country differences in these intercorrelations. For the following power analysis, all α -levels are 0.05.

We calculated the necessary sample size to achieve 95% power using multiple Monte-Carlo simulation analyses in Mplus with 10,000 replications. As the relations between basic numeracy, COVID-19 health numeracy, and COVID-19 health-related attitudes and behaviours have yet to be explored, using effect sizes from previous literature for the purpose of power calculations would not be feasible. Most of the values in the following analyses were chosen based on reasonable expectation of correlations or minimum effect size of interest. However, in the multigroup analysis, we have decided on detecting a standardized correlation coefficient difference of 0.2 based on practical concerns, as the number of participants needed to detect a smaller difference would make the number of participants required prohibitively large.

Simple Correlation Model

To calculate the power to observe a correlation between the main variables of interest, we simulated a model with three variables with mean of 0 and variance of 1 and all intercorrelations set to be 0.14. A correlation coefficient of 0.14 corresponds to a small effect size (MacKinnon et al., 2004). A Monte-Carlo simulation revealed that a total of 675 participants would be sufficient to achieve 95% power for each regression coefficient.

Measurement Invariance of the Main Variables

As the current study operationalizes the main constructs as latent variables and we are employing multigroup analysis, measurement invariance between the countries must be considered. Measurement invariance consists of a number of steps that tests that different of the factor/latent variable structure is indifferent across countries. These include (1) configural invariance - the overall factor structure (i.e., the number of factors and their indicator variables), (2) metric invariance - the loadings of the indicator variables, (3) scalar invariance - the intercepts of the indicator variables, and (4) strict measurement invariance - the residual variances of the indicator variables (Putnick & Bornstein, 2016). The presence of invariance and non-invariance is often informative, and hypotheses were considered in table 1.

Calculations of power for measurement invariance are typically not calculated. This is because violations of different levels of measurement invariance are typically assessed using model fit indices (e.g., Comparative Fit Index and Root Mean Square Error of Approximation),

and these indices are indifferent to sample size. For all models below, a Monte-Carlo simulation study was conducted across two countries. For each country, we simulated three latent factors with mean of 0 and variance set to 1. Intercorrelations between all factors were set to 0.14. Each latent variable contained four indicators with factor loading of 0.5, residual of 0.75, and intercept of 3. A factor loading of 0.5 for each indicator variable can be interpreted as a standardized coefficient, which corresponds to a 25% explained variance and 75% residual variance.

Configural Invariance

To simulate a violation of configural invariance, one indicator item for factor 1 from the second country was instead loaded on to another with a factor loading of 0.5. Power was calculated as the proportion of simulations in which the loading of the errant indicator item was not statistically significant. Results indicated 675 participants would be sufficient for 95% power to detect a violation.

Metric Invariance

To simulate a violation of metric invariance, one item each from factor 2 and factor 3 was loaded on their respective factors at 0.2 instead of 0.5 (see Table 1 for hypotheses). Power was calculated as the proportion of simulations in which the chi-square difference test between the configural invariant and metric invariant model was significant. Results indicate 675 participants would be sufficient for 95% power to detect a violation.

Scalar Invariance

To simulate a violation of scalar invariance, one item intercept from factor 1 was set to 4 instead of 3. Power was calculated as the proportion of simulations in which the chi-square difference test between the metric invariant and scalar invariant model was significant. Results indicate 675 participants would be sufficient for 95% power to detect a violation.

Latent Mean Differences

To simulate latent mean differences, the latent mean of the second group was set to be 0.3 - a small to medium effect size (Cohen, 2013). Power was calculated as the proportion of simulations in which a chi-square difference test was obtained when testing between a model where latent means were restricted to be the same and another model where latent means were allowed to be estimated freely.

Multigroup Analysis

We simulated the analyses that will be used to address whether there are country-level differences in the correlation between the main variables by modelling a multigroup analysis. Using the simple correlation model above, we stipulated a model with same number of participants in each country. Specifically, we aim to detect between-country differences in the intercorrelation.

The comparisons were carried out pairwise using the MODEL CONSTRAINT command in Mplus. To calculate power, we simulated a multigroup analysis with two countries, and set to detect a difference in correlation of 0.2 - a small to medium effect size (Cohen, 2013). Monte-Carlo simulations revealed that 675 participants per country would be needed to achieve 95% power.

In sum, to achieve 95% power for all three analyses, a total of 675 participants each across the United States, Canada and the United Kingdom (total sample 2025 participants) would be required. For data generation code, simulated data, and analysis code, see the supplementary material (https://osf.io/qpdnt/?view_only=08ad84266beb41eaa5113d0acb2f890).

Preprocessing of Data

Exclusion Criteria

Due to the nature of the online sampling, we will exclude participants who are inattentive. Two types of inattentiveness are commonly observed, one being general inattentiveness and the other being marked by frequently selecting the same answer for entire blocks and completing a survey in short periods of time (Meade & Craig, 2012). We will use several criteria to exclude inattentive participants (Huber et al., 2019). First, we will remove respondents with multiple submissions, who respond multiple times using an IP check (Reips, 2000). Second, we will include a seriousness check (Aust et al., 2013). Data for participants indicating they were not serious will be discarded. Third, three instructed response items adapted from Barends and de Vries (Barends & de Vries, 2019) will be asked on different steps of the survey (For a list of three items, see supplementary material (https://osf.io/qpdnt/?view_only=08ad84266beb41eaa5113d0acb2f890)). Data for participants who respond incorrectly to more than one of these three questions will be discarded. Fourth, participants with more than 25% missing data from selecting “prefer not to answer” will be discarded. **Finally Fifth**, participants whose survey completion time is less than half of the 5% trimmed mean of the complete sample of participants would be flagged as potentially not paying attention, and responses will be examined for inconsistencies (Maniaci & Rogge, 2014). **Finally, to ensure that participants have not been influenced by a reception of a COVID-19 vaccine or the participation in a COVID-19 vaccination trial, we included two questions inquiring whether participants have received a COVID-19 vaccine or participated in a COVID-19 vaccination trial. Participants who have answered in the affirmative for either question will be excluded.**

Outlier Identification and Missing Data

Univariate outliers will be detected using the absolute deviation around the median, where continuous values ± 3 Median Absolute Deviation (MAD) will be considered outliers (Leys et al., 2013). Multivariate outliers will be detected using Mahalanobis distance test with a $p < .001$

(Tabachnick et al., 2007). Most questions in the survey require rating-scale responses that have well-defined ranges, and the removal of outliers for these questions may underestimate actual variability in the population. Therefore, for rating-scale questions, identified outliers will be examined for obvious errors or participant inattention. If no evidence for errors or inattention is found, the value will be retained in the final dataset. For questions without a well-defined range (i.e., the number line questions in the basic numeracy section) identified outliers will be examined for errors. If no errors are found, the outliers will be treated as missing data. **Finally, for the set of questions in the number line task, we will calculate the median absolute deviation of each participants' responses and will exclude participants if the median absolute deviation of their responses is below 0.1 (please see Preliminary Measurement Study for more details).**

We will conduct a missing values analysis to identify patterns of missing data in the items. We anticipate two patterns that often occur in online research. The first consists of missing random responses throughout the questionnaire, due either to inattention, or to participants not wanting to answer specific items. We will analyze frequency of non-response for each item and identify items with higher non-response frequency. These items are not missing at random, and we will determine if they can be explained by covariates included in the current study, at which point, we will include the covariates either in the model or as auxiliary variables that can be used to improve the estimation of missing data in full information maximum likelihood (FIML).

The second pattern of missingness that we anticipate is attrition throughout the questionnaire. Respondents who drop-off at different points along the questionnaire may differ from those who do not in three potentially different ways (i.e., less motivation, having restricted amounts of uninterrupted time available to complete the survey, lower overall math skills). We will be able to analyze this particular pattern by testing whether missingness at the later points is related to performance on earlier items. Regardless of the in-depth missingness analyses, FIML (Muthén & Muthén, 2010) is still the best approach, and no cases will be left out except when careless responding has been identified.

Analysis Plan

The main hypotheses of the current study are concerned with the interrelations between three hypothesized latent variables: basic numeracy, COVID-19 health numeracy, and COVID-19 health-related attitudes and behaviours. To assess these interrelations, several measures were newly developed for the current study. As a result, there is a large degree of uncertainty regarding whether the data would fit both the hypothesized measurement model (i.e., the relation between the latent factors and their indicators) and the structural model (i.e., the relation between latent factors). Given the unpredictable nature of the measures, it may be necessary to make substantial changes to the hypothesized statistical model post data collection. This is not ideal as it may increase the risk of overfitting the data. To alleviate this risk, we will conduct a preliminary measurement study of the variables described above with the primary purpose of assessing item

quality and the measurement model. The data from the pilot study will be used to inform substantive decisions regarding the measurement model that will be submitted as part of the stage 1 revision of the registered report before main data collection (see figure 3 and figure 4 for a decision tree for the pilot study and main analysis).

Figure 3. Decision Tree to Analyze the Preliminary Measurement Study Data.

Figure 4. Decision Tree to Analyze the Main Study Data.

Reliability and Validity

Subsequent to estimating the EFA and CFA in both the preliminary measurement study and main study (see below), **reliability** – the degree to which indicators of the latent factor consistently measure the same underlying construct – will be assessed using coefficient omega for each latent variable estimated (McDonald, 2013). We will use the typical cut-off point for acceptable reliability of 0.7 for each of the three latent variables (Kline, 2015; McDonald, 2013). Once reliability is established, convergent validity – the degree to which individual items are strongly related to the hypothesized factor – will be assessed using average variance extracted (AVE). We will use the typical cutoff score for acceptable convergent validity, where AVE for each of the three factor should be higher than 0.5. Finally, discriminant validity – the degree to which the degree to which individual items are not related to other factors – will be assessed using the Fornell & Larcker criterion (Fornell & Larcker, 1981; Hair et al., 2016). Specifically, we will adopt the convention that the square root of the AVE for each construct should be larger than the correlation of the specific construct with any of the other construct.

Preliminary Measurement Study

Data from 175 participants from each of the three countries of interest (totalling 525 participants) were collected. Preprocessing of data was conducted in the same manner as proposed for the main data set (see **Preprocessing of Data**). Participant exclusion criteria outline above were examined and judged to be adequate. However, in an examination of participants' responses for the number line task, we found some participants who answered the number line questions by placing their answers on the same point on the number line irrespective of the question prompt. To remove these non-responsive participants, we have added an additional exclusion criterion for the processing of the main data. Specifically, for the set of questions in the number line task, we will calculate the median absolute deviation of each participants' responses and will exclude participants if the median absolute deviation of their responses is below 0.1. In total, 79 participants were removed, which amount to a 15% exclusion rate. Therefore, for the main data analysis, we will collect an additional 15% participants from each country to reach the desired total participant of 675.

Table 1

Sample Statistics for All Items

Continuous Variables	Mean	S.D.	Skew	Kurtosis
Basic Numeracy Number line Q1	0.080	0.199	2.284	4.629
Basic Numeracy Number line Q2	0.005	0.134	1.181	3.007
Basic Numeracy Number line Q3	-0.079	0.188	-2.028	3.949
Basic Numeracy Number line Q4	0.081	0.166	2.753	8.335
Basic Numeracy Number line Q5	0.027	0.132	0.916	2.292

Basic Numeracy Number line Q6	-0.046	0.128	-2.783	10.446
Basic Numeracy Number line Q7	0.174	0.231	0.577	-0.907
Basic Numeracy Number line Q8	0.090	0.242	1.021	-0.067
Basic Numeracy Number line Q9	-0.102	0.192	-1.331	1.226
Basic Numeracy Number line Q10	0.179	0.230	0.529	-0.821
Basic Numeracy Number line Q11	0.105	0.228	0.797	-0.445
Basic Numeracy Number line Q12	-0.120	0.190	-1.147	0.841
Basic Numeracy Number line Q13	0.024	0.154	2.495	6.238
Basic Numeracy Number line Q14	-0.054	0.105	1.897	10.228
Basic Numeracy Number line Q15	-0.070	0.155	-2.381	5.939
Basic Numeracy Number line Q16	0.197	0.222	1.195	0.671
Basic Numeracy Number line Q17	0.104	0.174	1.208	1.083
Basic Numeracy Number line Q18	-0.061	0.107	-1.701	4.399
Basic Numeracy Number line Q19	0.247	0.200	1.122	0.916
Basic Numeracy Number line Q20	0.235	0.226	1.098	0.168
Basic Numeracy Number line Q21	0.119	0.233	1.247	0.625
Basic Numeracy Number line Q22	0.033	0.195	0.896	0.266
Basic Numeracy Number line Q23	-0.133	0.169	-0.902	0.505
Basic Numeracy Number line Q24	0.080	0.199	2.284	4.629
Attitudes and Behaviors Q1	8.808	1.437	-1.542	2.948
Attitudes and Behaviors Q2	8.360	1.876	-1.536	2.521
Attitudes and Behaviors Q3	8.924	1.512	-2.081	5.035
Attitudes and Behaviors Q4	9.309	1.444	-3.198	12.461
Attitudes and Behaviors Q5	9.067	1.433	-2.065	5.377
Attitudes and Behaviors Q6	8.868	1.647	-1.972	4.719
Attitudes and Behaviors Q7	9.094	1.456	-2.370	7.367
Attitudes and Behaviors Q8	8.982	1.733	-2.393	6.249
Attitudes and Behaviors Q9	8.622	1.707	-1.662	3.181
Attitudes and Behaviors Q10	8.425	1.699	-1.376	2.275
Attitudes and Behaviors Q11	8.859	1.797	-2.042	4.203
Attitudes and Behaviors Q12	8.805	1.760	-1.838	3.376
Attitudes and Behaviors Q13	8.186	2.257	-1.564	1.977

Categorical Variables

	% Correct	% Incorrect
Basic Numeracy Q1	0.736	0.264
Basic Numeracy Q2	0.112	0.888
Basic Numeracy Q3	0.670	0.330
Basic Numeracy Q4	0.456	0.544
Basic Numeracy Q5	0.592	0.408
Basic Numeracy Q6	0.800	0.200
Basic Numeracy Q7	0.153	0.847
COVID-19 Health Numeracy Q1	0.785	0.215
COVID-19 Health Numeracy Q2	0.819	0.181
COVID-19 Health Numeracy Q3	0.600	0.400
COVID-19 Health Numeracy Q4	0.472	0.528
COVID-19 Health Numeracy Q5	0.391	0.609

COVID-19 Health Numeracy Q6	0.671	0.329
COVID-19 Health Numeracy Q7	0.817	0.183
COVID-19 Health Numeracy Q8	0.441	0.559
COVID-19 Health Numeracy Q9	0.452	0.548
COVID-19 Health Numeracy Q10	0.843	0.157
COVID-19 Health Numeracy Q11	0.414	0.586
COVID-19 Health Numeracy Q12	0.494	0.506
COVID-19 Health Numeracy Q13	0.367	0.633
COVID-19 Health Numeracy Q14	0.805	0.195
COVID-19 Health Numeracy Q15	0.606	0.394
COVID-19 Health Numeracy Q16	0.609	0.391
COVID-19 Health Numeracy Q17	0.483	0.517
COVID-19 Health Numeracy Q18	0.468	0.532
Attitudes and Behaviors Q14	0.971	0.029
Attitudes and Behaviors Q15	0.960	0.040
Attitudes and Behaviors Q16	0.991	0.009
Attitudes and Behaviors Q17	0.989	0.011

Sample statistics are presented in table 1. Examination of the skewness and kurtosis of the continuous variables that most items modestly deviated from normality (Skewness < 2, Kurtosis < 7; Finch, West, & MacKinnon, 1997). However, all items from the COVID-19 Attitudes and Behaviors scale have very high means and negative skewness, indicating a ceiling effect is present. As this ceiling effect is observed for all items administered and that multiple other research studies with similar scales also yielded similar censored results (e.g., Czeisler et al., 2020; Wolf et al., 2020), we believe that this ceiling effect reflects the underlying population distribution. To account for this ceiling effect, all items from the COVID-19 Attitudes and Behaviors scale were treated as ordered categorical variables for the following analysis. Examination of the dichotomous variables suggest a good distribution of difficulty. However, 4 items from the COVID-19 Attitudes and Behaviors (the items asking whether different safety recommendation were issued by their local authorities) lacked variability, as most participants have responded in the affirmative. Therefore, we have removed these four items from analysis.

We next conducted separate EFAs for the set of items in each of the three main variables of interest. As there are multiple categorical variables, the WLSMV estimator and goemin rotation was used. Examination of the Scree plot suggests that a one factor solution would fit for each of the three variables. For the basic numeracy factor, factor loadings ranged from 0.320 to 0.863, for COVID-19 health numeracy, factor loadings ranged from 0.404 to 0.759, and for COVID-19 Attitudes and Behaviors, factor loadings ranged from 0.708 to 0.910. Aggregate scores of Basic numeracy and COVID-19 health numeracy are correlated at 0.778, basic numeracy and COVID-19 Attitudes and Behaviors are correlated at -0.113, and COVID-19 health numeracy and COVID-19 Attitudes and Behaviors are correlated at -0.090.

Reliability as measured by coefficient omega for basic numeracy, COVID-19 health numeracy, and COVID-19 Attitudes and Behaviors were 0.934, 0.901, and 0.960, respectively. As the coefficient omega for each latent variable is above 0.7, this suggest that the individual items have a satisfactorily level of internal consistency. However, average variance extracted (AVE) for basic numeracy, COVID-19 health numeracy, and COVID-19 Attitudes and Behaviors were 0.330, 0.370 and 0.638, respectively. This suggest that convergent validity was not established for basic numeracy and COVID-19 health numeracy. Similarly, AVE values for basic numeracy and COVID-19 health numeracy are lower than the squared correlations between the variables, suggesting discriminant validity was not established. In sum, these results suggest that a measurement model with individual items serving as indicators for the latent variables is not satisfactory.

We next proceeded to constructing parcels. Given that the EFAs suggest a one factor solution for each of the main variables, we proceeded with constructing parcels using the balancing approach, where the three items with strongest item-scale correlation are paired with the three items with the weakest item-scale correlation to form 3 parcels. Additional items are assigned to one of the three parcels successively, alternating directions through the parcels, until all items are assigned. For example, in the case of 12 indicators ranked by item-scale correlation, parcel#1 = 1, 6, 7, 12; parcel#2 = 2, 5, 8, 11; and parcel#3 = 3, 4, 9, 10 (Rogers & Schmitt, 2004). Three parcels were constructed for each latent variable. Model fit for the CFA was good ($\chi^2(24) = 33.54$, $p = 0.093$, CFI = 0.997, TLI = 0.995, RMSEA = 0.030).

For the basic numeracy factor, factor loadings ranged from 0.844 to 0.899, for COVID-19 health numeracy, factor loadings ranged from 0.731 to 0.776, and for COVID-19 Attitudes and Behaviors, factor loadings ranged from 0.913 to 0.950. Basic numeracy and COVID-19 health numeracy are correlated at 0.787, basic numeracy and COVID-19 Attitudes and Behaviors are correlated at -0.091, and COVID-19 health numeracy and COVID-19 Attitudes and Behaviors are correlated at -0.080.

Reliability as measured by coefficient omega for basic numeracy, COVID-19 health numeracy, and COVID-19 Attitudes and Behaviors were 0.906, 0.799, and 0.952, respectively. As the loadings were high and the coefficient omega for each latent variable is above 0.7, this suggest that the latent variables have high internal consistency. AVE for basic numeracy, COVID-19 health numeracy, and COVID-19 Attitudes and Behaviors were 0.764, 0.571 and 0.869, respectively. As the AVEs of the three latent variables are above 0.5, this suggest that convergent validity was established for basic numeracy and COVID-19 health numeracy. Similarly, the square root of the AVE for each construct is larger than the correlation of the specific construct with any of the other construct. This suggest that discriminate validity was established. In sum, these results suggest that a measurement model with parcels serving as indicators for the latent variables is satisfactory.

In light of these results from the preliminary measurement study, we will take the following actions for the final data collection: 1) we will add an additional participant exclusion criterion that removes participants when there are clear evidence of non-responsiveness, 2) we will remove 4 items from the COVID-19 Attitudes and Behaviors (the items asking whether different safety recommendation were issued by their local authorities) from the main analysis, and 3) we will proceed with analyzing the data using a CFA with parcels employing the balancing approach.

Item Parcels

The current study employs a large number of items in the measurement of the main variables. Operationalizing latent variables with a large number of indicator items is not ideal from a psychometric perspective. This is because individual items tend to be statistically less reliable than aggregates and have a larger likelihood to have correlated specific effects may subject the final model to nuisance parameters (i.e., correlated residuals; Little et al., 2002). To address this, we will employ an item parcel approach to form a smaller set of composite indicator items to inform the latent variables.

Given that the current survey employs multiple question formats (e.g., multiple choice, fill-in-the-blank, and number line) a random parceling approach would not be ideal, as random parceling assumptions that all items are interchangeable. Results from the preliminary measurement study have shown that each of the hypothesized latent variables are unidimensional. As such, parcels will be assigned using a balancing approach.

Main Analyses

After ensuring data quality with participant and outlier exclusion, we will pursue the research questions using a structural equation modelling approach. Basic numeracy, COVID-19 health numeracy, and COVID-19 health-related attitudes and behaviours will be operationalized as latent variables. All analyses will be carried out using Mplus 8.3 using the maximum likelihood estimator with robust standard errors (Muthén & Muthén, 2010). Where applicable, goodness of fit of the models will be assessed with χ^2 test statistic, the Comparative Fit Index (CFI), the Tucker-Lewis Index (TLI), and the root mean square error of approximation (RMSEA). Typical cut-off scores for excellent and adequate fit are CFI and TLI > 0.95 and > 0.90, and RMSEA < 0.06 and < 0.08, respectively (Hu & Bentler, 1999). In the case of inadequate fit (i.e., CFI and TLI < 0.9, RMSEA > 0.08), modification indices will be consulted for correlated residuals and other sources of misfit. Average

Simple Correlation Model

We will first examine the associations between the three main variables in a simple correlation model. As a first step, a model with only the three main (latent) variables would be

entered. As a second step, the robustness of the observed correlations will be tested by entering covariates as predictors of the three main variables.

Invariance Testing

The main objective of this research will be to investigate differences across countries in the means and variation of the main variables in the model (basic numeracy, COVID-19 health numeracy, and COVID-19 health-related attitudes and behaviors). This will include multiple-group measurement invariance analyses of the individual latent measures. This procedure is usually performed as a step to establish factorial validity across groups. Once established, it becomes possible to evaluate differences in latent means, variances, covariances, or specific correlations across different countries.

We will use the standard procedures in the confirmatory factor analytic literature for evaluating measurement invariance across countries (Kline, 2015). In short, this consists of a number of steps testing the invariance of different parts of the factor/latent variable structure. These include (1) configural invariance - the overall factor structure (i.e., the number of factors and their indicator variables), (2) metric invariance - the loadings of the indicator variables, and (3) scalar invariance - the intercepts of the indicator variables (Putnick & Bornstein, 2016). Strict measurement invariance is not required for the following analyses and will not be tested. These are tested in incremental stages, and the first three stages are essential to proceed with tests of latent means. Configural invariance will be assessed using model fit, where CFI and TLI > 0.90 and RMSEA < 0.08 is generally accepted as adequate model fit (Hu & Bentler, 1999). Metric and scalar invariance are evaluated using the chi-square difference test and the degradation of fit in CFI and RMSEA. Specifically, we will use the criterion of $\Delta\text{RMSEA} < .015$ and $\Delta\text{CFI} < -0.01$ (Chen, 2007; Cheung & Rensvold, 2002).

In the case where measurement invariance is not achieved, we will proceed with partial-invariance tests and identify the source of non-invariance. Some degree of researcher judgement would be required to gauge the severity of the invariance between countries. In the event that we do not meet configural invariance, modification indices will be examined for the source of the misfit. If the degree of misfit is small, the removal of the problematic indicator item could be a potential solution. This may necessitate the examination of between-country differences in the individual items constituting the parcel to find problematic items. If the degree of invariance is high, it may be necessary to estimate separate models for each country. In the event that we do not meet metric or scalar invariance in a given country, invariant items will be identified using the forward method (Jung & Yoon, 2016), effect size of invariance (d_{MACS}) will be calculated to gauge the magnitude of invariance (Nye & Drasgow, 2011). Identified invariant items will be allowed to vary across countries. We will proceed with partial scalar or metric invariance if less than half of the indicators are not invariance (Steenkamp & Baumgartner, 1998; Vandenberg & Lance, 2000). Lack of invariance in the factor structure is often seen as a nuisance, but it can also advance our

understanding of cross-cultural differences in how concepts are conceptualized. For example, if results indicate that the sole source of non-invariance in the latent variable COVID-19 health-related attitudes and behaviors is the factor loading of mask-wearing frequency, it can be interpreted that the amount of variance of mask wearing frequency that COVID-19 health-related attitudes and behaviors can explain is different between countries.

Finally, in the case where scalar invariance, latent variance equality and path equality is achieved (Vandenberg & Lance, 2000), latent mean differences between countries will be examined.

Multigroup Analysis

Once metric or scalar invariance or partial metric or scalar invariance is achieved, between-country differences in latent-correlations will be examined in two steps. As a first step, we will formulate a model with only the three main variables and examine the intercorrelations. As a second step, the robustness of the observed intercorrelations will be tested by entering covariates as predictors of the three main variables. Between-country differences in the correlations will be examined pairwise using the MODEL CONSTRAINT command in Mplus.

Exploratory Analyses

Potential Interactions with Covariates

While tangential to our main research questions, multiple covariates included in our analyses are theoretically interesting and may greatly influence the main variables in the study. Unfortunately, how these influences would take place are difficult to delineate a-priori. Therefore, we will explore these potential covariates in the following additional analyses. *All exploratory analysis will be examined, but only reported if statistically significant and theoretically interesting results are found.*

Political Leaning. One trend that can be observed is that opinions regarding the appropriate current and future community and personal actions taken in response to the pandemic differ depending on the political ideology to which citizens belong (Allcott et al., 2020). This suggests that the relation between COVID-19 health numeracy and COVID-19 health-related attitudes and behaviors may differ depending on political ideology. *We will explore this possibility by including an interaction term as a predictor of COVID-19 health numeracy and COVID-19 health-related attitudes and behaviors. Specifically, a COVID-19 health numeracy x Political leaning variable will be entered as a predictor of COVID-19 health-related attitudes and behaviors, and a COVID-19 health-related attitudes and behaviors x Political leaning variable will be entered as a predictor of COVID-19 health numeracy. In such a case, a significant coefficient between the interaction terms and COVID-19 health numeracy or COVID-19 health-related attitudes and behaviors would suggest that there is a significant interaction.*

COVID-19 Related Anxiety. While few studies have examined whether COVID-19-related anxiety is distinct from general anxiety, it is likely that the novelty and prevalence of COVID-19 would inspire differing emotional reactions that are at least somewhat distinct from general anxiety. Different individuals with comparable numeracy may nevertheless perceive the magnitude of the threat pertaining to the same information differently depending on COVID-19 related anxiety. Therefore, it is hypothesized that the relationship between COVID-19 health numeracy and COVID-19 health attitudes and behaviors may differ depending on COVID-19-related anxiety. We will explore this possibility by including an interaction term as a predictor of COVID-19 health numeracy and COVID-19 health-related attitudes and behaviors. Specifically, a COVID-19 health numeracy x COVID-19-related anxiety variable will be entered as a predictor of COVID-19 health-related attitudes and behaviors, and a COVID-19 health-related attitudes and behaviors x COVID-19-related anxiety variable will be entered as a predictor of COVID-19 health numeracy. In such a case, a significant coefficient between the interaction terms and COVID-19 health numeracy or COVID-19 health-related attitudes and behaviors would suggest that there is a significant interaction.

Gender. COVID-19 health numeracy and COVID-19 health attitudes and behaviors and attitudes may differ as a function of a participants' gender (Halpern et al., 2007; Lusardi, 2012). As such, the effects of gender will be assessed by entering the variable as a covariate for all three paths. Specifically, a basic numeracy x gender interaction term will be entered as a predictor of COVID-19 health numeracy and COVID-19 health attitudes and behaviors, a COVID-19 health numeracy x gender interaction term will be entered as a predictor of basic numeracy and COVID-19 health attitudes and behaviors, and a COVID-19 health attitudes and behaviors x gender interaction term will be entered as a predictor of basic numeracy and COVID-19 health numeracy. In such a case, a significant coefficient between the interaction terms and any of the variables of interest would indicate that there is a significant gender interaction.

Participant Exclusion Criteria and Basic Numeracy

One potential concern regarding the measurement of basic numeracy is that participants may have some trouble understanding and completing the number line question. Specifically, if participants failed to understand or misunderstood the instructions to the number line questions, their performance for the number line task would substantially differ from their performance in the non-number line basic numeracy questions. Three potential exclusion criterion may be capturing the failure to understand the number line question instruction: 1) the criterion that excludes participants with more than 25% missing data (as one can reason that participants may omit answering the number line questions if they do not understand the instructions), 2) the criterion that excludes participants based on Mahalanobis distance (as participants with a discrepancy between number line question and non-number line question performance would likely be flagged as a multivariate outlier), and 3) the criterion that excludes participants whose median absolute deviation of the number line task is below 0.1 (as participants who did not

understand the instructions for the number line task may “respond” by selecting the same point on the number line as a non-response).

This hypothesis will be explored by regressing the mean-score of the non-number line numeracy questions on three dichotomous variables each representing one of the exclusion criteria mentioned above. If a significant relation is obtained, it would indicate that indeed the participant exclusion criterion truncated the lower end of basic numerical abilities. We will account for this possibility in two ways. First, we will re-run the main analysis without the statistically significant exclusion criterion. Any discrepancy between this new analysis and the main analysis will be reported. Second, we will alternatively operationalize the basic numeracy latent factor without the number line questions and report any discrepancy found between the new analysis and the main analysis.

Additional Exploratory Variables

Due to the fast-changing nature of the COVID-19 pandemic, additional exploratory variables were added to the current study as part of the stage 1 registered report revisions. Specifically, we have added a number of items from the COVID States Project (Lazer et al., 2020). We have added two general questions regarding participants’ recent stay-at-home behaviors. The first item asks, “In the last 24 hours, how many people did you meet with that are not part of your immediate household?” Participants respond with 0, 1, 2, 3, 4-9, and > 9 as choices. The second item asks, “In the last 24 hours, have you been in a room (or another enclosed space) with people who were not members of your household? This might have been at a social gathering, a work meeting, or another type of even.” Participants are given “No, I have not,” “Yes, with 1-2 other people,” “Yes, with 3-4 other people,” “Yes, with 5-6 other people,” “Yes, with 7-8 other people,” “Yes, with 9-10 other people,” “Yes, with 11-50 other people,” “Yes, with over 100 other people,” as choices. Further, we have added a battery of yes/no questions asking, “In the last 24 hours, did you or any members of your household do any of the following activities outside of your home? Activities include going to work, going to the gym, visiting a friend, going to the doctor or visiting a hospital, going to a café, bar or restaurant, going to church or another place of worship, and taking mass transit. These items will be combined with our existing survey items about social distancing to create a social distancing index similar to Lazer et al. (2020).

Table 1. Design Table

Question	Hypothesis	Sampling Plan (power analysis)	Analysis Plan	Interpretation given to Different outcomes
Whether basic numeracy, COVID-19 health numeracy, and COVID-19 health-related attitudes and behaviours are intercorrelated.	Hyp. 1. Basic numeracy is positively correlated with COVID-19 health numeracy	A Monte-Carlo study was conducted by specifying the smallest effect size of interest for the relationship the two variables and determining how many participants are required to detect a statistically significant effect for path a 95% of the time out of 10,000 replications.	First, the three main variables will be entered in an SEM model to examine the intercorrelations. Next, the robustness of these correlations will be examined by entering covariates as regressors for the main variables.	Significant positive correlation: participants who score higher on the basic numeracy measures generally score higher for the COVID-19 health numeracy measures Significant negative correlation: participants who score higher on the basic numeracy measures generally score lower for the COVID-19 health numeracy measures Non-significant coefficient: Undetermined relationship between the variables
	Hyp. 2. Basic numeracy is positively correlated with COVID-19 health-related attitudes and behaviours	A Monte-Carlo study was conducted by specifying the smallest effect size of interest for the relationship the two variables and determining how many participants are required to detect a statistically significant effect for path a 95% of the time out of 10,000 replications.	First, the three main variables will be entered in an SEM model to examine the intercorrelations. Next, the robustness of these correlations will be examined by entering covariates as regressors for the main variables.	Significant positive correlation: participants who score higher on the basic numeracy measures generally score higher for the COVID-19 health-related attitudes Significant negative correlation: participants who score higher on the basic measures generally score lower for the COVID-19 health-related attitudes and behaviours measures Insignificant coefficient: Undetermined relationship between the variables
	Hyp. 3. COVID-19 health numeracy is positively correlated with COVID-19 health-related attitudes and behaviours	A Monte-Carlo study was conducted by specifying the smallest effect size of interest for the relationship the two variables and determining how many participants are required to detect a statistically significant effect for path a	First, the three main variables will be entered in an SEM model to examine the intercorrelations. Next, the robustness of these correlations will be examined by entering covariates as regressors for the main variables.	Significant positive correlation: participants who score higher on the COVID-19 health numeracy measures generally score higher for the COVID-19 health-related attitudes Significant negative correlation: participants who score higher on COVID-19 health numeracy measures generally score

		95% of the time out of 10,000 replications.		lower for the COVID-19 health-related attitudes and behaviours measures
Are the measurement properties of the constructs invariant across countries?	Invariance of the measurement properties. The factor structure, factor loadings, intercepts, and residuals of indicator items for the latent variables of basic numeracy will be fully invariant across countries. The factor structure, factor loadings, intercepts, and residuals of indicator items for the latent variables of COVID-19 health numeracy and COVID-19 health-related attitudes and behaviours may be partially non-invariant across countries.	A Monte-Carlo study was used to simulate violations of configural and metric invariance in accordance with the hypotheses.	Standard procedure to test for between country invariance will be used. Increasingly stringent between-country constraints will be placed on the analytical structure. These stages include:  1) Configural Invariance - Restricting the overall factor structure (i.e., the number of factors and their indicator variables) to be the same between countries 2) Metric Invariance - Restricting the loadings of the indicator variables to be the same between countries 3) Scalar Invariance - restricting the intercepts of the indicator variables to be the same between countries Constraints are evaluated using the chi-square difference test, and differences in other fit indices (CFI and RMSEA) whereby the less restrictive model is iteratively compared with the more restrictive model. A significant chi-square difference test would indicate that the added restrictions significantly reduce model fit.	Insignificant coefficient: Undetermined relationship between the variables If configural invariance is not achieved, one or more of the indicator variables do not indicate for the same latent variable. As such, one may need to redefine the latent variables, or countries will need to be analyzed separately and between-country differences cannot be examined. If configural invariance is achieved, it suggests that the same latent construct could be constructed with the same manifest variables across groups. If metric invariance is achieved, it suggests the measurement properties of the latent variables are comparable between countries. This will allow for the examination of differences in paths A, B and C' between countries. If scalar invariance is achieved, it suggests that mean differences in the latent construct are based on an equivalent scale and allow for the examination of latent mean differences between countries

Do the parameter estimates in the mediation model differ across countries?	We expect that there may be between-country differences in the strength of association between the main variables.	A Monte-Carlo study was conducted by specifying the smallest effect size of interest for the between country differences in intercorrelations.	In the event that some indicators are found to be non-invariant, we will test for partial invariance or remove the items prior to analysis of the full mediation model.	A significant difference between countries: the link between the latent variables are stronger for one country as compared to another. A non-significant difference between countries: insufficient evidence to suggest there are differences between countries.
	There may be mean differences between the latent factors	A Monte-Carlo study was used to simulate latent mean differences.	A model with latent means for both countries fixed to be equal will be compared with another model with latent means fixed to 0 for one country and freely estimated in the other. Constraints are evaluated using the chi-square difference test, and differences in other fit indices (CFI and RMSEA) whereby the less restrictive model is iteratively compared with the more restrictive model. A significant chi-square difference test would indicate that the added restrictions significantly reduce model fit.	Significant mean differences between different countries: The average level of basic numeracy, COVID-19 health numeracy, or COVID-19 health-related attitudes and behaviours are different between countries Non-significant mean differences between different countries: Insufficient evidence to suggest there are differences in the average level of basic numeracy, COVID-19 health numeracy, or COVID-19 health-related attitudes and behaviours

Data availability

All raw data (including pilot data) will be made available on the project's OSF page (https://osf.io/qpdnt/?view_only=08ad84266beb41ead5113d0acb2f890) upon acceptance of the Stage 2 manuscript.

Code availability

All code will be made available on the project's OSF page (https://osf.io/qpdnt/?view_only=08ad84266beb41ead5113d0acb2f890) upon acceptance of the Stage 2 manuscript.

Acknowledgements

The cost of the data collection will be supported by grants from the Canadian Institutes for Advanced Research as well as the Canada Research Chairs Program to DA. CAT is supported in part by a U.S. Department of Education Institute of Education Sciences (IES) Grant R305A160295 at Kent State University. EDW is the recipient of a Banting Postdoctoral Fellowship (NSERC) and a BrainsCAN Postdoctoral Fellowship at Western University, funded by the Canada First Research Excellence Fund (CFREF). MS is the recipient of a BrainsCAN Postdoctoral Fellowship at Western University, funded by the Canada First Research Excellence Fund (CFREF). LP is supported by a Brain and Mind Institute Postdoctoral Fellowship at Western University and by a Postdoctoral Trainee Award from the Children's Health Research Institute. JVH is supported by a postdoctoral fellowship from the Research Foundation-Flanders (FWO). ISA is supported by CAPES (Doc-Pleno, 88881.128282/2016-01). IC is supported by a Postdoctoral fellowship funded by Agence Nationale de la Recherche (ANR-18-CE28-0003). AR is supported by an NIH NICHD F32 award (HD102106-01) as well as a Scholar Award from the James S. McDonnell Foundation to Dr. Melissa Libertus. We thank Bea Goffin for comments on drafts of the manuscript.

Author contributions

The Authors' contribution was calculated according to the CRediT, the Contributor Roles Taxonomy, which can be found here:

https://osf.io/qpdnt/?view_only=08ad84266beb41ead5113d0acb2f890.

Competing interests

The authors declare no competing interests.

References

- Adler, N. E., Epel, E. S., Castellazzo, G., & Ickovics, J. R. (2000). Relationship of subjective and objective social status with psychological and physiological functioning: Preliminary data in healthy, White women. *Health Psychology, 19*(6), 586–592.
<https://doi.org/10.1037/0278-6133.19.6.586>
- Ali, R. (2020). *Comparing coronavirus deaths to drowning, auto accidents were probably bad examples*. <https://www.usatoday.com/story/entertainment/celebrities/2020/04/17/dr-phil-compares-coronavirus-deaths-car-accidents/5151534002/>
- Allcott, H., Boxell, L., Conway, J., Gentzkow, M., Thaler, M., & Yang, D. Y. (2020). Polarization and public health: Partisan differences in social distancing during the Coronavirus pandemic. *NBER Working Paper, w26946*.
- Aust, F., Diedenhofen, B., Ullrich, S., & Musch, J. (2013). Seriousness checks are useful to improve data validity in online research. *Behavior Research Methods, 45*(2), 527–535.
- Ballard, C. L., & Johnson, M. F. (2004). Basic math skills and performance in an introductory economics class. *The Journal of Economic Education, 35*(1), 3–23.
- Barends, A. J., & de Vries, R. E. (2019). Noncompliant responding: Comparing exclusion criteria in MTurk personality research to improve data quality. *Personality and Individual Differences, 143*, 84–89.
- Braithwaite, D. W., & Siegler, R. S. (2018). Developmental changes in the whole number bias. *Developmental Science, 21*(2), e12541.
- Brust-Renck, P. G., Nolte, J., & Reyna, V. F. (2017). Numeracy in Health and Risk Messaging. In *Oxford Research Encyclopedia of Communication*.
- Chen, F. F. (2007). Sensitivity of goodness of fit indexes to lack of measurement invariance. *Structural Equation Modeling: A Multidisciplinary Journal, 14*(3), 464–504.
- Cheung, G. W., & Rensvold, R. B. (2002). Evaluating goodness-of-fit indexes for testing measurement invariance. *Structural Equation Modeling, 9*(2), 233–255.

- Cohen, J. (2013). *Statistical power analysis for the behavioral sciences*. Routledge.
- Cokely, E. T., Galesic, M., Schulz, E., Ghazal, S., & Garcia-Retamero, R. (2012). Measuring risk literacy: The Berlin numeracy test. *Judgment and Decision Making*.
- Czeisler, M. É., Tynan, M. A., Howard, M. E., Honeycutt, S., Fulmer, E. B., Kidder, D. P., Robbins, R., Barger, L. K., Facer-Childs, E. R., & Baldwin, G. (2020). Public attitudes, behaviors, and beliefs related to COVID-19, stay-at-home orders, nonessential business closures, and public health guidance—United States, New York City, and Los Angeles, May 5–12, 2020. *Morbidity and Mortality Weekly Report*, *69*(24), 751.
- Davey, H. M., Barratt, A. L., Butow, P. N., & Deeks, J. J. (2007). A one-item question with a Likert or Visual Analog Scale adequately measured current anxiety. *Journal of Clinical Epidemiology*, *60*(4), 356–360.
- Deck, C., & Jahedi, S. (2015). The effect of cognitive load on economic decision making: A survey and new experiments. *European Economic Review*, *78*, 97–119.
- Dowker, A., & Nuerk, H.-C. (2016). Linguistic influences on mathematics. *Frontiers in Psychology*, *7*, 1035.
- Finch, J. F., West, S. G., & MacKinnon, D. P. (1997). Effects of sample size and nonnormality on the estimation of mediated effects in latent variable models. *Structural Equation Modeling: A Multidisciplinary Journal*, *4*(2), 87–107.
<https://doi.org/10.1080/10705519709540063>
- Fornell, C., & Larcker, D. F. (1981). Evaluating structural equation models with unobservable variables and measurement error. *Journal of Marketing Research*, *18*(1), 39–50.
- Galesic, M., & Garcia-Retamero, R. (2011). Graph literacy: A cross-cultural comparison. *Medical Decision Making*, *31*(3), 444–457.
- Geary, D. C. (2011). Cognitive predictors of achievement growth in mathematics: A 5-year longitudinal study. *Developmental Psychology*, *47*(6), 1539.

- Hair Jr, J. F., Hult, G. T. M., Ringle, C., & Sarstedt, M. (2016). *A primer on partial least squares structural equation modeling (PLS-SEM)*. Sage publications.
- Halpern, D. F., Benbow, C. P., Geary, D. C., Gur, R. C., Hyde, J. S., & Gernsbacher, M. A. (2007). The science of sex differences in science and mathematics. *Psychological Science in the Public Interest*, 8(1), 1–51.
- Hu, L., & Bentler, P. M. (1999). Cutoff criteria for fit indexes in covariance structure analysis: Conventional criteria versus new alternatives. *Structural Equation Modeling: A Multidisciplinary Journal*, 6(1), 1–55. <https://doi.org/10.1080/10705519909540118>
- Huber, S., Nuerk, H.-C., Reips, U.-D., & Soltanlou, M. (2019). Individual differences influence two-digit number processing, but not their analog magnitude processing: A large-scale online study. *Psychological Research*, 83(7), 1444–1464.
- Inglehart, R., Haerpfer, C., Moreno, A., Welzel, C., Kizilova, K., J, D.-M., Lagos, M., Norris, P., Ponarin, E., & Puranen, B. (2014). *World Values Survey: Round Six—Country-Pooled Datafile 2010-2014*. [Database]. Madrid: JD Systems Institute.
<http://www.worldvaluessurvey.org/WVSDocumentationWV6.jsp>
- Jasper, J. D., Bhattacharya, C., Levin, I. P., Jones, L., & Bossard, E. (2013). Numeracy as a predictor of adaptive risky decision making. *Journal of Behavioral Decision Making*, 26(2), 164–173.
- Jung, E., & Yoon, M. (2016). Comparisons of three empirical methods for partial factorial invariance: Forward, backward, and factor-ratio tests. *Structural Equation Modeling: A Multidisciplinary Journal*, 23(4), 567–584.
- Keller, C., & Siegrist, M. (2009). Effect of risk communication formats on risk perception depending on numeracy. *Medical Decision Making*, 29(4), 483–490.
- Kline, R. B. (2015). *Principles and practice of structural equation modeling*. Guilford publications.

- Landy, D., Charlesworth, A., & Ottmar, E. (2017). Categories of large numbers in line estimation. *Cognitive Science*, *41*(2), 326–353.
- Landy, D., Silbert, N., & Goldin, A. (2013). Estimating large numbers. *Cognitive Science*, *37*(5), 775–799.
- Lazer, D., Santillana, M., Perlis, R. H., Quintana, A., Ognyanova, K., Green, J., Baum, M. A., Simonson, M., Uslu, A. A., Chwe, H., Druckman, J., Lin, J., & Gitomer, A. (2020). *THE COVID STATES PROJECT: A 50-STATE COVID-19 SURVEY REPORT #26: TRAJECTORY OF COVID-19-RELATED BEHAVIORS* (No. 26; The COVID-19 Consortium for Understanding the Public's Policy Preferences Across States, p. <https://covidstates.org>). Northeastern University.
- Lewis, M. R., Matthews, P. G., & Hubbard, E. M. (2016). Neurocognitive architectures and the nonsymbolic foundations of fractions understanding. In *Development of Mathematical Cognition* (pp. 141–164). Elsevier.
- Leys, C., Ley, C., Klein, O., Bernard, P., & Licata, L. (2013). Detecting outliers: Do not use standard deviation around the mean, use absolute deviation around the median. *Journal of Experimental Social Psychology*, *49*(4), 764–766.
<https://doi.org/10.1016/j.jesp.2013.03.013>
- Little, T. D., Cunningham, W. A., Shahar, G., & Widaman, K. F. (2002). To parcel or not to parcel: Exploring the question, weighing the merits. *Structural Equation Modeling*, *9*(2), 151–173.
- Ludewig, U., Lambert, K., Dackermann, T., Scheiter, K., & Möller, K. (2019). Influences of basic numerical abilities on graph reading performance. *Psychological Research*, 1–13.
- Lusardi, A. (2012). *Numeracy, financial literacy, and financial decision-making*. National Bureau of Economic Research.

- Mackinnon, D. P., Lockwood, C. M., & Williams, J. (2004). Confidence limits for the indirect effect: Distribution of the product and resampling methods. *Multivariate Behavioral Research*, 39(1), 99–128.
- Maniaci, M. R., & Rogge, R. D. (2014). Caring about carelessness: Participant inattention and its effects on research. *Journal of Research in Personality*, 48, 61–83.
- Matthews, P. G., Lewis, M. R., & Hubbard, E. M. (2016). Individual differences in nonsymbolic ratio processing predict symbolic math performance. *Psychological Science*, 27(2), 191–202.
- McDonald, R. P. (2013). *Test theory: A unified treatment*. psychology press.
- Meade, A. W., & Craig, S. B. (2012). Identifying careless responses in survey data. *Psychological Methods*, 17(3), 437.
- Muthén, L. K., & Muthén, B. O. (2010). *Mplus: Statistical analysis with latent variables: User's guide*. Muthén & Muthén Los Angeles.
- Ni, Y., & Zhou, Y.-D. (2005). Teaching and learning fraction and rational numbers: The origins and implications of whole number bias. *Educational Psychologist*, 40(1), 27–52.
- Núñez-Peña, M. I., Guilera, G., & Suárez-Pellicioni, M. (2014). The single-item math anxiety scale: An alternative way of measuring mathematical anxiety. *Journal of Psychoeducational Assessment*, 32(4), 306–317.
- Nye, C. D., & Drasgow, F. (2011). Effect size indices for analyses of measurement equivalence: Understanding the practical importance of differences between groups. *Journal of Applied Psychology*, 96(5), 966.
- Peters, E., Hibbard, J., Slovic, P., & Dieckmann, N. (2007). Numeracy skill and the communication, comprehension, and use of risk-benefit information. *Health Affairs*, 26(3), 741–748.

- Putnick, D. L., & Bornstein, M. H. (2016). Measurement invariance conventions and reporting: The state of the art and future directions for psychological research. *Developmental Review, 41*, 71–90.
- Reips, U.-D. (2000). The Web experiment method: Advantages, disadvantages, and solutions. In *Psychological experiments on the Internet* (pp. 89–117). Elsevier.
- Reyna, V. F., Nelson, W. L., Han, P. K., & Dieckmann, N. F. (2009). How numeracy influences risk comprehension and medical decision making. *Psychological Bulletin, 135*(6), 943.
- Rogers, W. M., & Schmitt, N. (2004). Parameter recovery and model fit using multidimensional composites: A comparison of four empirical parceling algorithms. *Multivariate Behavioral Research, 39*(3), 379–412.
- Schneider, M., Merz, S., Stricker, J., De Smedt, B., Torbeyns, J., Verschaffel, L., & Luwel, K. (2018). Associations of number line estimation with mathematical competence: A meta-analysis. *Child Development, 89*(5), 1467–1484.
- Schwartz, S. R., McDowell, J., & Yueh, B. (2004). Numeracy and the shortcomings of utility assessment in head and neck cancer patients. *Head & Neck: Journal for the Sciences and Specialties of the Head and Neck, 26*(5), 401–407.
- Shepherd, M. (2020). *Is poor math literacy making it harder for people to understand COVID-19 Coronavirus?* <https://www.forbes.com/sites/marshallshepherd/2020/03/23/is-the-math-too-hard-for-people-to-understand-covid-19-coronavirus/#34da88bb6a9c>
- Siegel, E. (2020). *Why exponential growth is so scary for the COVID-19 coronavirus.* <https://www.forbes.com/sites/startswithabang/2020/03/17/why-exponential-growth-is-so-scary-for-the-covid-19-coronavirus/#367dd4ad4e9b>
- Spiegelhalter, D., Pearson, M., & Short, I. (2011). Visualizing uncertainty about the future. *Science, 333*(6048), 1393–1400.
- Steenkamp, J.-B. E., & Baumgartner, H. (1998). Assessing measurement invariance in cross-national consumer research. *Journal of Consumer Research, 25*(1), 78–90.

- Tabachnick, B. G., Fidell, L. S., & Ullman, J. B. (2007). *Using multivariate statistics* (Vol. 5). Pearson Boston, MA.
- Thompson, C. A., Taber, J., Coifman, K., & Sidney, P. (2020). *Math misconceptions may lead people to underestimate the true threat of COVID-19*. <https://theconversation.com/math-misconceptions-may-lead-people-to-underestimate-the-true-threat-of-covid-19-134520>
- Thompson, C. A., Taber, J. M., Sidney, P. G., Fitzsimmons, C., Mielicki, M., Matthews, P. G., Schemmel, E., Simonovic, N., Foust, J., & Aurora, P. (2020). *Math matters during a pandemic: A novel, brief educational intervention combats whole number bias to improve health decision-making and predicts COVID-19 risk perceptions and worry across 10 days*.
- Van Hoof, J., Verschaffel, L., & Van Dooren, W. (2015). Inappropriately applying natural number properties in rational number tasks: Characterizing the development of the natural number bias through primary and secondary education. *Educational Studies in Mathematics*, *90*(1), 39–56.
- Vandenberg, R. J., & Lance, C. E. (2000). A review and synthesis of the measurement invariance literature: Suggestions, practices, and recommendations for organizational research. *Organizational Research Methods*, *3*(1), 4–70.
- Weinfurt, K. P., Castel, L. D., Li, Y., Sulmasy, D. P., Balshem, A. M., Benson III, A. B., Burnett, C. B., Gaskin, D. J., Marshall, J. L., & Slater, E. F. (2003). The correlation between patient characteristics and expectations of benefit from Phase I clinical trials. *Cancer*, *98*(1), 166–175.
- Wolf, M. S., Serper, M., Opsasnick, L., O’Conor, R. M., Curtis, L. M., Benavente, J. Y., Wismer, G., Batio, S., Eifler, M., & Zheng, P. (2020). Awareness, attitudes, and actions related to COVID-19 among adults with chronic conditions at the onset of the US outbreak: A cross-sectional survey. *Annals of Internal Medicine*.

Woloshin, S., Schwartz, L. M., Moncur, M., Gabriel, S., & Tosteson, A. N. (2001). Assessing values for health: Numeracy matters. *Medical Decision Making*, 21(5), 382–390.

Appendix E

Numeracy and COVID-19: examining interrelationships between numeracy, health numeracy and behaviour

Nathan T.T. Lau¹, Eric D. Wilkey¹, Mojtaba Soltanlou¹, Rebekka Lagacé Cusiac¹, Lien Peters¹, Paul Tremblay¹, Celia Goffin¹, Isabella Starling Alves², Andrew David Ribner³, Clarissa Thompson⁴, Jo Van Hoof⁵, Julia Bahnmueller⁶, Aymee Alvarez¹, Elien Bellon⁷, Ilse Coolen⁸, Fanny Ollivier⁹, Daniel Ansari¹

¹Department of Psychology, Western University, Canada

²Department of Educational Psychology, University of Wisconsin-Madison, USA

³Learning Research and Development Center, University of Pittsburgh, USA

⁴Department of Psychological Sciences, Kent State University, USA

⁵Centre for Instructional Psychology and Technology, KU Leuven, Belgium

⁶Centre for Mathematical Cognition, Loughborough University, United Kingdom

⁷Parenting and Special Education, KU Leuven, Belgium

⁸Université de Paris, LaPsyDÉ, CNRS, F-75005 Paris, France

⁹Laboratoire de Psychologie, Cognition, Comportement et Communication, Université Rennes 2, France

Corresponding Author: Daniel Ansari, Department of Psychology, Western University, daniel.ansari@uwo.ca

Abstract

The COVID-19 pandemic has exposed people across the globe to large amounts of statistical data. Previous studies have shown that individuals' mathematical understanding of health-related information affects their attitudes and behaviours. Here, we investigate the relation between a) basic numeracy, b) COVID-19 health numeracy and c) COVID-19 health-related attitudes and behaviours. To do this, an online survey measuring these three variables will be distributed in Canada, the United States, and the United Kingdom. Basic Numeracy, COVID-19 health numeracy and COVID-19 health-related attitudes and behaviours are expected to be positively correlated with each other. Multigroup analysis will be used to investigate mean differences and differences in the strength of the correlation.

Keywords: numeracy, health numeracy, COVID-19, health policy and adherence

Introduction

The outbreak of the COVID-19 pandemic represents an unprecedented event. While human history is dotted with pandemics, never before have we been able to track and model a disease with such high sophistication. One of the many ways in which COVID-19 represents a watershed moment is the way in which it has pushed people across the planet to process and interpret rapidly evolving numerical information to inform their behaviours. Websites reporting COVID-19 statistics, such as total number of infections and total number of deaths, have been created (e.g., <https://coronavirus.jhu.edu>) and are widely cited in media reports on COVID-19, leading to unprecedented levels of attention to dynamic numerical information among the general population.

In addition to the numerical information pertaining to the consequences of the virus (e.g., number of cases, number of deaths), people across the planet are being introduced to unfamiliar and challenging mathematical concepts, such as ‘flattening the curve’, ‘exponential growth’, ‘false negative rates’ of COVID-19 tests, etc. Specifically, the degree to which individuals can make sense of COVID-19 numbers may influence the way they understand these critical health-related concepts which may, in turn, influence their adherence to public health advice and their perception of the risks posed by COVID-19.

Indeed, several experts in the study of mathematical cognition have recently discussed the role that numeracy (e.g., understanding proportions, large numbers) may play in how individuals estimate the risks associated with COVID-19 (Shepherd, 2020; Thompson, Taber, Coifman, et al., 2020; Thompson, Taber, Sidney, et al., 2020). For example, an online experimental study with over 1,200 U.S. participants found that those exposed to a short educational intervention consisting of a worked example with instructions on how to calculate fatality rates were more accurate on post-intervention health decision-making problems than those participants who were randomly assigned to a control condition (Thompson, Taber, Sidney, et al., 2020). Importantly, COVID-19 risk perceptions and worry increased throughout the 10-day study for those in the intervention condition, relative to those in the control condition. Since risk perception inherently involves magnitude comparisons (i.e., higher vs. lower risk), improving proportional understanding may have increased the accuracy involved in assessing risk magnitude. However, there are still open questions pertaining to the ways in which numeracy and understanding of health information related to COVID-19 may affect people's behaviour in response to the pandemic. Therefore, the aim of the present study is to address these knowledge gaps. In what follows, we first review the mathematical concepts and misconceptions that are relevant to understanding COVID-19 related information. We then go on to discuss how we aim to study the interrelations between (A) basic numerical processing and understanding (hereafter basic numeracy), (B) health numeracy directly relevant to COVID-19 (hereafter COVID-19 health numeracy) and (C) COVID-19 health-related attitudes and behaviours.

During the COVID-19 pandemic, policy makers and public figures frequently cite numerical information in order to defend decisions and influence public action. For example, some have argued against radical actions (e.g., stay-at-home orders and business closures) on the premise that equally large numbers of people die from other causes such as automobile accidents every year without causing nationwide shutdowns (Ali, 2020). As of April 7th (nearly a month after the first COVID-19 death was recorded in the US), John Hopkins University reported 31.4 thousand COVID-19 related deaths in the US compared to the 39.4 thousand deaths reported by the National Safety Council due to car accidents in the US over the course of 2018. However, comparing the current number of COVID-19 related deaths to the number of people who died in car accidents in a given year is like comparing apples and oranges, in part because both statistics have different denominators (i.e., one is per month while the other is per year). When taking the time period into account, we can observe that 31.4 thousand people died of COVID-19 while approximately 3 thousand people died of car accidents in the same one-month time period ($39,000/12=3,250$), leading us to draw drastically different conclusions about the severity of the pandemic. This example shows how important relational information (i.e., proportions) is when making decisions surrounding COVID-19.

Despite the importance of understanding relational information, there is a wealth of empirical evidence to show that processing proportions (fractions, decimals, percentages or odds ratios) is challenging for many adults (Lusardi, 2012; Spiegelhalter et al., 2011). What underlies these difficulties in processing proportions? One of the possible sources is the whole number bias. The whole number bias refers to the inappropriate use of whole number properties (i.e., natural numbers) when processing rational numbers (e.g., fractions and decimals; Ni & Zhou, 2005). In the specific case of fractions and odds ratios, the whole number bias also manifests when individuals pay more attention to the components of the proportion (i.e., the numerator and/or denominator) rather than to the magnitude of the entire ratio (i.e., the relation between the numerator and denominator; Braithwaite & Siegler, 2018; Lewis et al., 2016). For example, an individual might perceive a chance of 1 in 10 as being smaller than 10 in 100 simply because the numbers of the first odds ratio (i.e., 1 and 10) are smaller than the numbers in the second odds ratio (i.e., 10 and 100) although they represent the same proportion (i.e., 0.1).

In applied settings, paying attention to the magnitude of components rather than to the magnitude of the entire proportion can lead to other types of biases such as denominator neglect. Denominator neglect occurs when individuals compare two or more numbers of events without taking into account the total number of opportunities for that event to occur (Spiegelhalter et al., 2011). In the context of the COVID-19 pandemic, examples of this bias are common in the media when public figures state that the United States has carried out more tests than any other country, while ignoring the fact that the number of tests per capita (i.e., the number of tests accounting for population size) is much lower than several other countries at the time of writing. In addition to these misleading claims, other factors, such as how statistics are presented in the media, can also increase the occurrence of denominator neglect in the general population. For example, the media

often reports, either directly or with the use of graphs, absolute magnitudes such as the total number of COVID-19 cases. Although this information is important for health planning (e.g., estimating the number of ventilators needed), it also puts focus on the numerator and leads individuals comparing different regions to ignore total population size.

As is evident from the above, a strong understanding of proportions in a variety of formats is necessary in order to accurately understand COVID-19 related information. Moreover, the size of the numbers that are contained within COVID-19 related data may be a barrier to understanding that information and acting accordingly. For instance, it has been shown that many individuals have misrepresentations when it comes to their understanding of large numbers (i.e., in the order of thousands, millions and billions; Landy et al., 2013, 2017). Failing to grasp important magnitude differences between large numbers could result in altered perceptions such as minimizing the impact of the pandemic on a larger scale. Furthermore, processing proportions with large numbers may create additional cognitive load, even when individuals have an accurate representation of large numbers (Deck & Jahedi, 2015). Lastly, large numbers in COVID-19 data are a result of the virus' exponential spread, which is another concept that is poorly understood since people are more often exposed to linear (or arithmetic) growth (Siegel, 2020). All of these factors may translate into difficulty digesting complex mathematical information related to large-scale numbers associated with COVID-19. As a result, poor understanding of these concepts may lead people to undermine efforts to slow the spread of the disease by not following social distancing guidelines.

The review of the literature above suggests that having difficulties understanding mathematical concepts and accurately processing numerical quantities may influence the way in which individuals understand and act upon COVID-19 related information. Indeed, it has been shown that individuals with higher numeracy are more likely to pay attention to numerical information, interpret it correctly and make decisions accordingly relative to those with low numeracy (Peters et al., 2007). In contrast, individuals with relatively low numeracy are more susceptible to bias. For example, they are more likely to ignore numerical information and instead rely on intuitions based on emotional states and other extraneous factors, such as their trust in or distrust of the information source (Peters et al., 2007; Reyna et al., 2009). Furthermore, numeracy has been shown to influence how people evaluate risk (Keller & Siegrist, 2009; Schwartz et al., 2004; Woloshin et al., 2001). For example, individuals with higher numeracy levels have been shown to respond differently to high-risk vs low-risk situations while individuals with low numeracy do not (Keller & Siegrist, 2009). Another study on numeracy and adaptive decision making found that individuals with low numeracy were more likely to choose riskier options when small losses were inevitable (Jasper et al., 2013).

To the best of our knowledge, there has, to date, been no study that has examined the role of basic numeracy alongside applied, health-related numeracy on health outcomes. However, evidence from disparate areas of psychology and health informatics suggests that the three main

variables, basic numeracy, COVID-19 health numeracy and COVID-19 health-related attitudes and behaviours are related.

First, there is a wealth of evidence that suggests that those with stronger basic numeracy are better equipped to apply this knowledge to diverse and novel contexts (Ballard & Johnson, 2004; Geary, 2011; Ludewig et al., 2019). Given that public information about COVID-19 disseminated in the media requires the understanding of numerous mathematical concepts that are typically not encountered in the day-to-day, it is likely that those in the population that possess stronger basic numeracy would have a stronger grasp of the risks relating to the pandemic. Second, there exist many studies on health literacy, which have examined the association between applied mathematical skills, such as health- and non-health-related word problems, and health-related decisions and outcomes. Those who perform better at these problems tend to have better health outcomes in the future (Brust-Renck et al., 2017; Weinfurt et al., 2003). Finally, stronger basic numeracy may have an independent effect on health outcomes controlling for health numeracy. At an equal level of health numeracy, those with stronger basic numeracy may be more capable of understanding the underlying numerical concepts of COVID-19 health information. As such, they would be better equipped in spotting common pitfalls to which those less familiar with the underlying numerical concepts may fall prey.

The current study directly addresses this critical knowledge gap by assessing the relation of both (A) basic numeracy and (B) COVID-19 health numeracy to (C) COVID-19 health-related attitudes and behaviours in a correlational study. Basic numeracy involves numerical magnitude processing and conceptual understanding of rational numbers (e.g., proportions). COVID-19 health numeracy is our measure of health numeracy relevant for understanding numerical information about COVID-19 (e.g., risk factors and flattening the curve). And lastly, COVID-19 health-related attitudes and behaviours indexes people's perception of and adherence to policies such as social distancing and handwashing. It is hypothesized that all three variables are positively related with one another. Specifically, that basic numeracy is positively correlated with COVID-19 health numeracy, that COVID-19 health numeracy is positively correlated with COVID-19 health-related attitudes and behaviours, and that basic numeracy is positively correlated with COVID-19 health-related attitudes and behaviours.

The primary aims of the current study are (1) to assess the interrelations among basic numeracy, health numeracy, and health-related attitudes and behaviours and (2) determine whether the relations among the three variables differ across three different countries: Canada, the United States, and the United Kingdom. These countries were chosen because (1) they are all members of the G7 and thus have comparable economies and political systems, and (2) the response to COVID-19 (e.g., when lockdown and re-opening measures were implemented) as well as the case fatality and testing rates differed between the three countries. Furthermore, the majority language for all three countries is the same (i.e., English), thereby limiting potential confounds driven by linguistic differences in number representation (Dowker & Nuerk, 2016). Investigating the between-country

differences in the strength of the correlations and between-country differences in how covariates may influence this relationship will allow us to comment on whether and how the strength of the relations among the variables differ across countries, which can inform short-term government policies related to COVID-19.

Methods

Ethics Statement

The study was approved by the non-medical research ethics board at the University of Western Ontario and will be conducted according to their guidelines. Participants will be presented with a letter of information and implied consent will be acquired before starting the survey. Qualtrics panelists join from a variety of sources. They may be airline customers who chose to join in reward for SkyMiles, retail customers who opted in to get points at their favorite retail outlet, or general consumers who participate for cash or gift cards, etc. When participants are invited to take a survey, they are informed what they will be compensated.

Procedure and Materials

In total, 2,025 participants will complete an online survey hosted by Qualtrics (see Power Analysis). Qualtrics will collect the data using their participant panels. The survey will take approximately 20 minutes to complete and will include four sections: demographics and other cultural variables, basic numeracy, COVID-19 health numeracy, COVID-19 health-related attitudes and behaviors. The survey will start with the consent form, followed by the demographics section for all respondents. The order of the other three sections (i.e., basic numeracy, COVID-19 health numeracy and COVID-19 health-related attitudes and behaviours) will be randomized across respondents (see Figure 2). The survey will end with a few questions about the device that they used, a seriousness check (where participants will be asked whether they would keep their data if they were the experimenter), and a question about any technical problem they may have been faced with. For a list of all items, see supplementary material (https://osf.io/qpdnt/?view_only=08ad84266beb41eaa5113d0acb2f890).

Figure 2. Overview of the flow of the survey

Demographics and Other Potential Moderators

Detailed demographic information will be collected using 14 items covering age, gender, country of residence, ethnicity, socioeconomic status (SES; Adler et al., 2000), employment status (adapted from the World Value Survey, WVS-6; Inglehart et al., 2014) and living situation. In addition to these demographic variables, a short list of potential moderating or confounding variables will be collected in order to test for their possible effect on variables of interest ~~in the mediation model~~, namely COVID-19 news consumption (three items), anxiety (three items), political leaning (one item). For behaviors related to news consumption, participants will indicate how often they access news sources (e.g., news websites, government communication) or social sources (e.g., social media, friends and family) for news on a scale of 1 ("daily") to 5 ("less than monthly") and how well informed they feel about the pandemic on a scale of 1 ("not informed") to 4 ("very informed"). The items measuring anxiety were adapted from the Single-Item Math Anxiety scale (Núñez-Peña et al., 2014) and will measure general anxiety (Davey et al., 2007), anxiety related to COVID-19, and math anxiety. Participants will rate their anxiety level on a scale of 1 ("not anxious") to 10 ("very anxious"). For political leaning, participants will be asked to rate their beliefs on a visual analog scale with "left/liberal" label at the left end and "right/conservative" label at the right end (adapted from the WVS-6; Inglehart et al., 2014). Finally, information about the type of electronic device used to complete the survey will be collected due to the nature of the response to the number line tasks. It is expected that participants will use either computers or mobile devices to respond to the questionnaire, meaning that the total length of the number lines may vary from one device to the other. Therefore, information about the device used to answer the

survey will be collected in order to test for differences in number line accuracy due to screen size. Demographics items will be presented in a fixed order as no order effect will be expected.

Basic Numeracy

The basic numeracy measure is designed to capture elements of numeracy that are relevant for understanding health information related to COVID-19 and includes a total of 30 items. It was created by adapting and combining four existing measurement tools focused on the understanding and processing of proportions and large numbers.

First, numerical magnitude processing will be measured using the number line task, a task that has been consistently found to be correlated with different mathematical skills (Schneider et al., 2018). In this task, the participant will be presented with an empty number line bound on each side. The participant will be given a number and asked to indicate its location on a number line with respect to the two endpoints. Five versions of the task will be used to assess participants' understanding of whole numbers (six items), fractions (six items), percentages (three items), large numbers (five items), and nonsymbolic ratios (three items). For all number line items, accuracy will be scored using percentage absolute error (PAE), which is the absolute difference between the estimated and the correct answer divided by the scale of the number line. Therefore, a higher PAE indicates lower accuracy (i.e., estimation is further away from true value), while a low PAE indicates higher accuracy. Numbers were selected because they showed suitable variability for capturing individual differences in previous research (Landy et al., 2013; Thompson, Taber, Sidney, et al., 2020). For all five number line versions, at least 3 of the items were matched for relative position on the number line to provide points of comparison across number line types.

Second, conceptual knowledge of fractions and decimals will be measured using two multiple-choice items and two fill-in-the-blank items selected from the Fraction-knowledge Assessment (Matthews et al., 2016) in addition to two open-ended items selected from the Rational Number Sense Test (Van Hoof et al., 2015). Whereas number line items assess the processing of specific magnitudes, these items will assess broader conceptual understanding of rational numbers such as fractions and decimals (e.g., "How many possible fractions are between $\frac{1}{4}$ and $\frac{1}{2}$?"). These six items will be scored correct (1) or incorrect (0). Lastly, the one-item version of the Berlin Numeracy test (Cokely et al., 2012) will be included in the basic numeracy measure. Although it has been proven as a valid and reliable test of risk literacy, it is often used as a measure of general numeracy and will be used to validate the novel basic numeracy measure described in this section. The item included in this survey has been shown to discriminate participants roughly into a top and bottom half of risk literacy in a number of samples across several countries (Cokely et al., 2012). The item will be scored correct (1) or incorrect (0).

Basic numeracy items will be presented pseudorandomly: the items will be randomly present within a group of similar items to reduce task switching. For example, all items consisting

of non-symbolic ratios will be presented on the same page in a random order and will not be mixed with other variants of the number line task. A complete list of items can be found at: https://osf.io/qpdnt/?view_only=08ad84266beb41ead5113d0acb2f890).

COVID-19 Health Numeracy

COVID-19 health numeracy will measure understanding of COVID-19 concepts and statistics using a custom-made questionnaire composed of 18 multiple-choice items. Items cover three concepts important to understanding COVID-19 related data in the media: graph literacy, proportions, and odds ratios. Fictional data was used in all sections in order to avoid bias due to prior COVID-19 knowledge. All items will consist of multiple choice items and be scored correct (1) or incorrect (0). Therefore, a higher score indicates high understanding of COVID-19 related concepts and statistics, whereas a lower score indicates poor understanding of COVID-19 related concepts and statistics.

Nine items will measure graph literacy. Of these, five items were conceptually modeled after the graph literacy scale proposed by Galesic and Garcia-Retamero (Galesic & Garcia-Retamero, 2011). Individuals will be shown a graph representing the total number of confirmed COVID-19 cases over time for two fictional countries, one with linear growth and the other with exponential growth. Four of these items will have one of three difficulty levels related to graph literacy. The first difficulty level consists of retrieving information, such as data points, directly from the graph (one item). The second difficulty level consists of evaluating relationships between data points, for example, by finding where two curves intersect each other (two items). The third level consists of making inferences from the information given by the graph, without the information being directly observable in the graph (one item). Also, one item will measure explicit knowledge of linear and exponential functions. The last four graph literacy items will test individuals' knowledge of "flattening the curve" and be accompanied by a diagram showing curves for the number of cases over time with and without social distancing. Questions will test participants' understanding of the effects of social distancing on various aspects of potential infection rate scenarios, such as length of outbreak and the height and latency of the peak of infection of the pandemic, based on the diagram.

Six items will measure the ability to accurately process absolute and relative magnitudes in the context of a highly infectious disease. A table containing fictional data about COVID-19 (i.e., number of confirmed cases, number of deaths, number of tests and total population) for three fictional countries will be presented. Similar to the section on graph literacy, items will have one of two difficulty levels. The first difficulty level will assess the ability to retrieve relevant data from the table (e.g., "Which country has conducted the most COVID-19 tests?"; three items). The second difficulty level will assess the ability to obtain relevant proportions from the data in order

to accurately compare country statistics (e.g., “Which country has the highest fatality rate for COVID-19?”; three items).

Finally, three items will measure participants’ ability to compare odds ratios. A situation describing fictional fatality rates for COVID-19 alone as well as for two other hypothetical risk factors will be presented using odds ratios. Items will involve comparing the odds ratios associated with the different risk factors and calculating relative risk.

Similar to items in the basic numeracy section, COVID-19 health numeracy items will be presented pseudorandomly. In other words, items within a subsection (e.g., items about flattening of the curve) will always be presented together in a random order to reduce task switching. For a list of items, see the supplementary material (https://osf.io/qpdnt/?view_only=08ad84266beb41eaa5113d0acb2f890).

Attitudes and Behaviors Towards COVID-19

Attitudes and behaviors towards COVID-19 will be measured using a total of 17 items. The first 12 items will focus on four frequently issued recommendations by health authorities (e.g., WHO): washing hands frequently and thoroughly, staying home unless the travel is essential, social distancing, and wearing a mask in public. For each recommendation, participants will be asked to indicate (1) whether or not this recommendation was issued by their local authorities, (2) when applicable, the extent to which they have followed the respective recommendation on a scale of 1 (“Never”) to 10 (“Consistently all the time”), and (3) how useful they think this recommendation is in the fight against the COVID-19 pandemic on a scale of 1 (“Completely useless”) to 10 (“Extremely useful”). Next, two items will measure participants' perceived change in (1) their own behavior and (2) people around them (e.g., family, close friends) in response to the COVID-19 pandemic. Participants will respond to both items on a scale of 1 (“Not at all”) to 10 (“To a great extent”). The last three items will assess participants’ perceptions about the severity and impact of the COVID-19 pandemic by assessing the degree to which they believe the COVID-19 pandemic is a serious global threat, COVID-19 is a serious medical condition, and that the benefits of the recommended actions to fight the COVID-19 pandemic outweigh their psychological, economic, and cultural costs. For these items, participants will respond on a scale of 1 (“Not at all”) to 10 (“To a great extent”).

Items of attitudes and behaviors towards COVID-19 will be presented in a fixed order to reduce bias in response selection as respondents might justify their responses about their own behaviors based on the local authorities' recommendations. For a complete list of items, see the supplementary material (https://osf.io/qpdnt/?view_only=08ad84266beb41eaa5113d0acb2f890).

Power Analysis

The current study focuses on the interrelations among three main constructs of interest, COVID-19 health numeracy, and COVID-19 health-related attitudes and behaviours. We are particularly interested in the robustness of these intercorrelations controlling for the covariates, and whether there may be between country differences in these intercorrelations. For the following power analysis, all α -levels are 0.05.

We calculated the necessary sample size to achieve 95% power using multiple Monte-Carlo simulation analyses in Mplus with 10,000 replications. As the relations between basic numeracy, COVID-19 health numeracy, and COVID-19 health-related attitudes and behaviours have yet to be explored, using effect sizes from previous literature for the purpose of power calculations would not be feasible. Most of the values in the following analyses were chosen based on reasonable expectation of correlations or minimum effect size of interest. However, in the multigroup analysis, we have decided on detecting a standardized correlation coefficient difference of 0.2 based on practical concerns, as the number of participants needed to detect a smaller difference would make the number of participants required prohibitively large.

Simple Correlation Model

To calculate the power to observe a correlation between the main variables of interest, we simulated a model with three variables with mean of 0 and variance of 1 and all intercorrelations set to be 0.14. A correlation coefficient of 0.14 corresponds to a small effect size (MacKinnon et al., 2004). A Monte-Carlo simulation revealed that a total of 675 participants would be sufficient to achieve 95% power for each regression coefficient.

Measurement Invariance of the Main Variables

As the current study operationalizes the main constructs as latent variables and we are employing multigroup analysis, measurement invariance between the countries must be considered. Measurement invariance consists of a number of steps that tests that different of the factor/latent variable structure is indifferent across countries. These include (1) configural invariance - the overall factor structure (i.e., the number of factors and their indicator variables), (2) metric invariance - the loadings of the indicator variables, (3) scalar invariance - the intercepts of the indicator variables, and (4) strict measurement invariance - the residual variances of the indicator variables (Putnick & Bornstein, 2016). The presence of invariance and non-invariance is often informative, and hypotheses were considered in table 1.

Calculations of power for measurement invariance are typically not calculated. This is because violations of different levels of measurement invariance are typically assessed using model fit indices (e.g., Comparative Fit Index and Root Mean Square Error of Approximation),

and these indices are indifferent to sample size. For all models below, a Monte-Carlo simulation study was conducted across two countries. For each country, we simulated three latent factors with mean of 0 and variance set to 1. Intercorrelations between all factors were set to 0.14. Each latent variable contained four indicators with factor loading of 0.5, residual of 0.75, and intercept of 3. A factor loading of 0.5 for each indicator variable can be interpreted as a standardized coefficient, which corresponds to a 25% explained variance and 75% residual variance.

Configural Invariance

To simulate a violation of configural invariance, one indicator item for factor 1 from the second country was instead loaded on to another with a factor loading of 0.5. Power was calculated as the proportion of simulations in which the loading of the errant indicator item was not statistically significant. Results indicated 675 participants would be sufficient for 95% power to detect a violation.

Metric Invariance

To simulate a violation of metric invariance, one item each from factor 2 and factor 3 was loaded on their respective factors at 0.2 instead of 0.5 (see Table 1 for hypotheses). Power was calculated as the proportion of simulations in which the chi-square difference test between the configural invariant and metric invariant model was significant. Results indicate 675 participants would be sufficient for 95% power to detect a violation.

Scalar Invariance

To simulate a violation of scalar invariance, one item intercept from factor 1 was set to 4 instead of 3. Power was calculated as the proportion of simulations in which the chi-square difference test between the metric invariant and scalar invariant model was significant. Results indicate 675 participants would be sufficient for 95% power to detect a violation.

Latent Mean Differences

To simulate latent mean differences, the latent mean of the second group was set to be 0.3 - a small to medium effect size (Cohen, 2013). Power was calculated as the proportion of simulations in which a chi-square difference test was obtained when testing between a model where latent means were restricted to be the same and another model where latent means were allowed to be estimated freely.

Multigroup Analysis

We simulated the analyses that will be used to address whether there are country-level differences in the correlation between the main variables by modelling a multigroup analysis. Using the simple correlation model above, we stipulated a model with same number of participants in each country. Specifically, we aim to detect between-country differences in the intercorrelation.

The comparisons were carried out pairwise using the MODEL CONSTRAINT command in Mplus. To calculate power, we simulated a multigroup analysis with two countries, and set to detect a difference in correlation of 0.2 - a small to medium effect size (Cohen, 2013). Monte-Carlo simulations revealed that 675 participants per country would be needed to achieve 95% power.

In sum, to achieve 95% power for all three analyses, a total of 675 participants each across the United States, Canada and the United Kingdom (total sample 2025 participants) would be required. For data generation code, simulated data, and analysis code, see the supplementary material (https://osf.io/qpdnt/?view_only=08ad84266beb41eaa5113d0acb2f890).

Preprocessing of Data

Exclusion Criteria

Due to the nature of the online sampling, we will exclude participants who are inattentive. Two types of inattentiveness are commonly observed, one being general inattentiveness and the other being marked by frequently selecting the same answer for entire blocks and completing a survey in short periods of time (Meade & Craig, 2012). We will use several criteria to exclude inattentive participants (Huber et al., 2019). First, we will remove respondents with multiple submissions, who respond multiple times using an IP check (Reips, 2000). Second, we will include a seriousness check (Aust et al., 2013). Data for participants indicating they were not serious will be discarded. Third, three instructed response items adapted from Barends and de Vries (Barends & de Vries, 2019) will be asked on different steps of the survey (For a list of three items, see supplementary material (https://osf.io/qpdnt/?view_only=08ad84266beb41eaa5113d0acb2f890)). Data for participants who respond incorrectly to more than one of these three questions will be discarded. Fourth, participants with more than 25% missing data from selecting “prefer not to answer” will be discarded. **Finally Fifth**, participants whose survey completion time is less than half of the 5% trimmed mean of the complete sample of participants would be flagged as potentially not paying attention, and responses will be examined for inconsistencies (Maniaci & Rogge, 2014). **Finally, to ensure that participants have not been influenced by a reception of a COVID-19 vaccine or the participation in a COVID-19 vaccination trial, we included two questions inquiring whether participants have received a COVID-19 vaccine or participated in a COVID-19 vaccination trial. Participants who have answered in the affirmative for either question will be excluded.**

Outlier Identification and Missing Data

Univariate outliers will be detected using the absolute deviation around the median, where continuous values ± 3 Median Absolute Deviation (MAD) will be considered outliers (Leys et al., 2013). Multivariate outliers will be detected using Mahalanobis distance test with a $p < .001$

(Tabachnick et al., 2007). Most questions in the survey require rating-scale responses that have well-defined ranges, and the removal of outliers for these questions may underestimate actual variability in the population. Therefore, for rating-scale questions, identified outliers will be examined for obvious errors or participant inattention. If no evidence for errors or inattention is found, the value will be retained in the final dataset. For questions without a well-defined range (i.e., the number line questions in the basic numeracy section) identified outliers will be examined for errors. If no errors are found, the outliers will be treated as missing data. **Finally, for the set of questions in the number line task, we will calculate the median absolute deviation of each participants' responses and will exclude participants if the median absolute deviation of their responses is below 0.1 (please see Preliminary Measurement Study for more details).**

We will conduct a missing values analysis to identify patterns of missing data in the items. We anticipate two patterns that often occur in online research. The first consists of missing random responses throughout the questionnaire, due either to inattention, or to participants not wanting to answer specific items. We will analyze frequency of non-response for each item and identify items with higher non-response frequency. These items are not missing at random, and we will determine if they can be explained by covariates included in the current study, at which point, we will include the covariates either in the model or as auxiliary variables that can be used to improve the estimation of missing data in full information maximum likelihood (FIML).

The second pattern of missingness that we anticipate is attrition throughout the questionnaire. Respondents who drop-off at different points along the questionnaire may differ from those who do not in three potentially different ways (i.e., less motivation, having restricted amounts of uninterrupted time available to complete the survey, lower overall math skills). We will be able to analyze this particular pattern by testing whether missingness at the later points is related to performance on earlier items. Regardless of the in-depth missingness analyses, FIML (Muthén & Muthén, 2010) is still the best approach, and no cases will be left out except when careless responding has been identified.

Analysis Plan

The main hypotheses of the current study are concerned with the interrelations between three hypothesized latent variables: basic numeracy, COVID-19 health numeracy, and COVID-19 health-related attitudes and behaviours. To assess these interrelations, several measures were newly developed for the current study. As a result, there is a large degree of uncertainty regarding whether the data would fit both the hypothesized measurement model (i.e., the relation between the latent factors and their indicators) and the structural model (i.e., the relation between latent factors). Given the unpredictable nature of the measures, it may be necessary to make substantial changes to the hypothesized statistical model post data collection. This is not ideal as it may increase the risk of overfitting the data. To alleviate this risk, we will conduct a preliminary measurement study of the variables described above with the primary purpose of assessing item

quality and the measurement model. The data from the pilot study will be used to inform substantive decisions regarding the measurement model that will be submitted as part of the stage 1 revision of the registered report before main data collection (see figure 3 and figure 4 for a decision tree for the pilot study and main analysis).

Figure 3. Decision Tree to Analyze the Preliminary Measurement Study Data.

Figure 4. Decision Tree to Analyze the Main Study Data.

Reliability and Validity

Subsequent to estimating the EFA and CFA in both the preliminary measurement study and main study (see below), **reliability** – the degree to which indicators of the latent factor consistently measure the same underlying construct – will be assessed using coefficient omega for each latent variable estimated (McDonald, 2013). We will use the typical cut-off point for acceptable reliability of 0.7 for each of the three latent variables (Kline, 2015; McDonald, 2013). Once reliability is established, convergent validity – the degree to which individual items are strongly related to the hypothesized factor – will be assessed using average variance extracted (AVE). We will use the typical cutoff score for acceptable convergent validity, where AVE for each of the three factor should be higher than 0.5. Finally, discriminant validity – the degree to which the degree to which individual items are not related to other factors – will be assessed using the Fornell & Larcker criterion (Fornell & Larcker, 1981; Hair et al., 2016). Specifically, we will adopt the convention that the square root of the AVE for each construct should be larger than the correlation of the specific construct with any of the other construct.

Preliminary Measurement Study

Data from 175 participants from each of the three countries of interest (totalling 525 participants) were collected. Preprocessing of data was conducted in the same manner as proposed for the main data set (see **Preprocessing of Data**). Participant exclusion criteria outline above were examined and judged to be adequate. However, in an examination of participants' responses for the number line task, we found some participants who answered the number line questions by placing their answers on the same point on the number line irrespective of the question prompt. To remove these non-responsive participants, we have added an additional exclusion criterion for the processing of the main data. Specifically, for the set of questions in the number line task, we will calculate the median absolute deviation of each participants' responses and will exclude participants if the median absolute deviation of their responses is below 0.1. In total, 79 participants were removed, which amount to a 15% exclusion rate. Therefore, for the main data analysis, we will collect an additional 15% participants from each country to reach the desired total participant of 675.

Table 1

Sample Statistics for All Items

Continuous Variables	Mean	S.D.	Skew	Kurtosis
Basic Numeracy Number line Q1	0.080	0.199	2.284	4.629
Basic Numeracy Number line Q2	0.005	0.134	1.181	3.007
Basic Numeracy Number line Q3	-0.079	0.188	-2.028	3.949
Basic Numeracy Number line Q4	0.081	0.166	2.753	8.335
Basic Numeracy Number line Q5	0.027	0.132	0.916	2.292

Basic Numeracy Number line Q6	-0.046	0.128	-2.783	10.446
Basic Numeracy Number line Q7	0.174	0.231	0.577	-0.907
Basic Numeracy Number line Q8	0.090	0.242	1.021	-0.067
Basic Numeracy Number line Q9	-0.102	0.192	-1.331	1.226
Basic Numeracy Number line Q10	0.179	0.230	0.529	-0.821
Basic Numeracy Number line Q11	0.105	0.228	0.797	-0.445
Basic Numeracy Number line Q12	-0.120	0.190	-1.147	0.841
Basic Numeracy Number line Q13	0.024	0.154	2.495	6.238
Basic Numeracy Number line Q14	-0.054	0.105	1.897	10.228
Basic Numeracy Number line Q15	-0.070	0.155	-2.381	5.939
Basic Numeracy Number line Q16	0.197	0.222	1.195	0.671
Basic Numeracy Number line Q17	0.104	0.174	1.208	1.083
Basic Numeracy Number line Q18	-0.061	0.107	-1.701	4.399
Basic Numeracy Number line Q19	0.247	0.200	1.122	0.916
Basic Numeracy Number line Q20	0.235	0.226	1.098	0.168
Basic Numeracy Number line Q21	0.119	0.233	1.247	0.625
Basic Numeracy Number line Q22	0.033	0.195	0.896	0.266
Basic Numeracy Number line Q23	-0.133	0.169	-0.902	0.505
Basic Numeracy Number line Q24	0.080	0.199	2.284	4.629
Attitudes and Behaviors Q1	8.808	1.437	-1.542	2.948
Attitudes and Behaviors Q2	8.360	1.876	-1.536	2.521
Attitudes and Behaviors Q3	8.924	1.512	-2.081	5.035
Attitudes and Behaviors Q4	9.309	1.444	-3.198	12.461
Attitudes and Behaviors Q5	9.067	1.433	-2.065	5.377
Attitudes and Behaviors Q6	8.868	1.647	-1.972	4.719
Attitudes and Behaviors Q7	9.094	1.456	-2.370	7.367
Attitudes and Behaviors Q8	8.982	1.733	-2.393	6.249
Attitudes and Behaviors Q9	8.622	1.707	-1.662	3.181
Attitudes and Behaviors Q10	8.425	1.699	-1.376	2.275
Attitudes and Behaviors Q11	8.859	1.797	-2.042	4.203
Attitudes and Behaviors Q12	8.805	1.760	-1.838	3.376
Attitudes and Behaviors Q13	8.186	2.257	-1.564	1.977
Categorical Variables				
		% Correct	% Incorrect	
Basic Numeracy Q1		0.736	0.264	
Basic Numeracy Q2		0.112	0.888	
Basic Numeracy Q3		0.670	0.330	
Basic Numeracy Q4		0.456	0.544	
Basic Numeracy Q5		0.592	0.408	
Basic Numeracy Q6		0.800	0.200	
Basic Numeracy Q7		0.153	0.847	
COVID-19 Health Numeracy Q1		0.785	0.215	
COVID-19 Health Numeracy Q2		0.819	0.181	
COVID-19 Health Numeracy Q3		0.600	0.400	
COVID-19 Health Numeracy Q4		0.472	0.528	
COVID-19 Health Numeracy Q5		0.391	0.609	

COVID-19 Health Numeracy Q6	0.671	0.329
COVID-19 Health Numeracy Q7	0.817	0.183
COVID-19 Health Numeracy Q8	0.441	0.559
COVID-19 Health Numeracy Q9	0.452	0.548
COVID-19 Health Numeracy Q10	0.843	0.157
COVID-19 Health Numeracy Q11	0.414	0.586
COVID-19 Health Numeracy Q12	0.494	0.506
COVID-19 Health Numeracy Q13	0.367	0.633
COVID-19 Health Numeracy Q14	0.805	0.195
COVID-19 Health Numeracy Q15	0.606	0.394
COVID-19 Health Numeracy Q16	0.609	0.391
COVID-19 Health Numeracy Q17	0.483	0.517
COVID-19 Health Numeracy Q18	0.468	0.532
Attitudes and Behaviors Q14	0.971	0.029
Attitudes and Behaviors Q15	0.960	0.040
Attitudes and Behaviors Q16	0.991	0.009
Attitudes and Behaviors Q17	0.989	0.011

Sample statistics are presented in table 1. Examination of the skewness and kurtosis of the continuous variables that most items modestly deviated from normality (Skewness < 2, Kurtosis < 7; Finch, West, & MacKinnon, 1997). However, all items from the COVID-19 Attitudes and Behaviors scale have very high means and negative skewness, indicating a ceiling effect is present. As this ceiling effect is observed for all items administered and that multiple other research studies with similar scales also yielded similar censored results (e.g., Czeisler et al., 2020; Wolf et al., 2020), we believe that this ceiling effect reflects the underlying population distribution. To account for this ceiling effect, all items from the COVID-19 Attitudes and Behaviors scale were treated as ordered categorical variables for the following analysis. Examination of the dichotomous variables suggest a good distribution of difficulty. However, 4 items from the COVID-19 Attitudes and Behaviors (the items asking whether different safety recommendation were issued by their local authorities) lacked variability, as most participants have responded in the affirmative. Therefore, we have removed these four items from analysis.

We next conducted separate EFAs for the set of items in each of the three main variables of interest. As there are multiple categorical variables, the WLSMV estimator and goemin rotation was used. Examination of the Scree plot suggests that a one factor solution would fit for each of the three variables. For the basic numeracy factor, factor loadings ranged from 0.320 to 0.863, for COVID-19 health numeracy, factor loadings ranged from 0.404 to 0.759, and for COVID-19 Attitudes and Behaviors, factor loadings ranged from 0.708 to 0.910. Aggregate scores of Basic numeracy and COVID-19 health numeracy are correlated at 0.778, basic numeracy and COVID-19 Attitudes and Behaviors are correlated at -0.113, and COVID-19 health numeracy and COVID-19 Attitudes and Behaviors are correlated at -0.090.

Reliability as measured by coefficient omega for basic numeracy, COVID-19 health numeracy, and COVID-19 Attitudes and Behaviors were 0.934, 0.901, and 0.960, respectively. As the coefficient omega for each latent variable is above 0.7, this suggest that the individual items have a satisfactorily level of internal consistency. However, average variance extracted (AVE) for basic numeracy, COVID-19 health numeracy, and COVID-19 Attitudes and Behaviors were 0.330, 0.370 and 0.638, respectively. This suggest that convergent validity was not established for basic numeracy and COVID-19 health numeracy. Similarly, AVE values for basic numeracy and COVID-19 health numeracy are lower than the squared correlations between the variables, suggesting discriminant validity was not established. In sum, these results suggest that a measurement model with individual items serving as indicators for the latent variables is not satisfactory.

We next proceeded to constructing parcels. Given that the EFAs suggest a one factor solution for each of the main variables, we proceeded with constructing parcels using the balancing approach, where the three items with strongest item-scale correlation are paired with the three items with the weakest item-scale correlation to form 3 parcels. Additional items are assigned to one of the three parcels successively, alternating directions through the parcels, until all items are assigned. For example, in the case of 12 indicators ranked by item-scale correlation, parcel#1 = 1, 6, 7, 12; parcel#2 = 2, 5, 8, 11; and parcel#3 = 3, 4, 9, 10 (Rogers & Schmitt, 2004). Three parcels were constructed for each latent variable. Model fit for the CFA was good ($\chi^2(24) = 33.54$, $p = 0.093$, CFI = 0.997, TLI = 0.995, RMSEA = 0.030).

For the basic numeracy factor, factor loadings ranged from 0.844 to 0.899, for COVID-19 health numeracy, factor loadings ranged from 0.731 to 0.776, and for COVID-19 Attitudes and Behaviors, factor loadings ranged from 0.913 to 0.950. Basic numeracy and COVID-19 health numeracy are correlated at 0.787, basic numeracy and COVID-19 Attitudes and Behaviors are correlated at -0.091, and COVID-19 health numeracy and COVID-19 Attitudes and Behaviors are correlated at -0.080.

Reliability as measured by coefficient omega for basic numeracy, COVID-19 health numeracy, and COVID-19 Attitudes and Behaviors were 0.906, 0.799, and 0.952, respectively. As the loadings were high and the coefficient omega for each latent variable is above 0.7, this suggest that the latent variables have high internal consistency. AVE for basic numeracy, COVID-19 health numeracy, and COVID-19 Attitudes and Behaviors were 0.764, 0.571 and 0.869, respectively. As the AVEs of the three latent variables are above 0.5, this suggest that convergent validity was established for basic numeracy and COVID-19 health numeracy. Similarly, the square root of the AVE for each construct is larger than the correlation of the specific construct with any of the other construct. This suggest that discriminate validity was established. In sum, these results suggest that a measurement model with parcels serving as indicators for the latent variables is satisfactory.

In light of these results from the preliminary measurement study, we will take the following actions for the final data collection: 1) we will add an additional participant exclusion criterion that removes participants when there are clear evidence of non-responsiveness, 2) we will remove 4 items from the COVID-19 Attitudes and Behaviors (the items asking whether different safety recommendation were issued by their local authorities) from the main analysis, and 3) we will proceed with analyzing the data using a CFA with parcels employing the balancing approach.

Item Parcels

The current study employs a large number of items in the measurement of the main variables. Operationalizing latent variables with a large number of indicator items is not ideal from a psychometric perspective. This is because individual items tend to be statistically less reliable than aggregates and have a larger likelihood to have correlated specific effects may subject the final model to nuisance parameters (i.e., correlated residuals; Little et al., 2002). To address this, we will employ an item parcel approach to form a smaller set of composite indicator items to inform the latent variables.

Given that the current survey employs multiple question formats (e.g., multiple choice, fill-in-the-blank, and number line) a random parceling approach would not be ideal, as random parceling assumptions that all items are interchangeable. Results from the preliminary measurement study have shown that each of the hypothesized latent variables are unidimensional. As such, parcels will be assigned using a balancing approach.

Main Analyses

After ensuring data quality with participant and outlier exclusion, we will pursue the research questions using a structural equation modelling approach. Basic numeracy, COVID-19 health numeracy, and COVID-19 health-related attitudes and behaviours will be operationalized as latent variables. All analyses will be carried out using Mplus 8.3 using the maximum likelihood estimator with robust standard errors (Muthén & Muthén, 2010). Where applicable, goodness of fit of the models will be assessed with χ^2 test statistic, the Comparative Fit Index (CFI), the Tucker-Lewis Index (TLI), and the root mean square error of approximation (RMSEA). Typical cut-off scores for excellent and adequate fit are CFI and TLI > 0.95 and > 0.90, and RMSEA < 0.06 and < 0.08, respectively (Hu & Bentler, 1999). In the case of inadequate fit (i.e., CFI and TLI < 0.9, RMSEA > 0.08), modification indices will be consulted for correlated residuals and other sources of misfit. Average

Simple Correlation Model

We will first examine the associations between the three main variables in a simple correlation model. As a first step, a model with only the three main (latent) variables would be

entered. As a second step, the robustness of the observed correlations will be tested by entering covariates as predictors of the three main variables.

Invariance Testing

The main objective of this research will be to investigate differences across countries in the means and variation of the main variables in the model (basic numeracy, COVID-19 health numeracy, and COVID-19 health-related attitudes and behaviors). This will include multiple-group measurement invariance analyses of the individual latent measures. This procedure is usually performed as a step to establish factorial validity across groups. Once established, it becomes possible to evaluate differences in latent means, variances, covariances, or specific correlations across different countries.

We will use the standard procedures in the confirmatory factor analytic literature for evaluating measurement invariance across countries (Kline, 2015). In short, this consists of a number of steps testing the invariance of different parts of the factor/latent variable structure. These include (1) configural invariance - the overall factor structure (i.e., the number of factors and their indicator variables), (2) metric invariance - the loadings of the indicator variables, and (3) scalar invariance - the intercepts of the indicator variables (Putnick & Bornstein, 2016). Strict measurement invariance is not required for the following analyses and will not be tested. These are tested in incremental stages, and the first three stages are essential to proceed with tests of latent means. Configural invariance will be assessed using model fit, where CFI and TLI > 0.90 and RMSEA < 0.08 is generally accepted as adequate model fit (Hu & Bentler, 1999). Metric and scalar invariance are evaluated using the chi-square difference test and the degradation of fit in CFI and RMSEA. Specifically, we will use the criterion of $\Delta RMSEA < .015$ and $\Delta CFI < -0.01$ (Chen, 2007; Cheung & Rensvold, 2002).

In the case where measurement invariance is not achieved, we will proceed with partial-invariance tests and identify the source of non-invariance. Some degree of researcher judgement would be required to gauge the severity of the invariance between countries. In the event that we do not meet configural invariance, modification indices will be examined for the source of the misfit. If the degree of misfit is small, the removal of the problematic indicator item could be a potential solution. This may necessitate the examination of between-country differences in the individual items constituting the parcel to find problematic items. If the degree of invariance is high, it may be necessary to estimate separate models for each country. In the event that we do not meet metric or scalar invariance in a given country, invariant items will be identified using the forward method (Jung & Yoon, 2016), effect size of invariance (d_{MACS}) will be calculated to gauge the magnitude of invariance (Nye & Drasgow, 2011). Identified invariant items will be allowed to vary across countries. We will proceed with partial scalar or metric invariance if less than half of the indicators are not invariance (Steenkamp & Baumgartner, 1998; Vandenberg & Lance, 2000). Lack of invariance in the factor structure is often seen as a nuisance, but it can also advance our

understanding of cross-cultural differences in how concepts are conceptualized. For example, if results indicate that the sole source of non-invariance in the latent variable COVID-19 health-related attitudes and behaviors is the factor loading of mask-wearing frequency, it can be interpreted that the amount of variance of mask wearing frequency that COVID-19 health-related attitudes and behaviors can explain is different between countries.

Finally, in the case where scalar invariance, latent variance equality and path equality is achieved (Vandenberg & Lance, 2000), latent mean differences between countries will be examined.

Multigroup Analysis

Once metric or scalar invariance or partial metric or scalar invariance is achieved, between-country differences in latent-correlations will be examined in two steps. As a first step, we will formulate a model with only the three main variables and examine the intercorrelations. As a second step, the robustness of the observed intercorrelations will be tested by entering covariates as predictors of the three main variables. Between-country differences in the correlations will be examined pairwise using the MODEL CONSTRAINT command in Mplus.

Exploratory Analyses

Potential Interactions with Covariates

While tangential to our main research questions, multiple covariates included in our analyses are theoretically interesting and may greatly influence the main variables in the study. Unfortunately, how these influences would take place are difficult to delineate a-priori. Therefore, we will explore these potential covariates in the following additional analyses. *All exploratory analysis will be examined, but only reported if statistically significant and theoretically interesting results are found.*

Political Leaning. One trend that can be observed is that opinions regarding the appropriate current and future community and personal actions taken in response to the pandemic differ depending on the political ideology to which citizens belong (Allcott et al., 2020). This suggests that the relation between COVID-19 health numeracy and COVID-19 health-related attitudes and behaviors may differ depending on political ideology. *We will explore this possibility by including an interaction term as a predictor of COVID-19 health numeracy and COVID-19 health-related attitudes and behaviors. Specifically, a COVID-19 health numeracy x Political leaning variable will be entered as a predictor of COVID-19 health-related attitudes and behaviors, and a COVID-19 health-related attitudes and behaviors x Political leaning variable will be entered as a predictor of COVID-19 health numeracy. In such a case, a significant coefficient between the interaction terms and COVID-19 health numeracy or COVID-19 health-related attitudes and behaviors would suggest that there is a significant interaction.*

COVID-19 Related Anxiety. While few studies have examined whether COVID-19-related anxiety is distinct from general anxiety, it is likely that the novelty and prevalence of COVID-19 would inspire differing emotional reactions that are at least somewhat distinct from general anxiety. Different individuals with comparable numeracy may nevertheless perceive the magnitude of the threat pertaining to the same information differently depending on COVID-19 related anxiety. Therefore, it is hypothesized that the relationship between COVID-19 health numeracy and COVID-19 health attitudes and behaviors may differ depending on COVID-19-related anxiety. We will explore this possibility by including an interaction term as a predictor of COVID-19 health numeracy and COVID-19 health-related attitudes and behaviors. Specifically, a COVID-19 health numeracy x COVID-19-related anxiety variable will be entered as a predictor of COVID-19 health-related attitudes and behaviors, and a COVID-19 health-related attitudes and behaviors x COVID-19-related anxiety variable will be entered as a predictor of COVID-19 health numeracy. In such a case, a significant coefficient between the interaction terms and COVID-19 health numeracy or COVID-19 health-related attitudes and behaviors would suggest that there is a significant interaction.

Gender. COVID-19 health numeracy and COVID-19 health attitudes and behaviors and attitudes may differ as a function of a participants' gender (Halpern et al., 2007; Lusardi, 2012). As such, the effects of gender will be assessed by entering the variable as a covariate for all three paths. Specifically, a basic numeracy x gender interaction term will be entered as a predictor of COVID-19 health numeracy and COVID-19 health attitudes and behaviors, a COVID-19 health numeracy x gender interaction term will be entered as a predictor of basic numeracy and COVID-19 health attitudes and behaviors, and a COVID-19 health attitudes and behaviors x gender interaction term will be entered as a predictor of basic numeracy and COVID-19 health numeracy. In such a case, a significant coefficient between the interaction terms and any of the variables of interest would indicate that there is a significant gender interaction.

Participant Exclusion Criteria and Basic Numeracy

One potential concern regarding the measurement of basic numeracy is that participants may have some trouble understanding and completing the number line question. Specifically, if participants failed to understand or misunderstood the instructions to the number line questions, their performance for the number line task would substantially differ from their performance in the non-number line basic numeracy questions. Three potential exclusion criterion may be capturing the failure to understand the number line question instruction: 1) the criterion that excludes participants with more than 25% missing data (as one can reason that participants may omit answering the number line questions if they do not understand the instructions), 2) the criterion that excludes participants based on Mahalanobis distance (as participants with a discrepancy between number line question and non-number line question performance would likely be flagged as a multivariate outlier), and 3) the criterion that excludes participants whose median absolute deviation of the number line task is below 0.1 (as participants who did not

understand the instructions for the number line task may “respond” by selecting the same point on the number line as a non-response).

This hypothesis will be explored by regressing the mean-score of the non-number line numeracy questions on three dichotomous variables each representing one of the exclusion criteria mentioned above. If a significant relation is obtained, it would indicate that indeed the participant exclusion criterion truncated the lower end of basic numerical abilities. We will account for this possibility in two ways. First, we will re-run the main analysis without the statistically significant exclusion criterion. Any discrepancy between this new analysis and the main analysis will be reported. Second, we will alternatively operationalize the basic numeracy latent factor without the number line questions and report any discrepancy found between the new analysis and the main analysis.

Additional Exploratory Variables

Due to the fast-changing nature of the COVID-19 pandemic, additional exploratory variables were added to the current study as part of the stage 1 registered report revisions. Specifically, we have added a number of items from the COVID States Project (Lazer et al., 2020). We have added two general questions regarding participants’ recent stay-at-home behaviors. The first item asks, “In the last 24 hours, how many people did you meet with that are not part of your immediate household?” Participants respond with 0, 1, 2, 3, 4-9, and > 9 as choices. The second item asks, “In the last 24 hours, have you been in a room (or another enclosed space) with people who were not members of your household? This might have been at a social gathering, a work meeting, or another type of even.” Participants are given “No, I have not,” “Yes, with 1-2 other people,” “Yes, with 3-4 other people,” “Yes, with 5-6 other people,” “Yes, with 7-8 other people,” “Yes, with 9-10 other people,” “Yes, with 11-50 other people,” “Yes, with over 100 other people,” as choices. Further, we have added a battery of yes/no questions asking, “In the last 24 hours, did you or any members of your household do any of the following activities outside of your home? Activities include going to work, going to the gym, visiting a friend, going to the doctor or visiting a hospital, going to a café, bar or restaurant, going to church or another place of worship, and taking mass transit. These items will be combined with our existing survey items about social distancing to create a social distancing index similar to Lazer et al. (2020).

Table 1. Design Table

Question	Hypothesis	Sampling Plan (power analysis)	Analysis Plan	Interpretation given to Different outcomes
Whether basic numeracy, COVID-19 health numeracy, and COVID-19 health-related attitudes and behaviours are intercorrelated.	Hyp. 1. Basic numeracy is positively correlated with COVID-19 health numeracy	A Monte-Carlo study was conducted by specifying the smallest effect size of interest for the relationship the two variables and determining how many participants are required to detect a statistically significant effect for path a 95% of the time out of 10,000 replications.	First, the three main variables will be entered in an SEM model to examine the intercorrelations. Next, the robustness of these correlations will be examined by entering covariates as regressors for the main variables.	Significant positive correlation: participants who score higher on the basic numeracy measures generally score higher for the COVID-19 health numeracy measures Significant negative correlation: participants who score higher on the basic numeracy measures generally score lower for the COVID-19 health numeracy measures Non-significant coefficient: Undetermined relationship between the variables
	Hyp. 2. Basic numeracy is positively correlated with COVID-19 health-related attitudes and behaviours	A Monte-Carlo study was conducted by specifying the smallest effect size of interest for the relationship the two variables and determining how many participants are required to detect a statistically significant effect for path a 95% of the time out of 10,000 replications.	First, the three main variables will be entered in an SEM model to examine the intercorrelations. Next, the robustness of these correlations will be examined by entering covariates as regressors for the main variables.	Significant positive correlation: participants who score higher on the basic numeracy measures generally score higher for the COVID-19 health-related attitudes Significant negative correlation: participants who score higher on the basic measures generally score lower for the COVID-19 health-related attitudes and behaviours measures Insignificant coefficient: Undetermined relationship between the variables
	Hyp. 3. COVID-19 health numeracy is positively correlated with COVID-19 health-related attitudes and behaviours	A Monte-Carlo study was conducted by specifying the smallest effect size of interest for the relationship the two variables and determining how many participants are required to detect a statistically significant effect for path a	First, the three main variables will be entered in an SEM model to examine the intercorrelations. Next, the robustness of these correlations will be examined by entering covariates as regressors for the main variables.	Significant positive correlation: participants who score higher on the COVID-19 health numeracy measures generally score higher for the COVID-19 health-related attitudes Significant negative correlation: participants who score higher on COVID-19 health numeracy measures generally score

Are the measurement properties of the constructs invariant across countries?	Invariance of the measurement properties. The factor structure, factor loadings, intercepts, and residuals of indicator items for the latent variables of basic numeracy will be fully invariant across countries. The factor structure, factor loadings, intercepts, and residuals of indicator items for the latent variables of COVID-19 health numeracy and COVID-19 health-related attitudes and behaviours may be partially non-invariant across countries.	95% of the time out of 10,000 replications.	Standard procedure to test for between country invariance will be used. Increasingly stringent between-country constraints will be placed on the analytical structure. These stages include:  1) Configural Invariance - Restricting the overall factor structure (i.e., the number of factors and their indicator variables) to be the same between countries 2) Metric Invariance - Restricting the loadings of the indicator variables to be the same between countries 3) Scalar Invariance - restricting the intercepts of the indicator variables to be the same between countries Constraints are evaluated using the chi-square difference test, and differences in other fit indices (CFI and RMSEA) whereby the less restrictive model is iteratively compared with the more restrictive model. A significant chi-square difference test would indicate that the added restrictions significantly reduce model fit.	lower for the COVID-19 health-related attitudes and behaviours measures
				Insignificant coefficient: Undetermined relationship between the variables
				If configural invariance is not achieved, one or more of the indicator variables do not indicate for the same latent variable. As such, one may need to redefine the latent variables, or countries will need to be analyzed separately and between-country differences cannot be examined.
				If configural invariance is achieved, it suggests that the same latent construct could be constructed with the same manifest variables across groups.
				If metric invariance is achieved, it suggests the measurement properties of the latent variables are comparable between countries. This will allow for the examination of differences in paths A, B and C' between countries.
				If scalar invariance is achieved, it suggests that mean differences in the latent construct are based on an equivalent scale and allow for the examination of latent mean differences between countries

Do the parameter estimates in the mediation model differ across countries?	We expect that there may be between-country differences in the strength of association between the main variables.	A Monte-Carlo study was conducted by specifying the smallest effect size of interest for the between country differences in intercorrelations.	In the event that some indicators are found to be non-invariant, we will test for partial invariance or remove the items prior to analysis of the full mediation model.	A significant difference between countries: the link between the latent variables are stronger for one country as compared to another. A non-significant difference between countries: insufficient evidence to suggest there are differences between countries.
	There may be mean differences between the latent factors	A Monte-Carlo study was used to simulate latent mean differences.	A model with latent means for both countries fixed to be equal will be compared with another model with latent means fixed to 0 for one country and freely estimated in the other. Constraints are evaluated using the chi-square difference test, and differences in other fit indices (CFI and RMSEA) whereby the less restrictive model is iteratively compared with the more restrictive model. A significant chi-square difference test would indicate that the added restrictions significantly reduce model fit.	Significant mean differences between different countries: The average level of basic numeracy, COVID-19 health numeracy, or COVID-19 health-related attitudes and behaviours are different between countries Non-significant mean differences between different countries: Insufficient evidence to suggest there are differences in the average level of basic numeracy, COVID-19 health numeracy, or COVID-19 health-related attitudes and behaviours

Data availability

All raw data (including pilot data) will be made available on the project's OSF page (https://osf.io/qpdnt/?view_only=08ad84266beb41eaa5113d0acb2f890) upon acceptance of the Stage 2 manuscript.

Code availability

All code will be made available on the project's OSF page (https://osf.io/qpdnt/?view_only=08ad84266beb41eaa5113d0acb2f890) upon acceptance of the Stage 2 manuscript.

Acknowledgements

The cost of the data collection will be supported by grants from the Canadian Institutes for Advanced Research as well as the Canada Research Chairs Program to DA. CAT is supported in part by a U.S. Department of Education Institute of Education Sciences (IES) Grant R305A160295 at Kent State University. EDW is the recipient of a Banting Postdoctoral Fellowship (NSERC) and a BrainsCAN Postdoctoral Fellowship at Western University, funded by the Canada First Research Excellence Fund (CFREF). MS is the recipient of a BrainsCAN Postdoctoral Fellowship at Western University, funded by the Canada First Research Excellence Fund (CFREF). LP is supported by a Brain and Mind Institute Postdoctoral Fellowship at Western University and by a Postdoctoral Trainee Award from the Children's Health Research Institute. JVH is supported by a postdoctoral fellowship from the Research Foundation-Flanders (FWO). ISA is supported by CAPES (Doc-Pleno, 88881.128282/2016-01). IC is supported by a Postdoctoral fellowship funded by Agence Nationale de la Recherche (ANR-18-CE28-0003). AR is supported by an NIH NICHD F32 award (HD102106-01) as well as a Scholar Award from the James S. McDonnell Foundation to Dr. Melissa Libertus. We thank Bea Goffin for comments on drafts of the manuscript.

Author contributions

The Authors' contribution was calculated according to the CRediT, the Contributor Roles Taxonomy, which can be found here:

https://osf.io/qpdnt/?view_only=08ad84266beb41eaa5113d0acb2f890.

Competing interests

The authors declare no competing interests.

References

- Adler, N. E., Epel, E. S., Castellazzo, G., & Ickovics, J. R. (2000). Relationship of subjective and objective social status with psychological and physiological functioning: Preliminary data in healthy, White women. *Health Psychology, 19*(6), 586–592.
<https://doi.org/10.1037/0278-6133.19.6.586>
- Ali, R. (2020). *Comparing coronavirus deaths to drowning, auto accidents were probably bad examples.* <https://www.usatoday.com/story/entertainment/celebrities/2020/04/17/dr-phil-compares-coronavirus-deaths-car-accidents/5151534002/>
- Allcott, H., Boxell, L., Conway, J., Gentzkow, M., Thaler, M., & Yang, D. Y. (2020). Polarization and public health: Partisan differences in social distancing during the Coronavirus pandemic. *NBER Working Paper, w26946.*
- Aust, F., Diedenhofen, B., Ullrich, S., & Musch, J. (2013). Seriousness checks are useful to improve data validity in online research. *Behavior Research Methods, 45*(2), 527–535.
- Ballard, C. L., & Johnson, M. F. (2004). Basic math skills and performance in an introductory economics class. *The Journal of Economic Education, 35*(1), 3–23.
- Barends, A. J., & de Vries, R. E. (2019). Noncompliant responding: Comparing exclusion criteria in MTurk personality research to improve data quality. *Personality and Individual Differences, 143*, 84–89.
- Braithwaite, D. W., & Siegler, R. S. (2018). Developmental changes in the whole number bias. *Developmental Science, 21*(2), e12541.
- Brust-Renck, P. G., Nolte, J., & Reyna, V. F. (2017). Numeracy in Health and Risk Messaging. In *Oxford Research Encyclopedia of Communication.*
- Chen, F. F. (2007). Sensitivity of goodness of fit indexes to lack of measurement invariance. *Structural Equation Modeling: A Multidisciplinary Journal, 14*(3), 464–504.
- Cheung, G. W., & Rensvold, R. B. (2002). Evaluating goodness-of-fit indexes for testing measurement invariance. *Structural Equation Modeling, 9*(2), 233–255.

- Cohen, J. (2013). *Statistical power analysis for the behavioral sciences*. Routledge.
- Cokely, E. T., Galesic, M., Schulz, E., Ghazal, S., & Garcia-Retamero, R. (2012). Measuring risk literacy: The Berlin numeracy test. *Judgment and Decision Making*.
- Czeisler, M. É., Tynan, M. A., Howard, M. E., Honeycutt, S., Fulmer, E. B., Kidder, D. P., Robbins, R., Barger, L. K., Facer-Childs, E. R., & Baldwin, G. (2020). Public attitudes, behaviors, and beliefs related to COVID-19, stay-at-home orders, nonessential business closures, and public health guidance—United States, New York City, and Los Angeles, May 5–12, 2020. *Morbidity and Mortality Weekly Report*, 69(24), 751.
- Davey, H. M., Barratt, A. L., Butow, P. N., & Deeks, J. J. (2007). A one-item question with a Likert or Visual Analog Scale adequately measured current anxiety. *Journal of Clinical Epidemiology*, 60(4), 356–360.
- Deck, C., & Jahedi, S. (2015). The effect of cognitive load on economic decision making: A survey and new experiments. *European Economic Review*, 78, 97–119.
- Dowker, A., & Nuerk, H.-C. (2016). Linguistic influences on mathematics. *Frontiers in Psychology*, 7, 1035.
- Finch, J. F., West, S. G., & MacKinnon, D. P. (1997). Effects of sample size and nonnormality on the estimation of mediated effects in latent variable models. *Structural Equation Modeling: A Multidisciplinary Journal*, 4(2), 87–107.
<https://doi.org/10.1080/10705519709540063>
- Fornell, C., & Larcker, D. F. (1981). Evaluating structural equation models with unobservable variables and measurement error. *Journal of Marketing Research*, 18(1), 39–50.
- Galesic, M., & Garcia-Retamero, R. (2011). Graph literacy: A cross-cultural comparison. *Medical Decision Making*, 31(3), 444–457.
- Geary, D. C. (2011). Cognitive predictors of achievement growth in mathematics: A 5-year longitudinal study. *Developmental Psychology*, 47(6), 1539.

- Hair Jr, J. F., Hult, G. T. M., Ringle, C., & Sarstedt, M. (2016). *A primer on partial least squares structural equation modeling (PLS-SEM)*. Sage publications.
- Halpern, D. F., Benbow, C. P., Geary, D. C., Gur, R. C., Hyde, J. S., & Gernsbacher, M. A. (2007). The science of sex differences in science and mathematics. *Psychological Science in the Public Interest*, 8(1), 1–51.
- Hu, L., & Bentler, P. M. (1999). Cutoff criteria for fit indexes in covariance structure analysis: Conventional criteria versus new alternatives. *Structural Equation Modeling: A Multidisciplinary Journal*, 6(1), 1–55. <https://doi.org/10.1080/10705519909540118>
- Huber, S., Nuerk, H.-C., Reips, U.-D., & Soltanlou, M. (2019). Individual differences influence two-digit number processing, but not their analog magnitude processing: A large-scale online study. *Psychological Research*, 83(7), 1444–1464.
- Inglehart, R., Haerpfer, C., Moreno, A., Welzel, C., Kizilova, K., J, D.-M., Lagos, M., Norris, P., Ponarin, E., & Puranen, B. (2014). *World Values Survey: Round Six—Country-Pooled Datafile 2010-2014*. [Database]. Madrid: JD Systems Institute.
<http://www.worldvaluessurvey.org/WVSDocumentationWV6.jsp>
- Jasper, J. D., Bhattacharya, C., Levin, I. P., Jones, L., & Bossard, E. (2013). Numeracy as a predictor of adaptive risky decision making. *Journal of Behavioral Decision Making*, 26(2), 164–173.
- Jung, E., & Yoon, M. (2016). Comparisons of three empirical methods for partial factorial invariance: Forward, backward, and factor-ratio tests. *Structural Equation Modeling: A Multidisciplinary Journal*, 23(4), 567–584.
- Keller, C., & Siegrist, M. (2009). Effect of risk communication formats on risk perception depending on numeracy. *Medical Decision Making*, 29(4), 483–490.
- Kline, R. B. (2015). *Principles and practice of structural equation modeling*. Guilford publications.

- Landy, D., Charlesworth, A., & Ottmar, E. (2017). Categories of large numbers in line estimation. *Cognitive Science*, *41*(2), 326–353.
- Landy, D., Silbert, N., & Goldin, A. (2013). Estimating large numbers. *Cognitive Science*, *37*(5), 775–799.
- Lazer, D., Santillana, M., Perlis, R. H., Quintana, A., Ognyanova, K., Green, J., Baum, M. A., Simonson, M., Uslu, A. A., Chwe, H., Druckman, J., Lin, J., & Gitomer, A. (2020). *THE COVID STATES PROJECT: A 50-STATE COVID-19 SURVEY REPORT #26: TRAJECTORY OF COVID-19-RELATED BEHAVIORS* (No. 26; The COVID-19 Consortium for Understanding the Public's Policy Preferences Across States, p. <https://covidstates.org>). Northeastern University.
- Lewis, M. R., Matthews, P. G., & Hubbard, E. M. (2016). Neurocognitive architectures and the nonsymbolic foundations of fractions understanding. In *Development of Mathematical Cognition* (pp. 141–164). Elsevier.
- Leys, C., Ley, C., Klein, O., Bernard, P., & Licata, L. (2013). Detecting outliers: Do not use standard deviation around the mean, use absolute deviation around the median. *Journal of Experimental Social Psychology*, *49*(4), 764–766.
<https://doi.org/10.1016/j.jesp.2013.03.013>
- Little, T. D., Cunningham, W. A., Shahar, G., & Widaman, K. F. (2002). To parcel or not to parcel: Exploring the question, weighing the merits. *Structural Equation Modeling*, *9*(2), 151–173.
- Ludewig, U., Lambert, K., Dackermann, T., Scheiter, K., & Möller, K. (2019). Influences of basic numerical abilities on graph reading performance. *Psychological Research*, 1–13.
- Lusardi, A. (2012). *Numeracy, financial literacy, and financial decision-making*. National Bureau of Economic Research.

- Mackinnon, D. P., Lockwood, C. M., & Williams, J. (2004). Confidence limits for the indirect effect: Distribution of the product and resampling methods. *Multivariate Behavioral Research*, 39(1), 99–128.
- Maniaci, M. R., & Rogge, R. D. (2014). Caring about carelessness: Participant inattention and its effects on research. *Journal of Research in Personality*, 48, 61–83.
- Matthews, P. G., Lewis, M. R., & Hubbard, E. M. (2016). Individual differences in nonsymbolic ratio processing predict symbolic math performance. *Psychological Science*, 27(2), 191–202.
- McDonald, R. P. (2013). *Test theory: A unified treatment*. psychology press.
- Meade, A. W., & Craig, S. B. (2012). Identifying careless responses in survey data. *Psychological Methods*, 17(3), 437.
- Muthén, L. K., & Muthén, B. O. (2010). *Mplus: Statistical analysis with latent variables: User's guide*. Muthén & Muthén Los Angeles.
- Ni, Y., & Zhou, Y.-D. (2005). Teaching and learning fraction and rational numbers: The origins and implications of whole number bias. *Educational Psychologist*, 40(1), 27–52.
- Núñez-Peña, M. I., Guilera, G., & Suárez-Pellicioni, M. (2014). The single-item math anxiety scale: An alternative way of measuring mathematical anxiety. *Journal of Psychoeducational Assessment*, 32(4), 306–317.
- Nye, C. D., & Drasgow, F. (2011). Effect size indices for analyses of measurement equivalence: Understanding the practical importance of differences between groups. *Journal of Applied Psychology*, 96(5), 966.
- Peters, E., Hibbard, J., Slovic, P., & Dieckmann, N. (2007). Numeracy skill and the communication, comprehension, and use of risk-benefit information. *Health Affairs*, 26(3), 741–748.

- Putnick, D. L., & Bornstein, M. H. (2016). Measurement invariance conventions and reporting: The state of the art and future directions for psychological research. *Developmental Review, 41*, 71–90.
- Reips, U.-D. (2000). The Web experiment method: Advantages, disadvantages, and solutions. In *Psychological experiments on the Internet* (pp. 89–117). Elsevier.
- Reyna, V. F., Nelson, W. L., Han, P. K., & Dieckmann, N. F. (2009). How numeracy influences risk comprehension and medical decision making. *Psychological Bulletin, 135*(6), 943.
- Rogers, W. M., & Schmitt, N. (2004). Parameter recovery and model fit using multidimensional composites: A comparison of four empirical parceling algorithms. *Multivariate Behavioral Research, 39*(3), 379–412.
- Schneider, M., Merz, S., Stricker, J., De Smedt, B., Torbeyns, J., Verschaffel, L., & Luwel, K. (2018). Associations of number line estimation with mathematical competence: A meta-analysis. *Child Development, 89*(5), 1467–1484.
- Schwartz, S. R., McDowell, J., & Yueh, B. (2004). Numeracy and the shortcomings of utility assessment in head and neck cancer patients. *Head & Neck: Journal for the Sciences and Specialties of the Head and Neck, 26*(5), 401–407.
- Shepherd, M. (2020). *Is poor math literacy making it harder for people to understand COVID-19 Coronavirus?* <https://www.forbes.com/sites/marshallshepherd/2020/03/23/is-the-math-too-hard-for-people-to-understand-covid-19-coronavirus/#34da88bb6a9c>
- Siegel, E. (2020). *Why exponential growth is so scary for the COVID-19 coronavirus.* <https://www.forbes.com/sites/startswithabang/2020/03/17/why-exponential-growth-is-so-scary-for-the-covid-19-coronavirus/#367dd4ad4e9b>
- Spiegelhalter, D., Pearson, M., & Short, I. (2011). Visualizing uncertainty about the future. *Science, 333*(6048), 1393–1400.
- Steenkamp, J.-B. E., & Baumgartner, H. (1998). Assessing measurement invariance in cross-national consumer research. *Journal of Consumer Research, 25*(1), 78–90.

- Tabachnick, B. G., Fidell, L. S., & Ullman, J. B. (2007). *Using multivariate statistics* (Vol. 5). Pearson Boston, MA.
- Thompson, C. A., Taber, J., Coifman, K., & Sidney, P. (2020). *Math misconceptions may lead people to underestimate the true threat of COVID-19*. <https://theconversation.com/math-misconceptions-may-lead-people-to-underestimate-the-true-threat-of-covid-19-134520>
- Thompson, C. A., Taber, J. M., Sidney, P. G., Fitzsimmons, C., Mielicki, M., Matthews, P. G., Schemmel, E., Simonovic, N., Foust, J., & Aurora, P. (2020). *Math matters during a pandemic: A novel, brief educational intervention combats whole number bias to improve health decision-making and predicts COVID-19 risk perceptions and worry across 10 days*.
- Van Hoof, J., Verschaffel, L., & Van Dooren, W. (2015). Inappropriately applying natural number properties in rational number tasks: Characterizing the development of the natural number bias through primary and secondary education. *Educational Studies in Mathematics*, *90*(1), 39–56.
- Vandenberg, R. J., & Lance, C. E. (2000). A review and synthesis of the measurement invariance literature: Suggestions, practices, and recommendations for organizational research. *Organizational Research Methods*, *3*(1), 4–70.
- Weinfurt, K. P., Castel, L. D., Li, Y., Sulmasy, D. P., Balshem, A. M., Benson III, A. B., Burnett, C. B., Gaskin, D. J., Marshall, J. L., & Slater, E. F. (2003). The correlation between patient characteristics and expectations of benefit from Phase I clinical trials. *Cancer*, *98*(1), 166–175.
- Wolf, M. S., Serper, M., Opsasnick, L., O’Conor, R. M., Curtis, L. M., Benavente, J. Y., Wismer, G., Batio, S., Eifler, M., & Zheng, P. (2020). Awareness, attitudes, and actions related to COVID-19 among adults with chronic conditions at the onset of the US outbreak: A cross-sectional survey. *Annals of Internal Medicine*.

Woloshin, S., Schwartz, L. M., Moncur, M., Gabriel, S., & Tosteson, A. N. (2001). Assessing values for health: Numeracy matters. *Medical Decision Making*, 21(5), 382–390.

Appendix F

1. Regarding participant exclusions: please confirm that what you did here is what was pre-registered. If there was a deviation, please label it as a deviation. The final text should say something like, “As pre-registered...[what you did]. Additionally, we later decided to ... [how you deviated].” If there are extensive deviations from the pre-registered plan, then report the main analyses following the pre-registered plan alongside analysis based on the subset of participants with the authors’ preferred deviation as a robustness check.

Thank you for the comment. The participant exclusion criteria that we have included were formulated at two time points: 1) at stage 1 submission, and 2) after we have conducted a preliminary measurement study as part of stage 1 revisions. Therefore, all exclusion criteria were part of the pre-registration plan. We now see why this may be somewhat confusing to the reader. We have changed the manuscript to clarify this to the reader.

The manuscript is as follows:

“As preregistered, we used several criteria to exclude inattentive participants (Huber et al., 2019)...”

“Two additional exclusion criteria were added in light of the results in the preliminary measurement study (as part of stage 1 manuscript revisions)...”

2. Treatment of missing data: I do not believe that analysis of missing data was part of the pre-registered plan (but my memory might be poor). If I am correct, then the decision to include additional covariates in the analysis is a deviation from the pre-registered plan. If I am correct, then it should be marked as a deviation, and as with participant exclusions, results should be presented following the pre-registered plan with the deviation provided as a robustness check.

Thank you for the comment. The analysis of missing data and the inclusion of additional covariates as auxiliary variables was indeed preregistered. We have altered the section to clarify this to the reader:

“As preregistered, we conducted a missing values analysis to identify patterns of missing data in the items....”

3. Item parcels: You made the choice (seemingly post hoc, because the text is in red) to standardize items prior to parcel construction. Was this standardization done within each country separately? Or in the sample as a whole? In general, standardization when doing analyses with nested data (here, participants within countries) is a no-no because it changes the variance structure of the data. Relatedly, the descriptive statistics in Table 1 apparently reflect this standardization, obscuring relevant information like how accurate participants actually were in their numeracy knowledge. Unlike so much other psychology research, the scale of measurement here is not arbitrary, so the raw means and standard deviations are of interest. Therefore, I recommend a supplemental table with raw descriptive statistics, separated by country. I also recommend reconsidering the standardization choice prior to parceling for the main analysis. Is there another way to score the items before combining them that respects the multi-level data structure? Perhaps POMP

(percentage of maximum possible)? This critique also applies to the measurement pilot study (Table S2).

Thank you for the comment. Standardization was done for the whole sample. We have done this because some of the questions in the survey have different scaling and may bias results. Upon further consideration, we agree that due to the multilevel nature of the data, it is not ideal to standardize items prior to constructing parcels.

As such, we have redone the analysis with parcels calculated using the raw scores instead (using raw scores was also the approach used for the preliminary analysis). A comparison of the two analyses (parceling using raw scores vs. standardized scores) seem to suggest that standardization did not significantly alter the results.

Relatedly, we have revised table 1 so that items in table displayed descriptive statistics for the raw scores.

4. Related to the footnote in Table 1, are these statistics (means/variances/skew/kurtosis) raw values from the data, or estimates from a model? If from a model, provide the raw values somewhere, preferably stratified by country.

Thank you for the comment. They are raw values. We have added an appendix to include descriptive statistics stratified by country.

5. I like the way the two-factor vs. three-factor model for numeracy was handled (page 32). Make sure to correct the invariance analyses in Appendix C per the notes in point 9 below.

Thank you for the comment. The changes have been done for both 2-factor and 3-factor models.

6. I'm not sure why correlations are tested before invariance testing is done. This might not be a big deal, but it confused me.

Thank you for the comment. We agree this is somewhat cumbersome, but this is the order of analysis that has been preregistered (please see the flowchart in Figure 2) As such, we feel it is best if we were to follow this flow chart of analysis.

7. Table 4 duplicates information from the text. I prefer the table and am not sure the repeat of the fit statistics is needed in both spots.

Thank you for the comment. While we agree the text and table 4 present the same information, the text include information on the chi-square difference test which is not conveyed in the table. Further, a written version of the results may aid readers in understanding what has been done.

8. On p. 18, you report that strict invariance (i.e., equivalent residuals) will not be tested, yet strict invariance appears in Table 4. Then on p. 36, you use "strict invariance" in a distinct way, to refer to latent variance and path equality. This analysis should be relabeled, because as far as I know,

most people use “strict” invariance to refer to equivalent residuals, and not to equal latent variances and path equality.

Thank you for the comment. The sentence regarding not testing strict invariance was from a previous iteration of the manuscript. As we have made it clear in Figure 2 and subsequent description of analysis that we will explore latent means (requiring strict invariance) we have deleted the sentence regarding not testing strict invariance. Thank you for catching that mistake.

Further, we have revised the text to distinguish between strict invariance and variance-covariance equality.

The section as described is as follows:

“

Having established that there were no between-country differences in the relations between the main variables, we next explored whether there were latent mean differences between countries. To this end, we first tested whether strict invariance and variance-covariance equality could be established. Test for latent strict invariance and path equality was assessed through the chi-square difference test and change in model fit when compared with the scalar invariant model. Model fit for the strict invariance model was good, ($\chi^2(114) = 386.041, p < .001, CFI = .979, TLI = .980, RMSEA = .056$). Comparison between models revealed a significant chi-square difference test, ($\chi^2(18) = 64.962, p < .001$), but no significant change in CFI and RMSEA, ($\Delta CFI = .004, \Delta RMSEA = .001$). As such, results indicated that strict invariance was achieved.

Model fit for the variance-covariance equality model was good, ($\chi^2(120) = 367.285, p < .001, CFI = .978, TLI = .981, RMSEA = .055$). Comparison between models revealed an insignificant chi-square difference test, ($\chi^2(6) = 8.527, p < .202$), and no significant change in CFI and RMSEA, ($\Delta CFI = .001, \Delta RMSEA = .001$). As such, results indicated that path equality was achieved.

”

9. Degrees of freedom in Table 4. The change in df from configural to metric is 12, whereas the

change from metric to scalar is 18. I believe this reflects a specification error, because the change in df for both of these steps should be the same. Because you identified your factors with latent mean = 0 and variance = 1, only one country's latent variance (and then mean) should be so fixed when the equality constraint for loadings (and then intercepts) are added. So the invariance analyses will need to be re-run and re-reported. I didn't check, but this may affect the pilot study as well. This issue may also have downstream effects on the main hypothesis tests, but I didn't look into things sufficiently to determine for sure. Basically if the scalar model was misspecified, then the subsequent analyses that come after this point might also be misspecified. You have to count the change in df from one model to the next to be sure. What you report doing in the last paragraph on p. 36 is what should be done in the scalar model. Latent means are relaxed as equality constraints for intercepts are added, then later when you want to test the equality of latent means, an equality constraint for the latent means is re-added. That test – the difference between the model with the latent means forced to equality vs. one indicator country – is missing, meaning that there is currently no test of your claim that “participants in both Canada and the UK had higher COVID-19 Health Numeracy Skills when compared with participants in the US. While the UK had higher Basic Numeracy Skills than participants in the US but not Canada” (p. 36).

Thank you for spotting the mistake. We have revised and redone the invariance test. Results suggest that the misspecifications did not significantly alter the results of the current study. Invariance testing were not done in the pilot study due to a lack of sample size.

Relatedly, we have conducted the test where a model with constrained latent means versus a model with unconstrained latent means and have found results suggesting that constraining latent means to be equal does not significantly reduce model fit. As such, we have removed the analysis regarding between country differences in latent means.

10. Exploration begins on page 38 (per the paper's headings), but several references to exploration appear in the previous sections of the paper. I believe it would be cleaner/clearer if all of the exploratory analyses were separated from the confirmatory analyses. As it stands, it is hard to find the confirmatory tests, given that they are mixed in with other analyses.

Thank you for the comment. We have changed the verbiage in the results and label all confirmatory analysis as such.

11. Page 42 refers to a “negative correlation ... between basic numeracy and COVID-19 attitudes and behaviours” but the correlation you found was not statistically different from zero, making this discussion confusing and possibly misleading.

Thank you for the comment. We have removed that part of the discussion.

12. An additional unmentioned limitation is shortcomings related to the COVID attitudes and behavior measure. If that measure is unreliable, we would not expect correlations with it to be very strong. I might have missed it, but is there discussion of the reliability/validity of that measure somewhere? By validity, I do not mean distinctiveness from the numeracy measures, but rather

accuracy of the measure as established (perhaps in other research) by relations with other measures. The ceiling effect on this measure could also substantially limit its correlations with other measures.

Thank you for the comment. We have expanded the limitation section of the paper to include these points. The section, as described is as follows:

“Finally, while our study has utilized existing measures of basic numeracy, measurements of COVID-19 health related numeracy and COVID-19 attitudes and behaviors have been adapted from related sources. As such, it may be the case that the measurement of COVID-19 health related numeracy and COVID-19 attitudes and behaviors may not match the intended underlying concepts. However, this risk may be reduced as the questions, at face value, does clearly communicate to readers the intent of the survey. Nevertheless, future studies would be required to assess construct validity, such as examining our measures with other established measures of numeracy and health numeracy.”

13. Page 23: “missing random responses throughout the questionnaire” – consider changing to “missing sporadic responses throughout the questionnaire” because random has specific connotations in this context.

Thank you for the comment. We agree and the text has been amended.